DOI: 10.1038/s41467-018-05121-8　　**OPEN**

# Single neurons may encode simultaneous stimuli by switching between activity patterns

Valeria C. Caruso [1,2,3,4], Jeff T. Mohl [1,2,3,4], Christopher Glynn [5,9], Jungah Lee [1,2,3,4,10], Shawn M. Willett [1,2,3,4], Azeem Zaman [5,11], Akinori F. Ebihara [6], Rolando Estrada [7,8], Winrich A. Freiwald [6], Surya T. Tokdar [1,5] & Jennifer M. Groh [1,2,3,4]

How the brain preserves information about multiple simultaneous items is poorly understood. We report that single neurons can represent multiple stimuli by interleaving signals across time. We record single units in an auditory region, the inferior colliculus, while monkeys localize 1 or 2 simultaneous sounds. During dual-sound trials, we find that some neurons fluctuate between firing rates observed for each single sound, either on a whole-trial or on a sub-trial timescale. These fluctuations are correlated in pairs of neurons, can be predicted by the state of local field potentials prior to sound onset, and, in one monkey, can predict which sound will be reported first. We find corroborating evidence of fluctuating activity patterns in a separate dataset involving responses of inferotemporal cortex neurons to multiple visual stimuli. Alternation between activity patterns corresponding to each of multiple items may therefore be a general strategy to enhance the brain processing capacity, potentially linking such disparate phenomena as variable neural firing, neural oscillations, and limits in attentional/memory capacity.

[1] Duke Institute for Brain Sciences, Duke University, Durham, NC 27708, USA. [2] Center for Cognitive Neuroscience, Duke University, Durham NC, 27708, USA. [3] Department of Psychology and Neuroscience, Duke University, Durham NC, 27708, USA. [4] Department of Neurobiology, Duke University, Durham NC, 27708, USA. [5] Department of Statistical Science, Duke University, Durham NC, 27708, USA. [6] The Rockefeller University, New York, New York NY, 10065, USA. [7] Department of Computer Science, Duke University, Durham NC, 27708, USA. [8] Department of Computer Science, Georgia State University, Atlanta, GA 30302, USA. [9] Present address: Department of Decision Sciences, University of New Hampshire, Durham NH, 03824, USA. [10] Present address: Department of Psychology, University of Canterbury, Riccarton, Christchurch 8041, New Zealand . [11] Present address: Department of Statistics, Harvard University, Cambridge MA, 02138, USA. Correspondence and requests for materials should be addressed to V.C.C. (email: v.caruso@duke.edu) or to S.T.T. (email: surya.tokdar@duke.edu) or to J.M.G. (email: jmgroh@duke.edu)

In the natural world many stimuli or events occur at the same time. Sensory neurons in the brain are broadly tuned and potentially responsive to more than one such stimulus, raising the question of how information about multiple simultaneous items can be preserved. We investigated whether the brain solves this problem at the neuronal level via activity patterns that fluctuate between those evoked by each stimulus alone (Fig. 1a). If present, such fluctuations could allow each individual stimulus to be represented across time in a common neuronal ensemble. Such switching would contrast with possibilities such as signal summation or normalization/averaging, either of which results in information about each item being lost. Fluctuation is more related to winner-take-all but suggests that the winning stimulus might change across time for an individual neuron[1–11].

Specifically, we tested how neurons in the monkey inferior colliculus (IC) respond when two sounds are presented simultaneously from different locations and both must be behaviorally reported. The IC is an early[12,13] and nearly obligatory station along the auditory pathway[14], and thought to be essential for accurate sound localization behavior[15–17]. Importantly, the IC is thought to encode sound location via a monotonic firing rate (meter) code in which individual neurons respond broadly across a range of spatial positions but the level of activity is correlated with the location of the sound[18–21]. We developed a novel statistical approach to assess whether signal fluctuations in the IC serve to interleave information about both sounds at a variety of time scales.

We found that a subpopulation of IC neurons exhibited fluctuating activity consistent with switching between individual sound responses at different time scales. Such activity fluctuations were masked in conventional analysis of spiking activity (trial-and-time pooled spike counts), which indicated that average dual-sound activity levels were intermediate between the average single-sound responses. The state of the network prior to stimulus onset (assessed by the mean voltage of the local field potentials) predicted the slower whole-trial spiking fluctuations, and in turn, these fluctuations could predict which sound location the monkey reported first. We replicated the key observations regarding fluctuating activity in a separate data set involving inferotemporal cortex: neurons confronted with multiple object stimuli exhibited activity fluctuations consistent with switching between individual object responses. These observations support fluctuating activity as a viable and likely a general strategy for encoding simultaneously presented stimuli. We conclude by discussing how such a code can be read out, especially with regard to such factors as time scale and coordination across ensembles of neurons. Finally, we consider several broad implications of activity fluctuations for interpreting variability and other aspects of neural encoding.

## Results

**Time-and-trial pooled activity does not account for behavior.** We first tested whether monkeys can perceptually preserve information about multiple sounds presented simultaneously. Two monkeys performed a localization task (Fig. 1b) in which they made eye movements to each of the sounds they heard: one saccade on single-sound trials and two saccades in sequence on dual-sound trials (Fig. 1c and Supplementary Figure 1). The sounds were separated horizontally by 30 degrees and consisted of band-limited noise with different central frequencies. The sounds were thus physically distinguishable in principle, and humans are able to do so[22–24]. The monkeys learned the task successfully, and, like humans, typically performed better when the frequency separation between the two sounds was larger (Fig. 1c, Supplementary Figure 1, ~72% vs. ~77% correct for frequency differences of 3.4 vs. 6.8 semitones).

If the monkeys can report the locations of two sounds presented simultaneously, it follows that their brains, and the IC in particular, must preserve information about both sound items. To investigate the neural basis of this perceptual ability, we recorded 166 single units from the left and right IC of two monkeys (see Methods), and evaluated the activity patterns evoked on dual-sound trials (involving a given pair of sound locations and frequencies, AB) in comparison to the activity patterns evoked on the matching single-sound trials (A and B). We refer to such related sets of stimuli as triplets.

Conventional analysis of spike data typically involves two simplifications: spikes are counted within a fairly long window of time, such as a few hundred milliseconds, and activity is pooled across trials for statistical analysis. If IC neurons interleave signals related to each of the two sounds, then they might appear to show averaging responses on dual (or AB) trials when activity is pooled across time and across trials. But they should not appear to show summation responses, i.e., in which the responses on dual-sound trials resemble the sum of the responses exhibited on single-sound trials involving the component sounds. Such summation has been observed in some neural populations in areas such as primary visual cortex[25,26], the hippocampus[27], and the superior colliculus[28] when multiple stimuli are presented.

Indeed, IC dual-sound responses do not generally appear to sum the responses to the component single sounds. Using an analysis similar to that of[28], dual-sound responses were converted to Z-scores relative to either the sum or the average of the corresponding single-sound responses (see Methods). For 81% of the tested triplets, the Z-score values relative to the average were smaller than those relative to the sum, indicating that the responses more closely resembled averaging (Fig. 1d, see also Supplementary Figure 2 and 3)

However, such apparent averaging response patterns appear inconsistent with the behavioral results: if the neurons truly responded at an average firing rate, then presumably the monkeys should respond to dual-sounds as if there were only a single sound at the midpoint of the two sources (Supplementary Figure 2A). Since monkeys can indicate the locations of both sounds (Fig. 1c), fluctuating activity patterns that could preserve both items across time in a neuronal ensemble might provide a better explanation for so-called averaging response patterns.

**Evaluating activity fluctuations at various time scales.** To determine whether neural activity fluctuates within and/or between trials, creating an overall averaging response but retaining information about each sound at distinct moments, we developed a series of statistical analyses that test for the presence of various forms of alternation in firing rates. Several unknown parameters must be taken into consideration when testing for activity fluctuations, specifically, the time scale, repeatability, and potential correlations across the neural population. We made minimal assumptions about the time scale at which neurons might alternate between encoding each stimulus, and considered that switching behaviors might vary from trial to trial and/or across time within a trial, as suggested by the visual inspection of single unit responses.

Figure 1e, f shows the activity of two example neurons on dual-sound trials compared to their matched single-sound trials. The colored backgrounds illustrate the median and 25–75% quantiles of the activity on single-sound trials, in 50 ms time bins. Superimposed on these backgrounds is the activity on individual trials. Individual single-sound (A alone, B alone) trials align well with their corresponding 25–75% quantiles, by definition (Fig. 1e, f). But on dual-sound (AB) trials, for any given trial or time bin, some individual traces correspond well to the 25–75% quantiles of one of the component sounds, and on other trials or time bins

they correspond well to the 25–75% quantiles of the other component sound. For the neuron in Fig. 1e, there are whole trials in which the activity matches that evoked by sound A alone and others in which it better corresponds to that evoked by sound B alone. For the neuron in Fig. 1f, the firing pattern on dual-sound trials appears to switch back and forth between the levels observed for sounds A and B as the trial unfolds. In short, for these two examples, the activity on dual-sound AB trials does not appear to occur at a consistent value intermediate between those evoked on single-sound A and B trials, but can fluctuate between those levels at a range of time scales. The following analyses will quantitatively assess these fluctuations across trials and/or time.

**Evidence for fluctuations in whole-trial spike counts.** If neurons alternate firing rates at the time scale of trials, as appears to be the case for the neuron in Fig. 1e, then the spike counts from dual-sound responses should resemble a mix of spike counts from each of the component single-sound responses. We statistically tested this hypothesis against other reasonable competing possibilities using the subset of triplets whose spike counts on single-sound A and B trials could be well modeled by Poisson distributions with statistically different mean rates $\lambda^A$ and $\lambda^B$ ($N = 363$ triplets from $N = 108$ neurons, see Methods for details).

The competing scenarios to describe the corresponding dual-sound trials were:

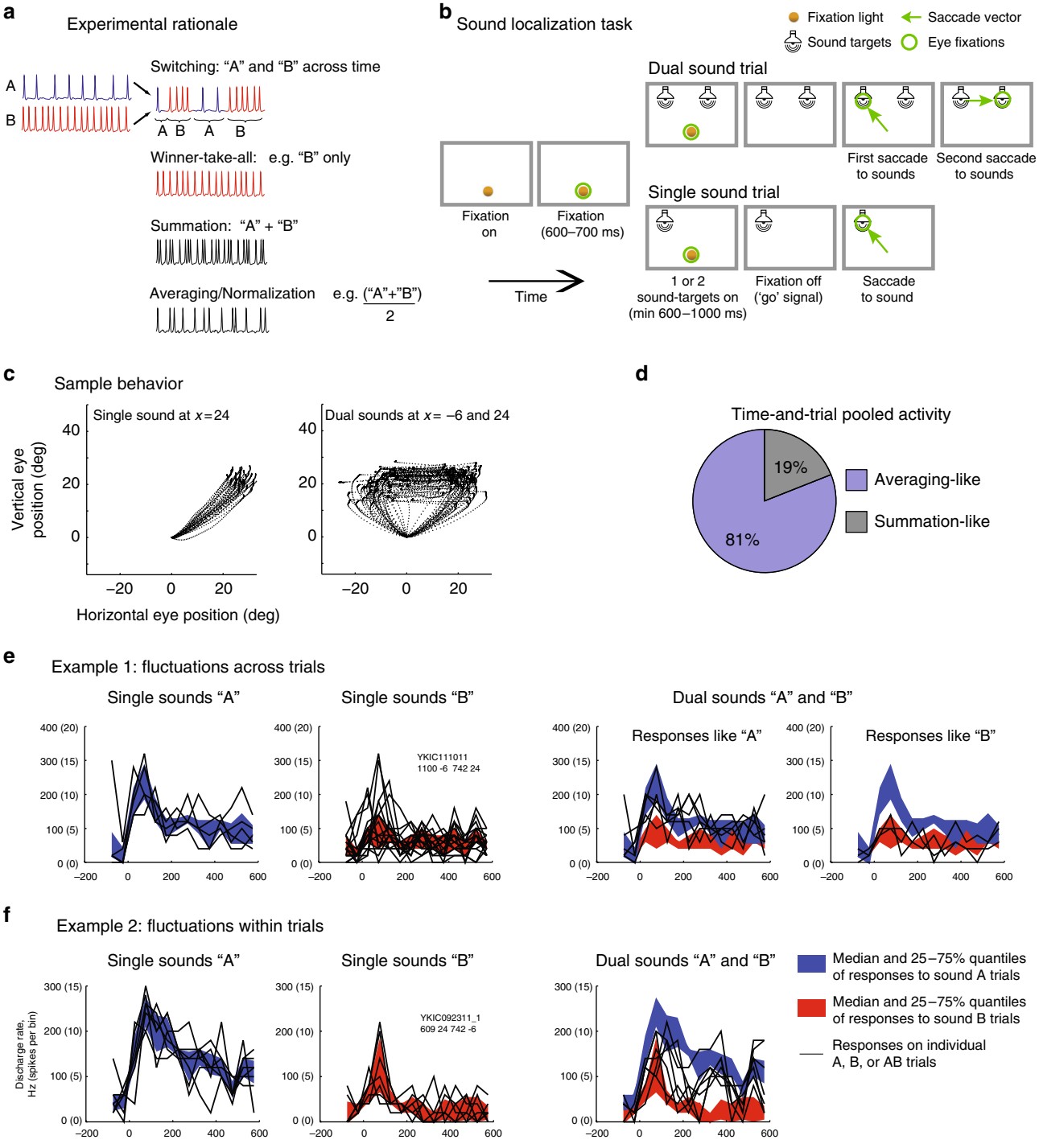

**Fig. 1** *(no caption text beyond panel labels)*

(a) **Mixture**: The spike counts observed on individual trials are best described as draws from a mixture of Poi($\lambda^A$) and Poi ($\lambda^B$) (Fig. 2a). This is consistent with fluctuating activity across trials.

(b) **Intermediate**: A single Poisson distribution best describes the spike counts, and this Poisson has a rate $\lambda^{AB}$ between $\lambda^A$ and $\lambda^B$ (Fig. 2b). This is consistent with either fluctuations at faster, sub-trial time scales or with true averaging/ normalization.

(c) **Outside**: A single Poisson distribution best describes the spike counts, but the rate $\lambda^{AB}$ is outside the range of $\lambda^A$ and $\lambda^B$ (i.e., it is greater than both or less than both; Fig. 2c). Summation-type responses would be captured under this heading, as would inhibitory interactions.

(d) **Single**: A single Poisson describes the dual-sound trial spike counts, but the rate $\lambda^{AB}$ is equal to one of the single-sound rates $\lambda^A$ or $\lambda^B$ (Fig. 2d). A winner- (or loser-)-take-all pattern would fit this category.

In summary, these four models capture the spectrum of possibilities at the whole-trial time scale. For each triplet, we computed the posterior probabilities of the four competing models as a (Bayesian) measure of goodness of fit to the data[29] (see Methods). In other words, for each triplet the model with the highest posterior probability can be interpreted as the best fit, and the posterior probability itself is a measure of the confidence in the model. We consider three levels of confidence from high to low (posterior probability >0.95, >0.50, and no threshold on the confidence level, that is, posterior probability > the minimum, 0.33, Fig. 2i).

The Mixture model had the strongest support of any of the models tested. For such triplets, the spike counts on dual-sound trials were better fit by a mixture of the single-sound Poisson distributions than by any single Poisson distribution (Fig. 2i, bar labeled Mixture). These response patterns indicate the presence of fluctuating activity at the level of individual trials; the neurons illustrated in Fig. 2e (same as Fig. 1e) and 2f (same as Fig. 3b) met these criteria. Of the 72 triplets (from $N = 46$ neurons) in which one model had a winning probability > 0.95, 50 triplets (69%) were categorized as Mixtures.

For the next largest category, the best fitting model involved a unique $\lambda^{AB}$ between $\lambda^A$ and $\lambda^B$ (Fig. 2i, bar labeled Intermediate). These triplets are ambiguous: they could exhibit a true intermediate firing rate on the dual-sound trials, or they could simply show alternation at a time scale more rapid than individual trials. The neurons illustrated in Fig. 2g (same as Figs. 1f and 3c) and 2h (same as Fig. 3d) were classified as Intermediate. Of the 72

triplets in which one model had a winning probability > 0.95, 18 triplets (25%) were categorized this way.

The remaining triplets were categorized as Single, or $\lambda^{AB} = \lambda^A$ or $\lambda^B$ (a narrowly defined category that consequently did not produce any winning model probabilities > 0.95) or Outside, $\lambda^{AB}$ greater or less than both $\lambda^A$ and $\lambda^B$. Single can be thought of as a winner-take-all response pattern. Outside may be consistent with a modest degree of summation in the neural population, particularly as $\lambda^{AB}$ was generally greater than both $\lambda^A$ and $\lambda^B$ in this subgroup. The small prevalence of the Single and Outside groups compared to the Mixture and Intermediates is in agreement with the aggregated analysis of spiking data (Supplementary Figures 2, 3). The overall pattern of classification in the whole-trial analysis was similar in the population of triplets that exhibited responses to both sounds individually vs. those that responded to only one of the two sounds (Supplementary Figure 5), with the exception that the bulk of the Outside triplets were indeed responsive to both A and B sounds individually.

**Evidence for within-trial fluctuations vs. stable averaging**. We next evaluated whether firing patterns fluctuated or remained stable across time within a trial. In particular, might triplets categorized as Intermediate in the whole trial analysis show evidence of fluctuating activity on a faster time scale?

We developed a novel statistical approach to study temporal patterns of the spike trains, the Dynamic Admixture of Poisson Process (DAPP) model. The computation is schematized in Fig. 3a (see also Methods). We focused on the same 363 triplets selected above. For each triplet, spike trains from individual single-sound trials were assumed to be independent realizations of a nonhomogeneous Poisson process with unknown time-dependent firing rates $\lambda^A(t)$ and $\lambda^B(t)$ for sounds A and B. Dual-sound trials were modeled as a weighted combination: $\lambda^{AB}(t) = \alpha(t)\lambda^A(t) + (1 - \alpha(t))\lambda^B(t)$ (Fig. 3a). The weight function $\alpha(t)$ was unique to each dual-sound trial and quantified the relative contribution of sound A on that trial at time $t$, while $1 - \alpha(t)$ quantified the complementary contribution of sound B. Thus, the dynamics of the $\alpha(t)$ function characterize the dynamics of each dual-sound trial (see examples in Fig. 3b–d).

For each selected triplet, we used a Bayesian analysis to estimate the function valued model parameters (see Methods) and predict the $\alpha(t)$ curves that the corresponding cell was likely to produce on future dual-sound AB trials (Fig. 3a). Each predicted curve was summarized by two features: its time average over the response period of a given trial and its maximum swing size, that is, the difference between its highest peak and lowest

**Fig. 1** Experimental rationale, task and visualization of individual trial activity. **a** In telecommunications, multiple signals can be conveyed along a single channel by interleaving samples of each, thus increasing the amount of information transmitted by a single physical resource. Here we investigated whether the brain might employ a similar strategy: do neurons encode multiple items (A and B) using spike trains that alternate between the firing rates corresponding to each item, at some unknown time scale? Such a strategy would preserve information about both items, in contrast to alternatives such as winner-take-all, summation, or averaging, which involve varying degrees of information loss. **b** Sound localization task. Two monkeys were successfully trained to report one or two simultaneous (bandlimited noise) sounds by saccading at them. See Supplementary Figure 1 for accuracy. **c** Eye traces of the saccades towards one (left) or two targets (right) during a sample session. **d** Time-and-trial aggregated dual-sound responses resemble the averaging more than the summation of single-sound responses. For 81% of the triplets tested, the absolute values of the Z-score of each dual-sound response relative to the average were smaller than those relative to the sum. **e**, **f** Visualization of individual trials of two IC neurons in which dual-sound responses alternate between firing rates corresponding to single-sounds, across trials for the neuron in **e**, or within trials for the neuron in **f**. In each panel, the red and blue shaded areas indicate the median and central 50% of the data on the single-sound trials. The black traces are the individual trials, for single-sound and dual-sound trials as indicated above the panel. For the neuron in **e**, individual traces on dual-sound trials were classified based on whether they matched the responses to single-sounds A and B (A vs. B assignment score, see Methods) and are plotted in two separate panels accordingly. For the neuron in **f**, the fluctuations occurred faster, within trials, and are plotted in the same panel. See Supplementary Figure 4 for peristimulus time histograms and frequency sensitivity of these two example neurons

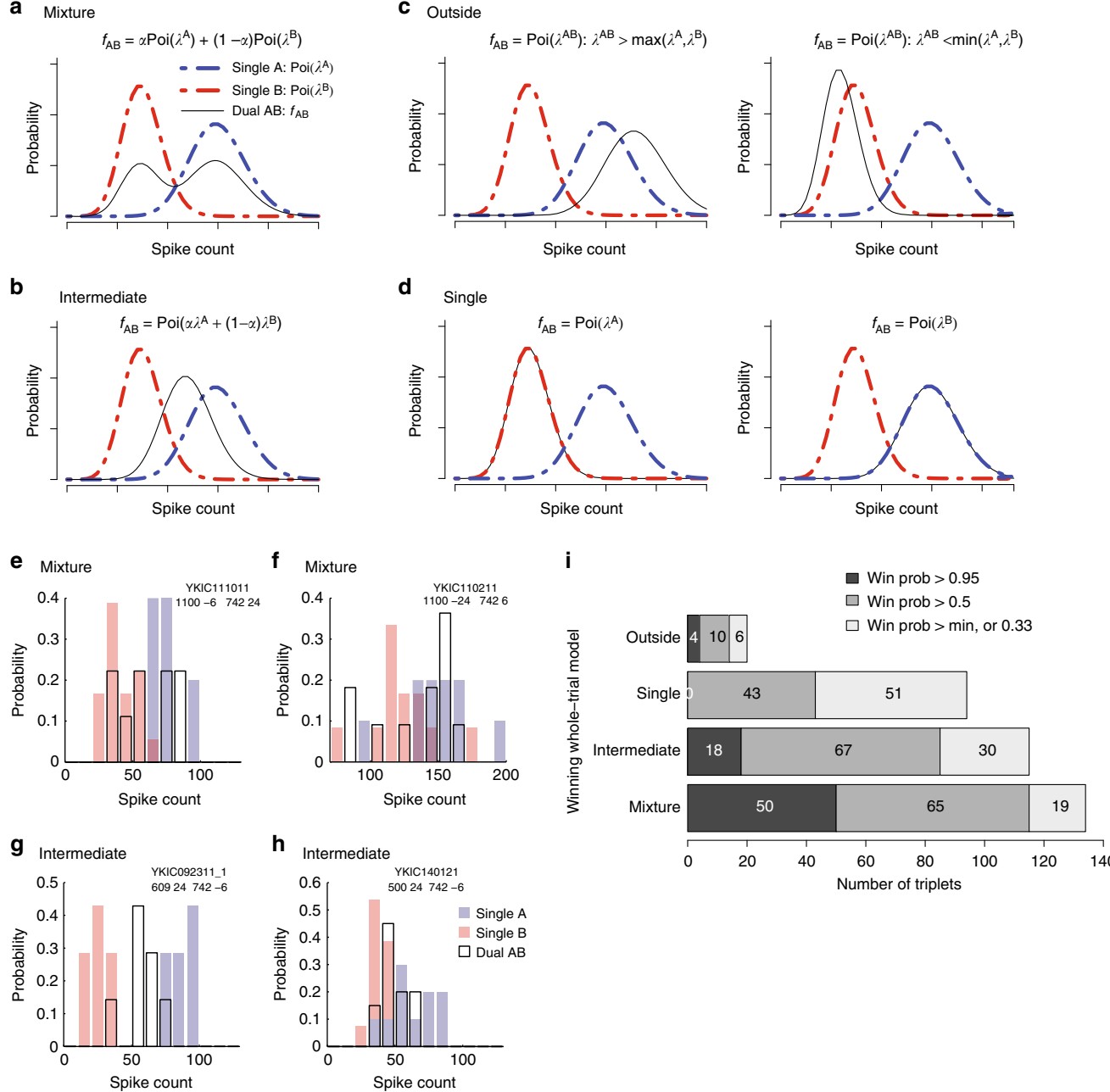

**Fig. 2** Whole-trial analysis. **a**–**d** show the four models that could describe the distribution of spike counts on individual dual-sound trials (0–600 or 0–1000 ms after sound onset, see Methods). **a** Mixture of the Poisson distributions of spike counts for the component single-sound trials, **b** Intermediate Poisson distribution, with rate between the rates of single-sounds responses, **c** Outside, Poisson distribution with rate larger or lower than the rates of single-sounds responses, **d** Single, Poisson distribution with rate equal to one of the two single-sound rates. **e**–**h** Four examples of spike count distributions for triplets classified as Mixtures or Intermediates. Red and blue shades indicate distributions of spike counts for single-sounds; black outlines indicate distributions for dual sounds. The triplets in **e**, **f** were classified as Mixture with winning probability > 0.95 (**e** shows the same triplet as Fig. 1e; **f** shows the same triplet as Fig. 3b). Triplets in **g**, **h** were classified as Intermediate with winning probability > 0.95 (**g** shows the same triplet as Fig. 1f and Fig. 3c; **h** shows the same triplet as Fig. 3d). **i** Population results of the whole-trial analysis. Shading indicates the confidence level of the assignment of individual triplets to winning models

trough on that trial. The triplet was then subjected to a two-way classification based on the distribution of these two features over the predicted curves (DAPP tags, Fig. 3a). The triplet was categorized as Wavy vs. Flat depending on whether the distribution of the maximum swing size peaked at high or low values, and as Central vs. Extreme according to whether the distribution of the time average $\alpha(t)$ had a peak close to 0.5 or had one to two peaks at the extreme values of 0 and 1. In addition to

this main classification scheme, triplets were subcategorized as exhibiting Symmetric or Skewed response patterns, indicating whether the $\alpha(t)$ curves reflected roughly equal contributions from the stimulus A and stimulus B response patterns or whether one or the other tended to dominate (Supplementary Figure 6).

The DAPP tags confirmed and extended the results of the whole-trial analysis. Figure 3e shows the different distributions of DAPP tags for the triplets categorized as Intermediate and

Mixture in the whole-trial analysis with winning probability > 0.95 (see also Supplementary Figures 6, 7). Mixture triplets tended to be classified as Flat-Extreme (70%), that is, the dynamics of these dual-sound responses were flat in time and matching either the responses to A ($\alpha \approx 1$) or B ($\alpha \approx 0$) on different trials. In addition, the distribution of the average $\alpha$

values tended to be either Symmetric or unlabeled with regard to symmetry, a sanity check that excludes winner-take-all responses (these would be characterized by Flat-Extreme-Skewed responses, Supplementary Table 1, Supplementary Figure 6).

In contrast, Intermediate triplets showed a combination of two types of labeling patterns relevant to our hypothesis. Some (22%)

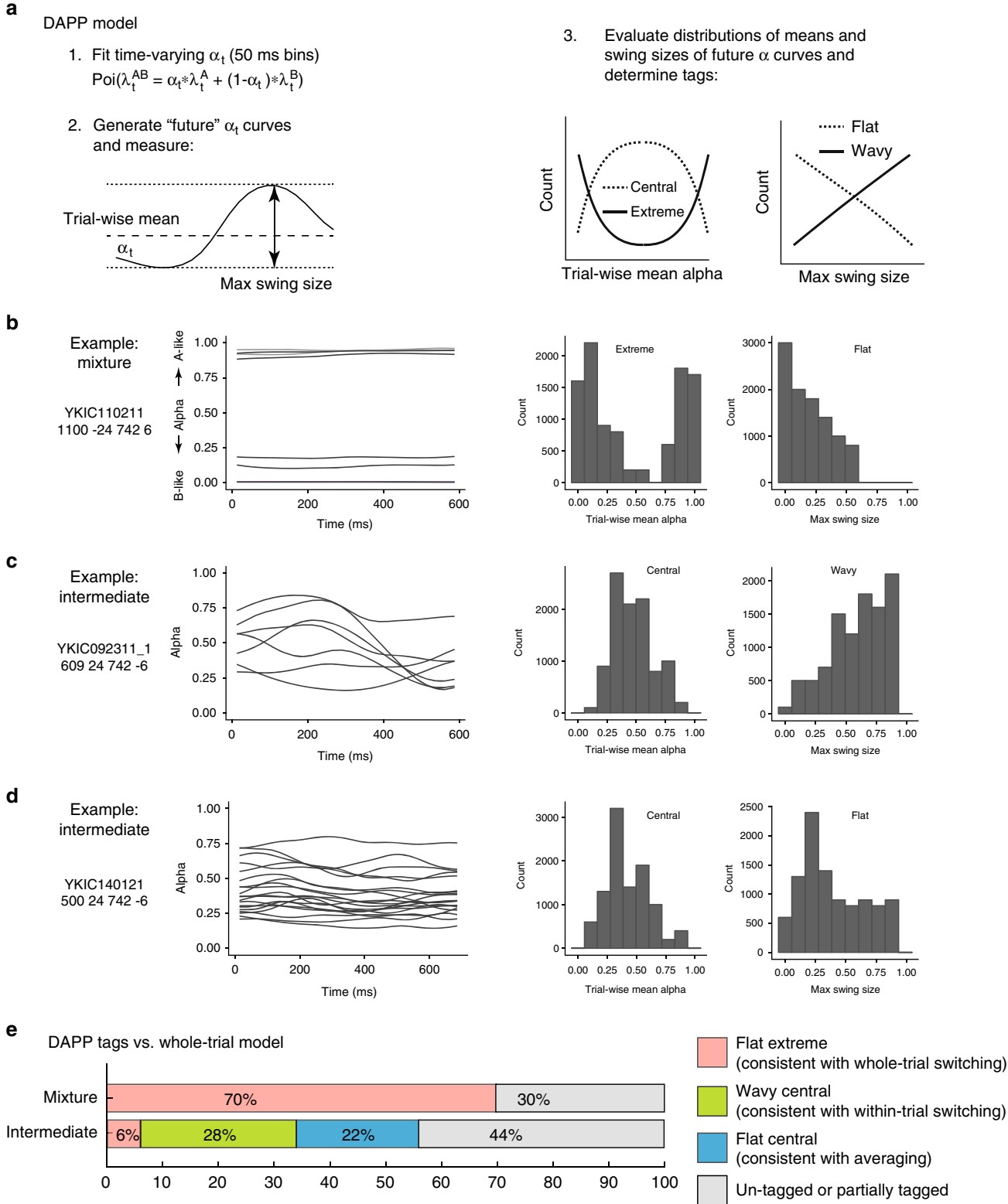

were classified as Flat-Central (and Symmetric), indicating stable $\alpha$ between 0 and 1, that is, stable firing at roughly the average of the responses evoked by each sound separately (see example cell in Fig. 3d). One triplet was classified as Flat-Extreme suggesting a stable $\alpha$ very close to either 0 or 1. Together these two firing patterns are consistent with some form of normalization occurring in this subpopulation. Other triplets (28%) were classified as Wavy-Central (and Symmetric) indicating responses that fluctuated symmetrically around a central value (see example cell in Fig. 3c).

In short, the DAPP analysis shows that the dynamics of the Mixture responses were consistent with fluctuations at the level of whole trials like the cell in Fig. 3b, whereas the dynamics of the Intermediate responses classified as Wavy-Central are suggestive of a neural code that could switch relatively rapidly between responses consistent with two single stimuli.

**Fluctuations appear coordinated and predict behavior.** We next considered the question of whether and how activity fluctuations are coordinated across the neural population, in two ways: (1) by evaluating activity correlations across time within trials between pairs of simultaneously recorded neurons, and (2) by evaluating whether the state of the local field potential prior to sound onset predicts between-trial fluctuations in activity e.g.,[30,31]. Finally, we determined whether the observed fluctuations are of functional relevance for the execution of the task by testing the relation between the trial-by-trial fluctuation and which target the monkey looked at first on that trial.

First, we evaluated correlations in within-trial switching based on how A-like vs. how B-like the responses were across time on individual trials. A total of 91 pairs of triplet conditions from 34 pairs of simultaneously recorded neurons (from among the 363 triplets used for the previous analyses) were assessed. For each 50 ms bin of a dual-sound trial in a given triplet, we assigned a probability score between 0 and 1 that the spike count in the bin was drawn from the Poisson distribution with rate equaling the bin's sound A rate. The complementary probability indicated the likelihood that the count was drawn from a Poisson distribution with a rate equaling the bin's sound B rate (Fig. 4a; see Methods: A vs. B assignment scores). We normalized these probabilities by converting them to Z-scores within a given time bin but across trials, to minimize the contribution of shared correlations due to stimulus responsiveness or changes in motivational state across time[32]. We then calculated the neuron-to-neuron correlation coefficients between the normalized assignment scores across time bins within each trial (i.e., one correlation coefficient value estimated per trial). This analysis is conceptually similar to conventional cross-correlation analysis of spike trains in neural pairs, but does not focus on precise timing of spikes or the relative latency between them[33,34]. The 50 ms time scale is consistent with

the frequency range in which spike-to-spike coherence has been observed in visual attention paradigms[35].

The observed correlations were generally positive, indicating that the activity was coordinated within the neural population. Figure 4 illustrates analysis of the dual-sound trials for a particular triplet in an example pair of neurons (Fig. 4a), and the distribution of the mean neuron-to-neuron correlations in the population for all the dual-sound conditions (Fig. 4b). The distribution of mean correlation coefficients was skewed positive (t-test, $p = 6.8 \times 10^{-6}$). Similar results were obtained when the raw spike counts were analyzed rather than the assignment scores (Supplementary Figure 8). This was the case even though we included triplets that were not categorized as showing wavy behavior in the DAPP analysis. It may be that coordinated activity fluctuations occur in more neurons than those that met our statistical criteria.

Next, we determined whether the state of the local field potential prior to sound onset predicts between-trial fluctuations in activity. We analyzed the LFP data recorded simultaneously with single unit spiking data. We combined data across triplets, creating two groups of trials based on whether the whole-trial spike count on a given dual-sound trial more closely resembled the responses evoked by sound A alone (where A is the contralateral sound) or sound B alone (see Methods: A vs. B assignment scores). Figure 5a shows the average LFP for the two groups of dual-sound trials. We quantified differences between these two groups with a t-test in the 600 ms windows before and after sound onset (each trial contributed one mean LFP value in each time window). As expected, the LFP signals statistically differed after sound onset in these two trial groupings (red vs. blue traces, time period 0–600 ms, $p = 1.0474 \times 10^{-5}$). But the average LFP voltage also differed prior to sound onset ($p = 0.0064$), suggesting that the state of activity in the local network surrounding an individual neuron at the time of sound onset is predictive of whether the neuron encodes the contralateral or the ipsilateral sound on that particular trial. What exactly that network state consists of is unknown; but it could be an altered balance in the levels of activity in the ICs contralateral and ipsilateral to the first-saccade target.

If fluctuations in neural activity are coordinated across the population, it follows that there should be a relationship between variability in neural activity and behavior. Accordingly, we investigated whether the activity on individual trials predicted whether the monkey would look first to sound A or sound B on that trial. We focused here on the whole-trial time scale as the most robust measure of activity fluctuations in our data. As noted in Methods, we trained the monkeys on sequential sounds first and this training strategy tended to promote performing the task in a stereotyped sequence. Partway through neural data collection, we provided monkey Y with additional training on the non-sequential task, after which that monkey began

**Fig. 3** Dynamic Admixture Point Process (DAPP) model: rationale and results. **a** The DAPP model fits smoothly time-varying weights ($\alpha$ and $(1-\alpha)$) capturing the relative contribution of A- and B-like response distributions to each AB dual-sound trial (point1). The dynamic tendencies of the $\alpha$ curves were then used to generate projected new $\alpha$ curves for hypothetical future draws from this distribution. The waviness and central tendencies were quantified by computing the max swing size and trial-wise mean for an individual trial drawn from the distribution (point 2). Low max swing sizes indicate flat curves and higher values indicate wavy ones (point 3, right panel). Similarly, the distribution of trial-wise means could be bimodal (Extreme) or unimodal (Central) (point 3, left panel). **b**–**d** Fit alphas for three example triplets (triplets in **b**–**d** are the same as in Fig. 2f, g, h, respectively) and the distributions of trial-wise means and max swing sizes for future draws from the alpha curve generator. **e** The pattern of DAPP results extended the whole-trial analysis results. Triplets categorized as Mixtures with a win probability > 0.95 tended to be tagged as Flat-Extreme (as example in **b**). Triplets categorized as Intermediates fell in two different main groups, Wavy-Central (as example in **c**) and Flat-Central (as example in **d**). Information about the Skewed vs. Symmetric tag is not shown. See Supplementary Table 1 and Supplementary Figures 6 and 7 for a complete listing of all the tag combinations and additional analyses

displaying less stereotypical behavior and sometimes saccaded first to A and sometimes first to B for a given AB dual-sound combination (see Fig. 5b for example). We then analyzed recording sessions after this training ($N = 73$ triplets) and we found that at both the whole trial and sub-trial time scales, the activity of individual neurons was predictive of what saccade sequence the monkey would choose on that particular trial. Specifically, the average dual-sound AB assignment score for a given triplet was computed separately for trials in which the first saccade was toward A vs. toward B. The average scores statistically differed between the two groups of dual-sound trials (t-test, $p = 5 \times 10^{-9}$, Fig. 5c) and in the expected direction, with more A-like scores occurring on trials in which the monkey looked at A first. This relationship was also present when looking at finer, 50 ms bin time scales (Fig. 5d).

**Multiplexing across time may be a general brain phenomenon.** The problem of encoding multiple simultaneous stimuli is neither limited to the auditory system nor to the localization of sounds, but ubiquitous in the brain and sensory information processing. To gain insight into whether activity fluctuations may contribute to preserving information about multiple stimuli in other systems, we turned to a different sensory system, vision, and a different neural substrate, cortex. Cells in an fMRI-identified face area, the middle fundus (MF) face patch[36] are highly face-selective and strongly tuned to head orientation. It is thus possible to find a stimulus eliciting a strong response in a given MF neuron, and another one that elicits a weak one (e.g., a non-face stimulus or a face at a non-preferred orientation). We found such stimuli for 105 MF neurons and recorded responses during single stimulus presentation or combinations of preferred and non-preferred stimuli within the receptive field (see Methods and[37]). Monkeys maintained fixation throughout stimulus presentation (400 ms).

Results of whole trial analysis are shown in Fig. 6a: the two most common response patterns at this time scale were Mixtures and Intermediates. The within-trial DAPP analysis (Fig. 6b) confirmed that most Mixtures could be classified as showing Flat-Extreme response patterns. For the Intermediates, in this data set

we found little evidence for Flat-Central; rather the largest category showed a Wavy-Central activity pattern, consistent with fluctuations at the sub trial time scale. In short, the evidence for item-preserving activity fluctuations was at least as strong in face patch MF as in the IC data set.

## Discussion

Our results show that the activity patterns of IC neurons fluctuate, and that these fluctuations may be consistent with encoding of multiple items in the same processing channels (i.e., the set of neural spike trains occurring in the IC). The time scale of these fluctuations ranges from the level of individual trials down to at least 50 ms bins within a trial. The fluctuations are positively correlated across pairs of neurons (at least, those recorded within the IC on a given side of the brain), are reflective of the state of local field potentials at the time of sound onset, and are predictive of the behavioral response to follow. These fluctuations may be a necessary element of the neural code, permitting the representation of more than one stimulus across time within a neural ensemble, and they may serve as a critical element of the coding of natural scenes replete with many simultaneous stimuli. The fact that similar activity patterns occurred in a completely different sensory system and brain region, face patch MF, supports this general interpretation.

The notion that aspects of signal fluctuations can be used to encode multiple items in a limited capacity channel has a history in telecommunications and engineering. Similar strategies have been postulated to occur in some form in the brain[1–11], but empirical evidence and statistical methods of assessment have been lacking.

A strength of our statistical approach is that we do not simply assess unimodality/bimodality (e.g.[10].) but anchor our consideration of the dual stimulus response distributions to the observed distributions for single stimulus trials. However, there are several limitations to the present statistical approach. First, the analyses could only be conducted on a subset of the data, requiring a good fit of a Poisson distribution to the single-sound trials and adequate separation of the responses on those trials. For the moment, it is unknown whether any of the excluded data

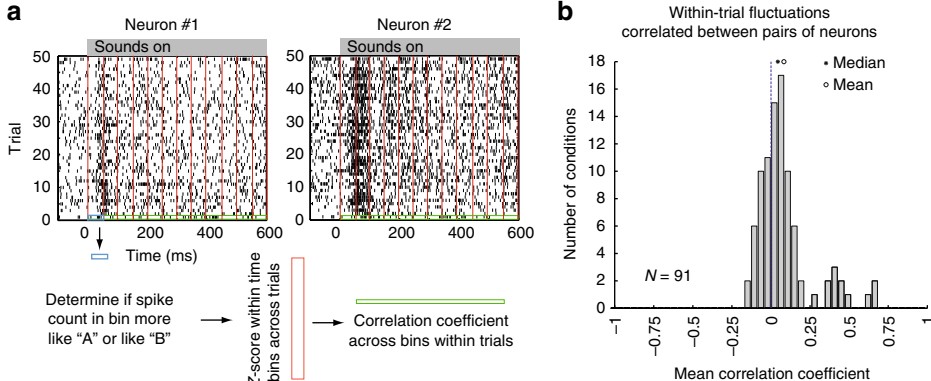

**Fig. 4** Pairs of neurons tend to show positive within-trial correlations. **a** Schematics of the analysis. Raster plots of two neurons recorded simultaneously; trials shown are for a particular set of dual-sound conditions. We evaluated the spike count in a given 50 ms time bin, trial, and member of the neuron pair for a given set of dual-sound conditions to determine if it was more similar to the spikes evoked during that bin on the corresponding sound A alone or B alone trials (blue box). We then converted these A vs. B assignment probabilities to a Z-score based on the mean and standard deviation of the assignment probabilities in that time bin on the other trials that involved the same stimulus conditions (red box). Finally, we computed the correlation coefficient between the set of Z score values for a given trial between the pair of simultaneously recorded neurons (green box). **b** Across the population of pairs of triplets recorded simultaneously, the distribution of mean correlation coefficients tended to be positive (t-test comparing the mean correlation coefficients to zero; $p = 6.8 \times 10^{-6}$). See Supplementary Figure 8 for the same analysis based on spike counts and broken down according to whether the neurons in the pair shared the same or had different preferences for sound A vs. sound B

exhibit meaningful response fluctuations. In principle, the modeling approach can be extended to other types of response distributions which should reduce the amount of data that is excluded. Second, the full range of time scales at which fluctuations occur is still undetermined. We focused on the whole-trial timescale and on 50 ms bins—biologically plausible for both the task and the stimuli. However, any faster fluctuations would likely have been (erroneously) categorized as Flat-Central in the DAPP model. Indeed, the preponderance of Flat-Extreme/Mixtures may not reflect the true state of the underlying population but rather the greater sensitivity of the analysis method for detecting slower fluctuations rather than faster ones. Third, our statistical approach based on the DAPP model involves a categorization step that summarizes the dominant features of a triplet. If a neuron sometimes behaves as a Flat-Extreme type and sometimes as a Wavy-Central type for a given triplet of conditions, it would likely be categorized as ambiguous. In other words, even though the DAPP model can pick up composite response patterns, the results we present ignore the existence of any such patterns.

The observed fluctuations have broad implications because they provide a novel account linking a number of other well-known aspects of brain function under a common explanation. First, it is widely recognized that neural firing patterns are highly variable. This variability is often thought to reflect some fundamental inability of neurons to code information accurately. Here,

we suggest that some of this variability may actually reflect interleaved periods of potentially quite accurate coding of different items. What else individual neurons may commonly be coding for in experiments involving presentation of only one stimulus at a time is not known, but possibilities include stimuli not deliberately presented by the experimenter, memories of previous stimuli, or mental imagery as suggested by the theory of embodied cognition[38]. Indeed, variability has been found to be higher when no deliberate stimulus is present at all[39]. In the present study, we were able to demonstrate signal in these fluctuations by virtue of statistical tests comparing each of the trial types in A-B-AB triplets, but it may be the case that fluctuations were occurring in the single stimulus trials as well. We could not test this because our analysis required having as benchmarks the response distributions corresponding to the potentially encoded items.

Second, as a concept, stimulus multiplexing via activity fluctuations across time provides insight into why limitations in certain types of cognition exist. Working memory capacity is limited; attention filters stimuli to allow in-depth processing of a selected set of items. These limitations may stem from using the same population of neurons for each attended or remembered item, a situation that may arise whenever neurons are broadly tuned. If this is the case, then the puzzle becomes why these limits are often greater than one. Stimulus multiplexing across time

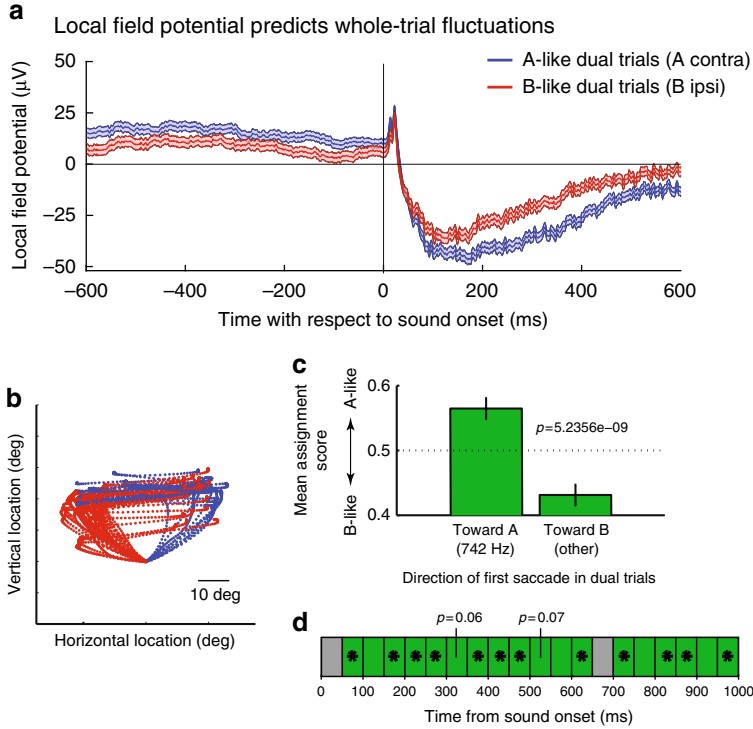

**Fig. 5** Fluctuations can be predicted from pre-stimulus LFP and in turn predict behavior. **a** To assess the relation between the LFP prior to sound onset and the spiking response after sound onset, we assigned each dual-sound trial to one of two groups, A-like or B-like, based on whether the spike count matched the response to single-sound A or B (see Methods). We then compared the average LFP voltage (without filtering for any particular frequency band) of the two groups. The average LFP (mean ± SE) is shown in blue for 1917 trials classified as A-like (A = contralateral sound) and in red for 1565 trials classified as B-like. The traces are significantly different in the 600 ms before sound onset and in the 600 ms after sound onset (two-tailed *t*-test, *p* < 0.01). **b–d** The target of the first saccade on dual-sound trials can be predicted by the spike count during sound presentation. **b** Eye trajectories during dual-sound trials to the same pair of single-sounds (one triplet). The traces are color-coded based on which of the two sounds the monkey looked at first in the response sequence. For clarity, all traces are aligned on a common starting position despite some variation in fixation accuracy. **c** The average assignment score of trials in which the monkey looked at sound A first is more A-like than that of trials in which the monkey looked at sound B first. Bars indicate SEM; *p* value is for a two-tailed *t*-test involving a total of *n* = 1171 trials. **d** The relationship between assignment score and first saccade target was also evident at the scale of 50 ms bins (green = positive correlation; *\*p* < 0.05 for *t*-test of assignment score on A-first vs. B-first trials)

suggests that cycling between different items allows evading what might otherwise be a one-item limit[2]. Here, we investigated only two time scales, 50 ms and whole trials. Additional work is needed to more fully explore the time scales on which this occurs and to tie the resulting information on duty cycle to perceptual capacity.

Third, brain oscillations are ubiquitous, have been linked specifically to attentional and memory processes[31], e.g. [40], see also[41], and have been suggested as having a connection to the coding of multiple stimuli[2–8,42,43]. Field potential oscillations indicate that neural activity fluctuates, although they capture only the portion of such fluctuation that is coordinated across the underlying neural population and is regular in time. It remains to be determined to what degree field potential oscillations reflect cause vs effect. In other words, field potential oscillations could stem from the activity of neural circuits involved in controlling multiplexing, or they could reflect the activity of the neural circuits subject to the effects of such control. In a highly interconnected system such as the brain, both are likely to occur.

Our findings are particularly relevant to the domain of attention at both the behavioral and neural mechanism levels. Multiplexing implies that multiple stimuli are handled by a common circuit in a serial fashion. Serial processing of multiple stimuli has been suggested using several different behavioral paradigms such as searching for or tracking targets among distractors e.g., refs. [44,45]. There is also evidence for sampling of the visual scene across time: the detectability of visual stimuli can fluctuate in an oscillatory fashion[46], and this is correlated with the phase of EEG signals in the 4–12 Hz range[47,48]. A similar relationship between auditory perception and EEG phase has also recently been demonstrated[49]. On the whole, this line of investigation suggests that the brain samples stimuli in a periodic or sequential manner.

The scope of such sampling processes may be more extensive than is possible to demonstrate with behavioral assays alone. Our neural analyses do not require that activity fluctuations be either coordinated or periodic at the level of individual neurons. Instead, neural ensembles may fluctuate asynchronously with other ensembles and the fluctuations may not always be periodic. Perceptual studies such as those described above can likely only reveal effects when the underlying neural activity is highly coordinated across the neural population. This would account for why the perceptual impact of a sampling mechanism of the brain is not evident in daily life, but requires experiments designed to elicit or align with maximum synchrony across component neural ensembles to reveal it.

Single-neuron recording studies concerning the impact of attention on (chiefly visual) representations have debated two main possibilities: that attention biases the competition between stimuli for processing in neural populations vs. that it operates as either a filter or spotlight that adjusts the gain of neural responses to unattended vs. attended stimuli (for review, see ref. [50]). The biased competition view bears the closer relationship to multiplexing. Stated strongly, under this theory neurons should respond as if only the attended stimulus is present, ignoring distractors, and the response rate should be the same as if the attended stimulus was the only one presented. If attention were to shift between items across time, as it might have in our study, neural activity would be expected to fluctuate accordingly.

Most research to assess this theory has not investigated fluctuations but has instead pooled the activity across time and trials (see ref. [10] for an exception). The results have generally produced responses to dual stimuli that are intermediate between the responses evoked by each stimulus alone but with a bias in favor of the attended stimulus (e.g., refs. [51,52]). It will be of interest to ascertain whether such intermediate responses and incomplete bias in favor of the attended stimulus reflects underlying activity fluctuations that serve to preserve information about both stimuli while enhancing the salience of the attended one. Such a pattern would help account for the otherwise curious observation that intermediate responses also apparently occur when neither stimulus is attended (e.g., ref. [52]). In short, stimulus multiplexing across time may operate both in concert with attention and independently of it. We note that we did not attempt to assess whether attention was shifting between the two stimuli that were presented, so future work will be needed to tease apart these possibilities.

The need for multiplexing extends to any domain where neural tuning is broad. The meter/firing rate code for sound location in the IC is one example. The IC's sensitivity to sound frequency can only partially ameliorate this problem, because here too the tuning is very broad: a pure tone in the frequency range employed in this study has been shown to evoke activity in 40–80% of IC neurons[53]. Indeed, about 2/3 of the triplets in the present sample responded to both of the two sounds individually (Supplementary Figure 5). Most natural sounds are spectrally rich and will activate hills of neural activity with even greater overlap. That said, it is possible that the resolution of the frequency map might sharpen via lateral inhibition when more than one sound is present; such a mechanism might work in concert with stimulus multiplexing across time.

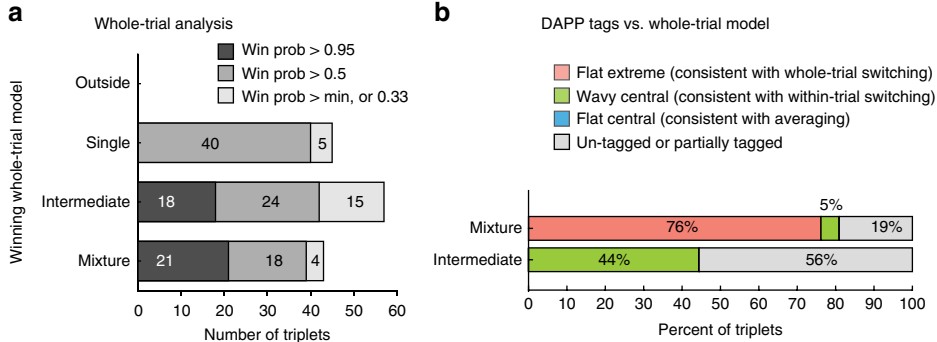

**Fig. 6** Evidence of fluctuations in face patch MF. **a** Population results of whole-trial analysis. As in Fig. 2i, shading indicates the confidence level of the assignment of individual triplets to winning models. **b** Population results of DAPP analysis by winning model from whole-trial analysis (win probability > 0.95). Results resemble those found in the IC, except that there was no evidence of Flat-Central among the Intermediates; Wavy-Central was the predominant label for this group among triplets that could be categorized

In the case of our particular experimental paradigm, several additional questions arise. How do signals related to different items come to be multiplexed? Are they later de-multiplexed? If so, how?

To some degree, sounds are multiplexed in the world. That is, the sound waves from multiple sources sum in the world and are never purely distinct from one another. The air pressure waves arriving at each ear reflect the combined contribution of all sound sources. Where and how signals may be de-multiplexed critically depend on the nature of the representation to which a de-multiplexed output could be written. In barn owls, which have maps of auditory space, the coding bottleneck intrinsic to meter/rate coding does not occur, and two sounds produce two separate active populations[54–57]. Such distinct peaks suggest that the multiplexed-in-the-air signals have been de-multiplexed and segregated into two hills of activity.

In primates and several other mammals, neural representations of space employ meters (rate codes) rather than maps throughout the pathway from sound input to eye movement output, as far as is currently known[18–21,58–62]. This is the case even at the level of the superior colliculus[63], an oculomotor structure which has a well-deserved reputation for mapping when activity is evoked by non-auditory stimuli[64,65].

Given that different types of codes exist in different species, and given that coding format is not known in all the circumstances in which multiplexing might apply (e.g., attention, working memory), we developed two different models to illustrate a range of different de-multiplexing possibilities (Fig. 7) based on the nature of the recipient representation. In the first (Fig. 7a), a multiplexed signal in a meter is converted into two hills of activity

in a map, using a basic architecture involving graded thresholds and inhibitory interneurons suggested previously[66]. Adding an integration mechanism such as local positive feedback loops would then serve to latch activity on at the appropriate locations in the map, producing a more sustained firing pattern. No clock signal is necessary for this model.

In the second model (Fig. 7b), there are multiple output channels, each capable of encoding one item. An oscillating circuit that knows about the timing of the input gates signals to each output channel at the appropriate moments. As in the first model, a local positive feedback mechanism acts to sustain the activity during the gaps in the input. This model thus retains the efficient coding format of a meter but requires a controlling signal with knowledge of when to latch input flow through to each output channel. In our data, fluctuations were at least somewhat coordinated across pairs of simultaneously recorded neurons, in agreement with this model. It is possible that within-trial fluctuating units lie at the input stage of such a circuit, and that between-trial fluctuating units actually lie at the output stage. A given unit might be allocated to either the A or the B pools based on state of the network (as detected by the LFP measurements) on different trials.

Although the stimulus multiplexing across time that we observed here has parallels in engineering (particularly with time division multiplexing in telecommunications) these two models also help illustrate that multiplexing is unlikely to occur exactly the same way in biological systems as it does in technological ones. For example, the time course may be more fluid in biological systems, and different neural ensembles may operate asynchronously and/or aperiodically as opposed to operating under a

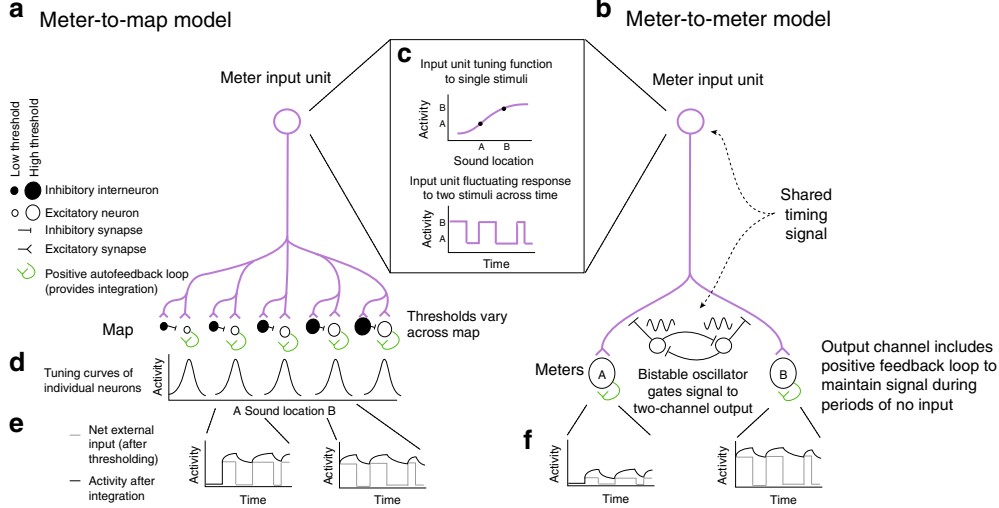

**Fig. 7** Two possible mechanisms for de-multiplexing a fluctuating signal. A clock signal that knows about coding transitions is not necessarily needed if signals are read out into a map (**a**), but is required if signals are retained in a meter or rate-coded format (**b**). Both models have an input signal that employs a meter code for sound location (purple, **c** top panel), and this signal is assumed to fluctuate between two response levels when two sounds are present (**c** bottom panel). In the meter-to-map model (**a**), the second stage consists of group of excitatory neurons (open circles) with varying thresholds, paired with inhibitory neurons (filled circles) that have slightly higher thresholds. These neurons all receive input from the input meter signal. An individual excitatory neuron is activated when the input signal exceeds its own threshold and is lower than the threshold of the paired inhibitory interneuron, producing tuning curves like those shown in **d**. The net drive across time to two examples A and B is shown schematically as the thin gray line in the two inset graphs (**e**). These two neurons would turn on and off out of phase with each other based on their external inputs alone. Adding a positive autofeedback loop to each excitatory neuron in the map (green) provides integration of the activity of each neuron and permits the activity to be sustained across periods of time when there is no external drive (dark line). This model is derived from a portion of the Vector Subtraction model of[66]. In the meter-to-meter model (**b**), the input is forked to an A meter channel neuron and a B meter channel neuron. A bistable oscillator coupled to the same unknown clock (shared timing signal) that controls the input fluctuations would be needed to appropriately route the output to these two units. The resulting output, in the absence of positive autofeedback, would also fluctuate but between an off state and a level that corresponds to signaling the presence of sound A or B respectively (**f** gray lines). Adding positive autofeedback would again allow bridging across the off states (dark lines)

regular, shared, clock cycle of some kind. The strongest shared element of the analogy with telecommunications is the transmission of more than one item or type of information via signal fluctuations at some time scale.

Similarity of results in the IC with auditory stimuli and MF with visual stimuli suggest that multiplexing may be a general mechanism that is commonly at play to enhance the total processing power of the brain. The statistical tools developed here can be applied to any triplet data. Additional studies with both single stimulus conditions, to define the distributions of signals, and dual stimulus conditions, to evaluate fluctuations between membership in those distributions, will be important for delineating the extent of this phenomenon. Digging under the hood of the time-and-trial pooled activity to look at activity patterns on a moment by moment basis will be essential to advancing our understanding of how the brain operates dynamically to maximize its processing power.

## Methods

**General procedures**. All procedures conformed to the guidelines of the National Institutes of Health (NIH Pub. No. 86–23, Revised 1985) and were approved by the Institutional Animal Care and Use Committee of Duke University. Two adult rhesus monkeys (*Macaca mulatta*) participated (monkey P, and monkey Y, both female). Under general anesthesia and in sterile surgery we first implanted a head post holder to restrain the head and a scleral search coil to track eye movements[67,68]. After recovery with suitable analgesics and veterinary care, we trained the monkeys in the experimental task. In a second surgery, we implanted a recording cylinder (2 cm diameter) over the right (monkey Y) or left (monkey Y, P) IC respectively. We determined the location of the cylinder with stereotactic coordinates and verified it with MRI scans e.g. [53].

**Sound localization task**. The monkeys performed a single-sound or dual-sound localization task (Fig. 1b) by making saccades toward one or two simultaneously-presented auditory targets with one or two saccades as appropriate. All sound targets were located in front of the monkey at eye level; the horizontal location, frequency and intensity were varied pseudorandomly as described below (Recording Procedures). Each trial began with 600–700 ms of fixation of a visual stimulus (light emitting-diode, LED, located straight ahead and 10–14° below the speakers). During fixation we presented one sound (single-sound trials) or two simultaneous sounds (dual-sound trials). After a fixation time of either 600–800 ms (Data Set I, some of Data Set II) or 1000–1100 ms (remainder of Data Set II), the fixation light was extinguished and the monkey was required to make a single saccade on single-sound trials or a sequence of two saccades (in either order) on dual-sound trials. Trials were considered correct if each saccade was directed within 10–17.5 degrees horizontally and 20–40 degrees vertically of a target due to vertical inaccuracies in localizing non-visual targets in primates[69], and if the gaze was maintained on the final target for 100–200 ms. On correct trials monkeys were rewarded with juice drops.

**Training**. Training was accomplished in three stages. We initially trained the monkeys to report the location of single visual targets by saccading to them. We then introduced single auditory targets. As these were novel and unexpected in the silent experimental booth, monkeys readily saccaded to them[70]. To help the monkeys calibrate their auditory saccades, a visual feedback was added on trials where the auditory saccade was not initiated correctly within 700 ms. The feedback was presented only at the most peripheral target locations (+/−24 degree) and only for a few initial days of training. Finally, we trained monkey to localize dual-sound targets. Initially we presented the two sounds sequentially in a specific order, then we gradually reduced the temporal gap between them until the sounds were simultaneous.

In the final version of the task, monkeys were allowed to look at the targets in either order, as noted above. However, due to the initial training with sequential sounds, they retained stereotyped patterns of saccades in which they tended to look first to whichever sound location had been presented first during the sequential and partial overlap stages of training. Monkey P was trained with more central target locations (e.g., −6 or 6 degree targets) initially occurring first and more peripheral targets (e.g., −24 or 24 degree targets) occurring second, and monkey Y was trained with sounds initially occurring in the opposite sequence. Midway through neural data collection, we provided additional training to monkey Y to encourage free choice of which sound to look at first. This allowed us to investigate the relationship between each behavioral response and the neural representation at that moment.

**Recording procedure and strategy**. The behavioral paradigm and the recordings of eye gaze and single cell activity were controlled using the Beethoven program

(Ryklin Software). Recordings were made with one or two tungsten electrodes (FHC, impedance between 1 and 3 MΩ at 1 kHz). Each electrode was lodged in a stainless-steel guide tube (manually advanced through the dura) and controlled independently with an oil hydraulic pulse micropositioner (Narishige International USA, Inc. and NAN INSTRUMENTS LTD, Israel). First, we localized the IC (and isolated single neurons) while the monkey listened passively to sounds of different frequencies. We then collected single unit spiking activity and local field potential while the monkey performed the single-sound and dual-sound localization tasks. We used a Multichannel Acquisition Processor (MAP system, Plexon Inc., Dallas, TX) and Sort Client software. The single unit spiking activity was filtered between 150 kHz and 8 kHz and sampled at 20 kHz, while the LFP signal was filtered between 0.7 and 300 Hz and sampled at either 20 or 1 kHz (see 'Local field potential'). Data were collected as long as the neurons were well isolated and the monkey performed the tasks.

Neural signals were recorded primarily from two functionally-defined subregions of the IC, the low frequency area and the tonotopic area[53]. Neurons in the low frequency tuned area generally respond best to low frequencies and there is little heterogeneity in tuning, whereas neurons recorded in the tonotopic area had best frequencies that could be either low or high depending on the position of the recording electrode.

**Data sets and sound stimuli**. The spiking activity of 166 single neurons was recorded in two datasets involving the same task but differing in which sound levels and frequencies were included. A total of 68 of these neurons were recorded as pairs from separate electrodes positioned in the IC on the same side of the brain at a minimum spatial separation of 2 mm. Local field potentials (LFP) were also recorded from 87 of these recording sites.

In both datasets, the sounds consisted of bandpass noise with a bandwidth of +/−200 Hz. On dual-sound trials, the sounds were delivered from pairs of locations (24 degrees and −6 degrees), and (−24 and +6 degrees) i.e., 30 degrees apart. The two sounds differed in frequency, with one of the two sounds having a 742 Hz center frequency and the other differing by at least 0.285 octaves or multiples of this distance. Single-sound trials involved the same set of locations and frequencies as on dual-sound trials, but with only a single-sound presented at a time. All sounds were frozen within an individual session; that is, all trials with a given set of auditory parameters involved the same time series signal delivered to the relevant speaker.

In data set I ($N = 98$ neurons), the sounds presented on dual-sound trials were 742 Hz and a sound from the set (500, 609, 903, 1100 Hz); these frequencies were ±0.285 octave or ±0.57 octaves above or below 742 Hz, or ±3.4 and 6.8 semitones. Combining two sounds will produce a combination that is louder than either component. Sound levels were therefore calibrated to provide two sets of conditions: dual sounds for which the component sounds involve the same signals to the audio speakers as on single-sound trials, producing a louder dual sound, and dual sounds for which the level of the component sounds was reduced so that the overall loudness was the same on dual as on single trials. The levels used for the components were 51 and 55 dB, producing sound levels of minimum 55 or maximum 60 dB on dual-sound trials. The same-signal comparison involved using the 55 dB component levels, singly and on dual-sound trials. The same-loudness comparison involved using the 55 dB levels on single-sound trials and the 51 dB levels for the components of dual-sound trials. Calibrations were performed using a microphone (Bruel and Kjaer 2237 sound level meter) placed at the position normally occupied by the animal's head.

Because results did not differ substantively when comparisons were made between same-signal and same-loudness conditions (Supplementary Figure 2 vs. Supplementary Figure 3), we pooled across sound levels for subsequent analyses, and we dispensed with the multiple sound levels for data set II (monkey Y only, $N = 68$ neurons), using either 50 or 55 dB levels for all components. We also incorporated additional sound frequencies, [1340 1632 1988 Hz], to improve the odds that responses to each of the component sounds differed significantly. Again, one of the two sounds on dual-sound trials was 742 Hz; the other sound frequency was either from the original list of [500 609 903 1100] or from the new frequencies. Most of the neurons in this data set were tested with [500 742 1632].

**Cell inclusion and trial counts**. The $N = 166$ neurons ($N = 98$ from Data Set I and $N = 68$ from Data Set II) included for analysis were drawn from a larger set of 325 neurons. Neurons were excluded from analysis if the neuron proved unresponsive to sound (Student's *t*-test, spike counts during the 600 ms after sound onset compared to the same period immediately prior to sound onset, one-tailed, $p > 0.05$), or if there were too few correct trials (minimum of five correct trials for each of the components [A, B, and AB trials] that formed a given triplet of conditions or if there were technical problems during data collection (e.g., problems with random interleaving of conditions or with computer crashes). The average number of correct trials for a given set of stimulus conditions in the included dataset ($N = 166$) was 10.5 trials. The total number of included triplets was 1484. All analyses concerned correctly performed trials.

**Summation vs. averaging in time-and-trial pooled activity**. To evaluate IC activity using conventional analysis methods that pool across time and/or across

trials, we counted action potentials during two standard time periods. The baseline period (Base) was the 600 ms period before target onset, and the sensory-related target period (Resp) was the 600 ms period after target onset (i.e., ending before, or at the time of, the offset of the fixation light, Fig. 1b).

Summation/averaging indices: We quantified the activity on dual-sound trials in comparison to the sum and the average of the activity on single-sound trials, expressed in units of standard deviation (Z-scores), similar to a method used by[28]. Specifically, we calculated,

$$\text{PredictedSum}_{A,B,} = \text{mean}(\text{Resp}_A) + \text{mean}(\text{Resp}_B) - \text{mean}\left(\text{Base}_{A,B}\right) \tag{1}$$

and

$$\text{PredictedAvg}_{A,B,} = (\text{mean}(\text{Resp}_A) + \text{mean}(\text{Resp}_B))/2 \tag{2}$$

where $\text{Resp}_A$ and $\text{Resp}_B$ were the number of spikes of a given neuron for a given set of single-sound conditions A and B (location, frequency, and intensity) that matched the component sounds of the dual-sound trials being evaluated. As the response may actually include a contribution from spontaneous baseline activity, we subtracted the mean of the baseline activity for the single-sounds ($\text{Base}_{A,B}$). Without this subtraction, the predicted sum would be artificially high because two copies of baseline activity are included under the guise of the response activity.

The Z scores for the dual-sound trials were computed by subtracting these predicted values from the mean of the dual-sound trials (mean($\text{Resp}_{AB}$)) and dividing by the mean of the standard deviations of the responses on single-sound trials:

$$\text{Zsum}_{AB} = \frac{\text{mean}(\text{Resp}_{AB}) - \text{PredictedSum}_{A,B}}{\text{mean}(\text{std}(\text{Resp}_A), \text{std}(\text{Resp}_B))} \tag{3}$$

and

$$\text{ZAvg}_{AB} = \frac{\text{mean}(\text{Resp}_{AB}) - \text{PredictedAvg}_{A,B}}{\text{mean}(\text{std}(\text{Resp}_A), \text{std}(\text{Resp}_B))} \tag{4}$$

If the dual response was within +/−1.96 standard deviations of the predicted sum or predicted average, we could say the actual dual response was within the 95% confidence intervals for addition or averaging of two single responses, respectively.

**Analysis of whole-trial fluctuations and inclusion criteria.** Our statistical tests for fluctuations in neural firing were conducted on triplets, or related sets of single and dual-sound trials (A, B, AB trials). To evaluate whether neural activity fluctuates across trials in a fashion consistent with switching between firing patterns representing the component sounds, we evaluated the Poisson characteristics of the spike trains on matching dual and single-sound trials (triplets: AB, A and B). Spike train data from each trial was summarized by the total spike count between 0–600 ms or 0–1000 ms from sound onset (i.e., whatever the minimum duration of the overlap between fixation and sound presentation was for that recorded neuron, see section Sound localization task). We modeled the distribution of spike counts in response to single-sounds A and B as Poisson distributions with unknown rates $\lambda^A$, denoted $\text{Poi}(\lambda^A)$, and $\lambda^B$, denoted $\text{Poi}(\lambda^B)$. Four hypotheses were considered for the distribution of sound AB spike counts:

1.  a mixture distribution $\alpha \cdot \text{Poi}\left(\lambda^A\right) + (1 - \alpha) \cdot \text{Poi}\left(\lambda^B\right)$ with an unknown mixing weight $\alpha$ (Mixture)
2.  a single $\text{Poi}(\lambda^{AB})$ with some $\lambda^{AB}$ in between $\lambda^A$ and $\lambda^B$ (Intermediate)
3.  a single $\text{Poi}(\lambda^{AB})$ where $\lambda^{AB}$ is either larger or smaller than both $\lambda^A$ and $\lambda^B$ (Outside)
4.  a single $\text{Poi}(\lambda^{AB})$ where $\lambda^{AB}$ exactly equals one of $\lambda^A$ and $\lambda^B$ (Single)

Relative plausibility of these competing hypotheses was assessed by computing their posterior probabilities with equal prior weights (1/4) assigned to the models, and with default Jeffreys' prior[71] on model specific Poisson rate parameters, and a uniform prior on the mixing weight parameter $\alpha$. The Jeffreys' prior was truncated to appropriate ranges for the Intermediate and Outside models. Posterior model probabilities were calculated by computation of relevant intrinsic Bayes factors[29].

Triplets were excluded if either of the following applied: (1) the Poisson assumption on A and B trial counts was not supported by data; or (2) $\lambda^A$ and $\lambda^B$ were not well separated. To test the Poisson assumption on single-sound trials A and B of a given triplet, we used an approximate chi-square goodness of fit test with Monte Carlo p-value calculation. For each sound type, we estimated the Poisson rate by averaging counts across trials. Equal probability bins were constructed from the quantiles of this estimated Poisson distribution, with number of bins determined by expected count of five trials in each bin or at least three bins— whichever resulted in more bins. A lack-of-fit statistic was calculated by summing across all bins the ratio of the square of the difference between observed and expected bin counts to the expected bin count. Ten thousand Monte Carlo samples

of Poisson counts, with sample size given by the observed number of trials, were generated from the estimated Poisson distribution and the lack-of-fit statistic was calculated from each one of these samples. p-value was calculated as the proportion of these Monte Carlo samples with lack-of-fit statistic larger than the statistic value from the observed data. Poisson assumption was considered invalid if the resulting Monte Carlo p-value < 0.1.

For triplets with valid Poisson assumption on sound A and B spike counts, we tested for substantial separation between $\lambda^A$ and $\lambda^B$, by calculating the intrinsic Bayes factor of the model $\lambda^A \neq \lambda^B$ against $\lambda^A = \lambda^B$ with the non-informative Jeffreys' prior on the $\lambda$ parameters: $\lambda^A$, $\lambda^B$ or their common value. The triplet was considered well separated in its single sounds if the logarithm of the intrinsic Bayes factor equaled three or more, which is the same as saying the posterior probability of $\lambda^B \neq \lambda^A$ exceeded 95% when a priori the two models were given 50–50 chance.

It should be noted that the sensitivity/specificity of detection were not equal across the four competing hypotheses. Because the Single hypothesis is a limiting case of each of the other three hypotheses, the method's sensitivity to this response pattern is lower than for the other three possible outcomes. This was verified on simulated data for which the truth could be known; simulated Mixtures, Intermediates, and Outsides were commonly correctly categorized with >95% confidence, whereas Singles were correctly categorized but at a lower level of confidence.

**Dynamic Admixture Point Process Model.** To evaluate whether neural activity fluctuates *within* trials, we developed a novel analysis method we call a Dynamic Admixture Point Process model (DAPP) which characterized the dynamics of spike trains on dual-sound trials as an admixture of those occurring on single-sound trials. The analysis was carried out by binning time into moderately small time intervals. Given a predetermined bin-width $w = T/C$ for some integer C, we divided the response period into contiguous time intervals $I_1 = [0;w)$; $I_2 = [w; 2w)$ ... $I_C = [(C-1)w,Cw)$ and reduced each trial to a C-dimensional vector of bin counts $(X^e_{j1},...,X^e_{jC})$ for $e \in \{A;B;AB\}$ and $j = 1,..., n_e$. Mathematically, $X^e_{jC} = N^e_j(Ic)$. The results reported here were based on $w = 50$ (with time measured in ms and $T = 600$ or 1000), but we also repeated the analyses with $w = 25$ and noticed little difference.

Our model for the bin counts was the following. Below we denote by $t^*_c$ the mid-point $(c-1/2)w$ of sub-interval $I_c$.

1.  $X^e_{jc} \sim \text{Poi}\left(w \cdot \lambda^e\left(t^*_c\right)\right)$, $e \in \{A,B\}$, $c \in \{1,...,C\}$, $j \in \{1,...,n_e\}$. We assume both $\lambda^A(t)$ and $\lambda^B(t)$ are smooth functions over $t \in [0, T]$.

2.  $X^{AB}_{jc} \sim \text{Poi}\left(w \cdot \lambda_j\left(t^*_c\right)\right)$, where $\lambda_j(t) = \alpha_j(t) + \{1 - \alpha_j(t)\}\lambda^B(t)$ with $\alpha_j:[O, T] \rightarrow (0,1)$ being unknown smooth functions.

We modeled $\alpha_j(t) = S(\eta_j(t))$, where $S(t) = 1/(1 + e^{-t})$, and, each $\eta_j(t)$ was taken to be a (smooth) Gaussian process with $E\{\eta_j(t)\} \equiv \phi_j$, $\text{Var}\left\{\eta_j(t)\right\} \equiv \psi_j$, and, $\text{Cor}\{\eta_j(t), \eta_j(t^{'})\} = \exp\{-0.5(t - t^{'})^2/\ell_j^2\}$. The three parameters $(\phi_j, \psi_j, \ell_j)$ respectively encoded the long-term average value, the total swing magnitude and the waviness of the $\alpha_j(t)$ curve.—Intuitively, these parameters can be thought of as related to the means and variances of the distribution of weight values $\alpha$ regardless of time within a trial, as well as the correlation between the weight observed at one point in time and the weight observed at another on any given trial. While the temporal imprint carried by each $\alpha_j$ was allowed to be distinct, we enforced the dual trials to share dynamic patterns by assuming $\left(\phi_j, \psi_j, \ell_j\right), j = 1, \dots, n_{AB}$, were drawn from a common, unknown probability distribution $P$, which we called a dynamic pattern generator and viewed as a characteristic of the triplet to be estimated from the data.

To facilitate estimation of $P$, we assumed it decomposed as $P = P_{\phi\psi} \times P_\ell$, where $P_{\phi\psi}$ was an unknown distribution on $(-\infty,\infty) \times (0,\infty)$ generating $(\phi_j,\psi_j)$, and, $P_\ell$ was an unknown distribution on $(0,\infty)$ generating $\ell_j$. To simplify computation, we restricted $\ell_j$ to take only a finitely many positive values, representative of the waviness range we are interested in (in our analyses, we took these representative values to be {75, 125, 200, 300, 500}, all in ms). This restricted $P_\ell$ to be a finite dimensional probability vector.

We performed an approximate Bayesian estimation of model parameters. Note that only $\lambda^A(t)$ and $\lambda^B(t)$ were informed by the single-sound trial data. All other model parameters were informed only by the dual-sound trial data conditionally on the knowledge of $\lambda^A(t)$ and $\lambda^B(t)$. To take advantage of this partial factorization of information sources, we first smoothed each set of single-sound trial data to construct a conditional gamma prior for the corresponding $\lambda^e\left(t^*_c\right), e \in \{A, B\}, c = 1, \dots, C$, where the gamma distribution's mean and standard deviation were matched with the estimate and standard error of $\lambda^e\left(t^*_c\right)$. A formal Bayesian estimation was then carried out on all model parameters jointly by (a) using only the dual-sound trial data, (b) utilizing the conditional gamma priors on $\lambda^A(t)$ and $\lambda^B(t)$, and, (c) assuming a Dirichlet process prior[72] on $P_{\phi\psi}$ and an ordinary Dirichlet prior on $P_\ell$. This final step involved a Markov chain Monte Carlo computation whose details will be reported in a separate paper.

Next, the estimate of the generator $P$ was utilized to repeatedly simulate $\alpha(t)$ functions for hypothetical, new dual trials for the triplet. For each simulated $\alpha(t)$ curve, we computed its maximum swing size $|\alpha| = \max_t \alpha(t) - \min_t \alpha(t)$, and, time aggregated average value $\bar{\alpha} = \int_0^T \alpha(t)\,dt/T$. The waviness index of the triplet was computed as the odds of seeing an $\alpha(t)$ function exhibiting a swing of at least 50%

between its peak and trough:

$$r_w = \frac{P(|\alpha| > 0.5)}{P(|\alpha| < 0.5)}$$

where $P$ denotes the sampling proportion of the simulated $\alpha$ draws. The triplet's extremeness index was computed as the odds of seeing an $\alpha(t)$ function with its long-term average $\bar{\alpha}$ being closer to the mid-way mark of 50% than the extremes:

$$r_c = \frac{P(\bar{\alpha} \in (0.25, 0.75))}{P(\bar{\alpha} \notin (0.25, 0.75))}$$

The two indices were then thresholded to generate a 2-way classification of all triplets. On waviness, a triplet was categorized as Wavy, Flat or Ambiguous according to whether $r_w > 1.3$, $r_w > 0.77$, or, $0.77 \leq r_w \leq 1.3$, respectively. On extremeness, the categories were Central, Extreme, or, Ambiguous according to whether $r_c > 3.24$, $r_c < 1.68$, or, $1.68 \leq r_c \leq 3.24$, respectively.

In addition to the Flat/Wavy and Extreme/Central classification, a third parameter was evaluated for each triplet: the degree of skewness in the distribution of $\bar{\alpha}_*$:

$$r_s = \max\left( \left\{ \frac{P(\bar{\alpha}_* < 0.5)}{P(\bar{\alpha}_* > 0.5)}, \frac{P(\bar{\alpha}_* > 0.5)}{P(\bar{\alpha}_* < 0.5)} \right\} \right)$$

which ranges in $(1, \infty)$. Each triplet's Flat/Wavy/Central/Extreme tag could then be subcategorized as either Skewed or Symmetric depending on whether $r_S > 4$ or $r_S < 2$ (with no label in the middle). This subcategorization step was useful for distinguishing the dynamic admixtures associated with the whole-trial categorizations of Single and Outside from Intermediate and Mixture, with Single and Outside tending to be classified as Skewed. Supplementary Table 1 and Supplementary Figures 6 and 7 give the full results of the main 2-way classification together with the symmetry/skewness subclassification, cross tabulated with the classification done under the whole trial spike count analysis.

**A vs. B assignment scores**. A vs. B assignment scores were computed for several analyses (the example shown in Fig. 1e, f; pairs of recorded neurons; the relationship between spiking activity and local field potential; and the relationship between saccade sequences and spiking activity). For each triplet, every dual-sound trial received an A-like score and a B-like score, either for the entire response window (600–1000 ms after sound onset) or for 50 ms time bins. The scores were computed as the posterior probability that the spike count in each dual-sound trial was drawn from the Poisson distribution of single-sound spike counts.

For the pairs analysis, the A vs. B assignment scores were computed within each 50 ms time bin independently for each pair of neurons recorded simultaneously. The scores were normalized across trials by subtracting the mean score and dividing by the standard deviation of scores for that bin (a Z-score in units of standard deviation). Only conditions for which both recorded neurons exhibited reasonably different responses to the A vs. the B sound and for which there were at least five correct trials for A, B, and AB trials were included (t-test, $p < 0.05$). A total of 206 conditions were included in this analysis.

**Local field potential analysis**. We analyzed the local field potential from 87 sites in both monkeys (30 sites from monkey P's left IC, 31 sites from monkey Y's right IC and 26 sites from monkey Y's left IC). The LFP acquisition was either recorded in discrete temporal epochs encompassing behavioral trials (roughly 1.2 to 2 s long) and at a sampling rate of 20 kHz (Dataset I, part of Dataset II), or as a continuous LFP signal during each session, at a sampling rate of 20 or 1 kHz (rest of Dataset II). We standardized the LFP signals by trimming the continuous LFP into single trial intervals and down-sampling all signals to 1 kHz. The MAP system filters LFP signals between 0.7 and 300 Hz; no additional filtering was applied. For each site we subtracted the overall mean LFP value calculated over the entire session, to remove any DC shifts, and we excluded trials that exceeded 500 mV.

For the voltage-and-time domain analysis presented in Fig. 5a, for each triplet, we assigned individual dual-sound trials to two groups based on the total spike count in a 600 ms response window (see Methods: A vs. B assignment scores). The average LFP was then compared across the two groups in two 600 ms windows before and after sound onset (baseline and response periods). The results reported here refer to these mean-normalized LFP signals. We obtained similar results when the amplitude of each trial's LFP was scaled as a proportion of the maximum response within the session.

**Face patch MF recording procedures**. The full experimental procedure is described in[37]. We give a summary here. All procedures conformed to the US National Institutes of Health Guide for Care and Use of Laboratory Animals, and were approved by The Rockefeller University Institutional Animal Care and Use Committee (IACUC).

The localization of face patch MF was guided via fMRI as described in[37]. A total of 105 single neurons were recorded from MF in two male adult macaques (monkey Q, *Macaca mulatta*, and monkey J, *Macaca fascicularis*). The monkeys were head-restrained and performed a fixation task while viewing visual stimuli on a CRT screen placed 57 cm in front of them. Eye position was monitored with an infrared pupil tracking system (ETL-200, ISCAN Inc.,Burlington, MA). The monkeys were rewarded with juice for maintaining the eyes within a $\leq 2 \times 2$ degree square window around the fixation point.

All stimuli were controlled by custom software written in C (Visiko) running on a Windows PC. For each neuron, three visual stimuli (400 ms, $4 \times 4$ degrees in size) were selected from among a pool of face and object stimuli: a face that elicited a strong response, dubbed the preferred face; a face that elicited a weak or no response, dubbed the non-preferred face; and an object that also elicited a poor response, dubbed the non-preferred object. They were presented either alone or in pairs consisting of the preferred face and one or the other of the non-preferred stimuli. Thus, there were two triplets per cell suitable for analysis. The stimuli were randomly interleaved with each other and with other conditions not analyzed here see[37].

Stimulus positions on the screen were such that the preferred face was always at the center of the neuron's receptive field whereas the non-preferred stimulus could occupy one of eight equidistant locations adjacent to the preferred face. The exact location of the non-preferred face/object was ignored in the present analysis, but excessive heterogeneity in the responses due to variation in location would have caused the triplet to be excluded on the grounds of not exhibiting a sufficiently Poisson-like spike count distribution on the relevant single-stimulus trials.

The data were otherwise analyzed as described in the preceding Analysis section.

**Data availability**. The data and computer code that support the findings of this study are available from the corresponding authors upon reasonable request.

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

## Acknowledgements

We are grateful for expert technical assistance from Jessi Cruger, Karen Waterstradt, Christie Holmes, Stephanie Schlebusch, Tom Heil, and Eddie Ryklin. We have benefitted from thoughtful discussions with Michael Lindon, Liz Romanski, Marlene Cohen, Nicolas Brunel, Aaron Batista, Stephen Lisberger, Marty Woldorff, Daniel Pages, David Bulkin, Kurtis Gruters, Bryce Gessell, Na Young Jun, Cindy King, Luke Farrell, and David Murphy. We thank Bao Tran-Phu, Will Hyung, Stephen Spear, Francesca Tomasi, and Ashley Wilson for assistance with animal training and/or recordings. Financial support for the research was provided by the National Science Foundation (0924750) to J.M.G. and the National Institutes of Health (5R01DC013906-02) to S.T.T. and J.M.G.

## Author contributions

V.C.C. and J.M.G. coordinated the project; V.C.C and J.L. recorded the IC data; A.F.E. and W.A.F. designed and executed the visual face patch recordings; S.T.T. and C.G. developed new statistical methods; S.T.T., C.G., A.Z., V.C.C., J.L., J.T.M., S.M.W., J.M.G.

analyzed the data; V.C.C., J.M.G., S.T.T. wrote the manuscript; V.C.C., J.T.M., S.M.W., J.M.G., S.T.T., W.A.F., R.E. reviewed the manuscript

## Additional information

**Competing interests:** : The authors declare no competing interests.

