## [Peer Review File · Nature Communications]

Reviewers' comments:

Reviewer #1 (Remarks to the Author):

The authors of the manuscript address the problem how the brain preserves information about multiple simultaneous items. Specifically, they test the hypothesis that in the presence of more than one stimulus, neurons capable to respond to both of them engage in time division multiplexing of their firing rate to preserve information about multiple stimuli. Their major claims are,

(1) that part of the neurons in the inferior colliculus, an important subcortical structure in the ascending auditory pathway, show indeed responses that fluctuate between two different rates, each expected for one of the two different auditory stimuli presented simultaneously,

(2) that these fluctuations correlate between neurons,

(3) that the stimulus indicated by the firing rate directly after stimulus onset can be predicted from the local field potential, and

(4) that the stimulus reported first (by gazing to the location of the sound source) can be predicted from the firing rate fluctuations.

They conclude from their experimental observations that such alternating activity patterns corresponding to multiple items may therefore reflect a general strategy to enhance the brain's processing capacity. Furthermore, they suggest that these fluctuating patterns indicate a potential connection between three disparate phenomena, including neural firing variability, neural oscillations, and limits in attentional/memory capacity.

The main topic of the manuscript is of great and broad interest in the neurosciences. The experimental data convincingly demonstrate that responses fluctuate slowly between levels expected for the two different auditory stimuli presented simultaneously. If this would indeed reflect time division multiplexing for preserving information about multiple simultaneous items, the paper would certainly be highly influential.

My main concern is the interpretation of the data in terms of time division multiplexing for preserving information about multiple simultaneous stimuli.

1) Since the seminal work of Moran and Desimone (Science 229:782-784; 1985) it is well known that selective attention switches the response of neurons that are simultaneously confronted with multiple stimuli in their receptive field between different levels corresponding to the strength observed, if only the attended stimulus is shown. Switching between different levels of firing rate, power of oscillatory activity, and correlation structure in dependence of cognitive processing is therefore a well-established phenomenon. If the time course of the cognitive process governing these different states would not be known, this would result in very similar fluctuations of the response level as described in the present manuscript. This raises the question whether the observed fluctuations reflect uncontrolled variations of cognitive processes that require selective processing of one of the stimuli rather than time division multiplexing for preserving information about multiple simultaneous items. Selective attention for one of the stimuli might be such a source of rather slow switching between different response levels. This possibility is not entirely implausible for several reasons: (1) The behavioral task does not enforce selective attention in a certain direction. It may sometimes be directed to the A and sometimes to the B stimulus, and in part of the trials it may switch from one to the other. This would predict exactly the slow fluctuations observed as well as their correlation. (2) The behavioral result would fit too: The animals tend to saccade first to the currently attended stimulus. (3) The task does not strictly require keeping representations of both stimuli active. If the animal recognizes that more than one stimulus is present, it would be sufficient to attend one of the two stimuli to identify its location. This location predicts the second location (a stimulus at the position close to the midline is always associated with a second stimulus at the position more distant from the midline at the other side of the midline and vice versa). Generating a motor plan for the two saccades therefore does not need to localize the position of both stimuli. Such an interpretation would be in line with a large part of trials in which no within-

trial switching is observed, but a constant rate fitting the expectation for one of the stimuli throughout the trial. The authors may have good reasons to exclude this possibility. However, it appears to be so straight forward that they should inform the reader on the reasons to discard such an interpretation of their data.

2) A second, conceptual problem associated with the interpretation of the data as evidence for time division multiplexing is the switching frequency of this multiplexing. The authors explicitly relate their understanding of time division multiplexing to the common meaning in telecommunication (see lines 27 and 28). According to this context, a certain resource for data transmission (like a transmission line) or for data processing (like an analog-to-digital converter) is switched between multiple data sources with a frequency high enough to have the information of all sources available within a time span that allows a subsequent stage to assess the information of all sources as if they would be simultaneously present. Typically, the multiplexing frequency is at least 'n' times higher than the processing frequency of the stage receiving the information, with 'n' being the number of multiplexed data sources. This suggests that the multiplexed information source in the auditory pathway should alternate with a sufficiently high frequency. The authors need to explain how this expectation reconciles with a major part of the trials that show not even one switch between the two stimuli throughout the whole trial.

3) The prediction of greater oscillatory activity in the LFP for trials with evidence for within-trial switching suggests that the frequencies that reflect half of the switching frequency should be enhanced. Since the acquired evidence seems to suggest not more than one or two changes per trial, it is not clear why the increase apparently covers a wide range of frequencies, at least up to 200 Hz. This (apparent) contradiction needs to be resolved.

Line 28-30

Line 44-46

For clarity it should be stated clearly which property of the LFP is predictive.

Line 90, 91

This might be partially misleading. As argued above, it might be sufficient to keep a motor plan which is only indirectly "information about both sound items".

Line 154

Aside from the number of triplets, the number of neurons from which these 363 triplets derive should be reported.

Line 196

Aside from the number of triplets, the number of neurons from which these 72 triplets derive should be reported.

Line 243

The claim that "The DAPP model fits smoothly time-varying weights (α and $(1 - \alpha)$) ..." is difficult to follow since there is no typical example showing the time-varying weights.

Line 288-290

It is not clear why the whole "intermediate" category should be associated with stronger oscillations. Those cases which are categorized "flat" and "central" seem to change their state only from trial to trial, corresponding to very low frequencies. Since there is no description how the plots in figure 3D were generated, it is unclear how such very low frequencies would have been analyzed within trial periods that may hardly contain a full wave. Furthermore, such slow fluctuations may reflect any changes occurring in pace with the trials. Please clarify the rationale for including cases which are

categorized "flat" and "central" and the methods for analyzing the frequency content.

Line 291

For the same reason as above: Wouldn't it make more sense to compare the cases which are categorized "wavy" and "central" against those cases which are categorized "flat" and "central" plus the mixed cases?

Line 310-334

The similarity measure computed here is most likely fairly close to a kind of normalized spike rate measure yielding essentially the well-known spike rate correlations. The main difference arises if the cells have opposite preferences (in terms of spike rate) since in that case the correlation measure computed here should become positive whereas a spike rate correlation would be negative. To better understand how the result relates to the well-known spike rate correlations it would be helpful to know how many of the 91 triplets showed preference for the same stimulus for both neurons of a pair and how often it was opposite.

Within this context: The distribution of correlation values in Figure 4B and Figure 6 appears to be similar (please provide median values for the distributions). This would be expected in particular if most pairs have sites that both prefer the same sound source. If this is not the case, one would expect that the negative spike count correlations reflect cases with opposite preferences and these should become positive if not the spike count but the "A-likeness" is correlated. How do the authors explain the apparently similar and substantial proportion of negative correlations in both distributions?

Line 336-349

Since the discriminative feature of the LFP is the average value over the 600 ms before stimulus onset as suggested also by figure 4c, this finding deserves a discussion with an attempt to explain how such an apparently simple baseline shift before stimulus onset could bias the initial selection of the stimulus. The higher signal level for the A-like dual trials is associated with a higher response amplitude. Is this expected for contralateral stimuli? This higher response amplitude could explain the higher level of the pre-stimulus signal if e.g. the data acquisition system contains digital filters which are not strictly causal, such that values at a given point in time are influenced not only by previous but also by subsequent values.

Line 375

The term "correlations" is too unspecific. It would help the reader and prevent misunderstandings to use terms like spike count or rate correlations to inform the reader more precisely about the measure correlated.

Line 432

The discussion lacks a critical evaluation of alternative explanations, in particular with respect to attention as described above. Furthermore, selective attention has been related to sampling processes that might be related to time division multiplexing as described by the authors. A few related articles that might be helpful are:

- (1) Busch, Dubois, VanRullen (2009) The phase of ongoing EEG oscillations predicts visual perception. *J. Neurosci.* 29, 7869–7876.
- (2) Busch, VanRullen (2010) Spontaneous EEG oscillations reveal periodic sampling of visual attention. *PNAS* 107, 16048–16053.
- (3) Landau, Fries (2012) Attention Samples Stimuli Rhythmically. *Curr. Biol.* 22, 1000–1004
- (4) Holcombe, Chen (2013) Splitting attention reduces temporal resolution from 7 Hz for tracking one object to <3 Hz when tracking three. *J. Vis.* 13, 12.

Line 451-456

Since the authors claim in the subsequent discussion a close relation between firing pattern variability and time division multiplexing, they might consider the work of Mitchell, Sundberg and Reynolds (Neuron 63:879-888, 2009) who systematically tested different bin width for the computation of correlations and found that much of the variability originates from fluctuations below 5 Hz. 50 ms - bins would be well chosen to resolve such fluctuations.

Line 464-470

As recognized by the authors themselves, a general explanation of firing pattern variability by time division multiplexing suffers from the problem that probably the majority of corresponding studies used single stimuli and not multiple stimuli within the same receptive field. The suggested solution to assume that the missing stimuli are "stimuli not deliberately presented by the experimenter, memories of previous stimuli, or mental imagery" is not convincing. If these sources would cause responses that need to be multiplexed with the single stimulus present, these responses should show up during trials with no stimulus shown or in pre-stimulus periods. Such irregularly occurring responses would quickly indicate that the neuron is driven by unintended stimuli or processes. Since this in general is not the case it is difficult to see how time division multiplexing could be a general explanation for the variability of neuronal responses.

Line 475-481

Time division multiplexing would indeed explain why a limited number of items larger than one can be simultaneously processed. However, without citing examples in which the assumption of their competition for the same set of neurons is confirmed, a typical alternative is that multiple sets of neurons implement parallel processing, especially in neural structures with well-ordered maps. The argument would be therefore more convincing if the authors could cite specific examples in which such parallel processing is excluded instead of referring to working memory or selective attention in general.

Line 486-491

These statements are all true, but it is not clear how they relate to the topic discussed. Furthermore, the relation between time division multiplexing and oscillations remains somewhat unclear. Do the authors want to suggest that in general oscillations are a consequence of time division multiplexing? How would that be reconciled with numerous studies that propose different functional roles for oscillations?

Line 495-500

Here, the authors seem to raise the important question which mechanism causes the multiplexing. The subsequent statement terminating this consideration that "it seems likely that the externally-multiplexed sound waves interact with neural circuit states at the time that the incoming signal arrives to govern how individual neurons respond on a moment by moment basis" is hard to understand and far from a well-defined explanation or hypothesis. The paragraph is therefore not supportive for the discussion of the results.

Line 513-531

The description of the models is not sufficient to understand them easily. Furthermore, it is not clear whether the claim that they have developed them includes any kind of formal simulation or testing. The authors may consider whether they are required for the message they want to convey.

Line 1001

From which monkey is the data shown in that figure?

Reviewer #2 (Remarks to the Author):

The manuscript shows sensory responses in macaque inferior colliculus appear to represent concurrent multiple sounds as time division multiplexing, it is temporal flipping of neuronal activity between coding of single sounds, like binocular rivalry of visual systems flipping between eyes. The analyses show it could occur in trial-wise manner between trials and moment-wise manner within each trial, while the latter conform to the time division multiplexing. The hypothesis is novel. The analyses developed to address the presence of such coding scheme look pretty and parametrically differentiating the types of multi-sound coding successfully in subsets of data. Additionally, the ms. shows similar coding of multiple s of multiple faces in inferior-temporal face patch. It is interesting to see similar results in brainstem and high cortical region of different sensory systems. I think the results appeal well, and the ms. is a good candidate for publication. One thing to note. Though I perhaps followed the model description except for the algorithm estimating ϕ and ψ , it would be better if authors give extra efforts to explain mathematical terms for neuroscientists who would be reading the paper as much as they can. I listed my concerns below.

(1) According to the results, larger fraction of units was flat-extreme than intermediate-wavy. Correct? However, most of discussion focuses on the latter. That could be potentially misleading readers about how IC codes multiple sounds. Since most of single units are recorded in separate occasions, it may be possible that the manner animals paid attention to sounds could have shifted between recordings, and resulted in different schemes of coding. I guess authors can deny such possibilities I just describe. However, it may be fair to say that IC neurons code multiple sounds by the flat-extreme way more often than 'wavy', based on the given data. Probably it will need larger database to clarify such possibilities though.

(2) Many units and pairs of units are registered in the data set multiple times as different pairs of sounds are examined. One worrisome is that category of "wavy" units, for example, consisted of a few units counted multiple times to create biased results to have larger fraction than "flat". Are there any assurances of having no such biases?

Somehow related. Since some units are counted multiple times, do such units exhibit similar common response type categories "wavy" or "flat extreme" among different triplets, or do they switch response categories between triplets of sounds?

(3) I guess sound A could be presented from left or right speaker. However, IC neurons are sensitive to sound localization. Did Authors use only particular lateralized cases (e.g. sound A from only left and sound B from only right) or combine sound A from left and right? I am not sure if the ms. clarify this.

(4) What are the response thresholds relative to the sound levels of stimuli? Did all sampled units have monotonic rate-level (not azimuth but sound level) functions? A chief concern is that very small fraction of finding single units as the "summation-like" or the "outside" category, particularly in additive direction, may be simply due to the sound level relative to single units' threshold levels.

(5) Spectrograms of LFP. While DAPP-alpha functions go up and down slowly, why not checking whether LFP changed in the similar time scale? Thresholded test of spectrograms seem to show significant difference in that low frequency end, too. I think it is better to use wavelet transforms to reveal the frequency range similar to the dynamics of alpha in addition to high frequency gamma. Also, low frequency part may differentiate further between "intermediate flat" and "intermediate wavy," if it follows DAPP-alpha.

(6) What is the reaction time? Does it correlate with neuronal activity: e.g. more A-like responses give faster saccade to A? Is the first reaction time to dual sounds same as those to single sounds?

(7) Please make the statistics reporting consistent through the ms. Some p-val are not clear what statistics tests are used, and degree of freedom. Also, it is better to describe what statistics tests and decision criteria in Methods.

(8) Line 91 and Suppl. Fig. 1. Are the "71% and 77%" based on a typical session or multiple

sessions?

(9) Lines 114-117. According to the task training described in the Methods, animals were never encouraged to find a target in the middle. In other words, midpoint never was an option for monkeys to choose. Correct? If yes, then, the statements in the lines "... inconsistent with the behavioral results ... monkeys should respond to dual sounds as if ... midpoint," may be overstating, and need an additional remark that animals did not have such option.

Related. Were animals presented with dual sounds both coming from midpoint, or even single sounds from midpoint? Such trials may serve as controls.

(10) Ordinate axes labels in Figure 2: "Probability (%)." I am not sure if this is legitimate expression, because probability is usually between 0 and 1, as described in the legend for Fig. 2F. Also, I guess, in Bayes analyses and alpha in DAPP models, probabilities are calculated to range from 0 to 1. So, the word "probability" is interchanging somehow. So, I suggest changing "12" to "0.12" for changing "Probability (%)" to "Fraction (%)" or "fraction of trials" just in case. Another suggestion could be to remove ticks for axes in Figs. 2A-D, similarly to Fig. 3A3, since those are rather schematic illustrations of distributions.

(11) Lines 209-211. When neurons were categorized as "Single," did they actually exhibit significant responses to both sounds A and B? If not (i.e. neurons responded to only A), categorizing "single" may be due to stimuli selected for such neurons. Regardless of the answer is yes or no, this point is better clarified.

(12) Line 227. Left bracket at the beginning of the line is probably not needed. A period "." is probably needed after "A and B" in the middle of the line.

(13) Fig. 3B, alpha curves. Are those smoothed? I thought those should have steps every 50 ms.

(14) Line 258. Should "distribution" be "distributions"?

(15) Line 263. Should "is not showed" be "is not shown"?

(16) Fig. 3D. average local field potentials. Are those spectrograms of averaged LFP or averages of spectrograms of LFP? Average of how many? How were spectrograms derived? FFT? Please describe what was done.

(17) Fig. 4B. Do the distributions of xcorr coefficients differentiate after sorting pairs by paired types of categories like "wavy"- "flat-extreme", "wavy"- "wavy"? It would be interesting if positive.

(18) Unit of LFP. Was it really milli-volt? Were signals recorded after amplification? What's "N=274" in Fig. 4C, the number of sites or triplets? Figure legend says 1902 and 1618 trials.

Line 355. What feature of LFP was analyzed, and corresponded to the "state of activity"?

Line 352 says "mean LFP value". Does it mean the mean magnitude of LFP during the 600 ms window? Since LFP has apparent dynamic change ranging positive and negative during that period in Fig. 4C, isn't it better to use some other parameters like peak amplitude? It would nice to see error bars or confidence intervals of LFP traces.

Also, Figure 4C. In addition to units' firing during A-Like trials look like that of sound-A alone trials, do LFP of A-like trials look like LFP during sound A-alone trials?

(19) Fig. 4D. For this analysis, are there no difference when compared sound A-left and sound A-right?

(20) What are the numbers superimposed on bar graphs in Fig. 2F and Fig. 5A? In Fig. 2F, the numbers seem the numbers of units, as they seem to correlate with the confidence level fractions within bars. But, the numbers in Fig. 5A seem cumulative of fractions.

(21) Line 694 in Sound localization task, "Figure 2A" is probably "Figure 1B."

(22) Line 841. Which section is the "Events of Task"?

Reviewer #3 (Remarks to the Author):

This manuscript presents evidence to suggest that neurons in the IC of the monkey multiplexes information of about the locations of the sources of two, simultaneously active sounds. This is an

interesting and novel idea with parallels to artificial communications systems. Supplemental data from the face area of the cortex are also reported.

The ms is well written and illustrated, and the analyses of spike trains via DAPP is rigorous. However, the acoustics are not treated rigorously, and there are a number of questions, control experiments, and comments that I would like to see addressed:

1. I would like to read a bit more about the need for multiplexing in the sound-localization task. The results show evidence of multiplexing in a condition where it is not necessary. Since the A and B stimuli differ by more than about a third of an octave, I can imagine that there will be activity in two, separate iso-frequency laminae. These separate patches of activity would differ in their firing rate assuming that source-location is rate-coded, so the two sources should be separately localizable with or without multiplexing.

Conversely, multiplexing would be needed if sounds A and B, presented from different loci, had the same amplitude spectrum (but uncorrelated fine structure). Was this condition tested?

2. If the multiplexing enables the representation of the loci of the two sound sources, DAPP should show no evidence of multiplexing when sounds A and B are presented from the same location. Is this the case?

3. What is the difference between the center frequencies of sounds A and B relative to a cell's frequency tuning curve? If the center frequency of one of the sounds is outside a cell's tuning curve, the cell should not fire to it, and the multiplexing phenomenon should not be observed. It would be helpful to see the frequency tuning curves of the neurons shown in Figures 1E and F and some description of the breadths of the frequency tuning curves in the neurons sampled and their relationship to stimulus A and B.

4. I understand the rationale for integrating spike counts in 50-ms windows - the DAPP model assumes that source-locations are encoded in the number of spikes. However, perhaps a more direct test of the multiplexing hypothesis would be to examine the neuronal responses to the envelope of the frozen stimuli. Neurons in the IC lock their discharge to the envelope of the stimulus. Since the narrowband noises are frozen, their repeated presentation should reveal a PSTH that is typical of A or B. In the responses to dual-sound stimuli, the PSTH patterns should alternate between the A PSTH and the B PSTH if multiplexing occurs.

5. If the spike trains of IC neurons alternately represent sources A and B, does this mean that while the neuron is firing at a rate that encodes source A's location, is information about B's location momentarily lost?

Overview of response to reviewers:

We would like to thank all three reviewers for their comments. The reviews stand out for their combination of insight and thorough attention to detail, presented in a collegial package - a pleasure to work with. (We hope this is ok to say...)

We think we have satisfactorily addressed all comments, and in general the point by point reply should be straightforward to follow.

In particular we have added an extensive consideration of the relationship between our study and attention to the Discussion section. This section embraces the conceptual connection between time division multiplexing, the biased competition theory of attention (as distinct from the spotlight theory), and perceptual evidence for serial processing of multiple stimuli. In the response to reviewers, we also discuss the relationship between our study and classic studies such as those by Moran and Desimone, 1985 and Reynolds et al. 1999 as well as additional datasets and analyses that speak to the question. In short, regardless of nomenclature, we believe that our study significantly extends what is known about how the brain processes multiple items in a fundamentally new way.

The changes are detailed below and a “track changes” version of the manuscript is provided for your convenience. We’ve also made some minor optimizations to Figures 1 (adjusting the colored “PSTH”s) and 2 (rearranging the panels in 2E).

Reviewer #1 (Remarks to the Author):

The authors of the manuscript address the problem how the brain preserves information about multiple simultaneous items. Specifically, they test the hypothesis that in the presence of more than one stimulus, neurons capable to respond to both of them engage in time division multiplexing of their firing rate to preserve information about multiple stimuli. Their major claims are,

- (1) that part of the neurons in the inferior colliculus, an important subcortical structure in the ascending auditory pathway, show indeed responses that fluctuate between two different rates, each expected for one of the two different auditory stimuli presented simultaneously,
- (2) that these fluctuations correlate between neurons,
- (3) that the stimulus indicated by the firing rate directly after stimulus onset can be predicted from the local field potential, and
- (4) that the stimulus reported first (by gazing to the location of the sound source) can be predicted from the firing rate fluctuations.

They conclude from their experimental observations that such alternating activity patterns corresponding to multiple items may therefore reflect a general strategy to enhance the brain’s processing capacity. Furthermore, they suggest that these fluctuating patterns indicate a potential connection between three disparate phenomena, including neural firing variability, neural oscillations, and limits in attentional/memory capacity.

The main topic of the manuscript is of great and broad interest in the neurosciences. The experimental data convincingly demonstrate that responses fluctuate slowly between levels expected for the two different auditory stimuli presented simultaneously. If this would indeed reflect time division multiplexing for preserving information about multiple simultaneous items, the paper would certainly be highly influential.

My main concern is the interpretation of the data in terms of time division multiplexing for preserving information about multiple simultaneous stimuli.

1) Since the seminal work of Moran and Desimone (Science 229:782-784; 1985) it is well known that selective attention switches the response of neurons that are simultaneously confronted with multiple stimuli in their receptive field between different levels corresponding to the strength observed, if only the attended stimulus is shown. Switching between different levels of firing rate, power of oscillatory activity, and correlation structure in dependence of cognitive processing is therefore a well-established phenomenon. If the time course of the cognitive process governing these different states would not be known, this would result in very similar fluctuations of the response level as described in the present manuscript. This raises the question whether the observed fluctuations reflect uncontrolled variations of cognitive processes that require selective processing of one of the stimuli rather than time division multiplexing for preserving information about multiple simultaneous items. Selective attention for one of the stimuli might be such a source of rather slow switching between different response levels. This possibility is not entirely implausible for several reasons: (1) The behavioral task does not enforce selective attention in a certain direction. It may sometimes be directed to the A and sometimes to the B stimulus, and in part of the trials it may switch from one to the other. This would predict exactly the slow fluctuations observed as well as their correlation. (2) The behavioral result would fit too: The animals tend to saccade first to the currently attended stimulus. (3) The task does not strictly require keeping representations of both stimuli active. If the animal recognizes that more than one stimulus is present, it would be sufficient to attend one of the two stimuli to identify its location. This location predicts the second location (a stimulus at the position close to the midline is always associated with a second stimulus at the position more distant from the midline at the other side of the midline and vice versa). Generating a motor plan for the two saccades therefore does not need to localize the position of both stimuli. Such an interpretation would be in line with a large part of trials in which no within-trial switching is observed, but a constant rate fitting the expectation for one of the stimuli throughout the trial. The authors may have good reasons to exclude this possibility. However, it appears to be so straightforward that they should inform the reader on the reasons to discard such an interpretation of their data.

We completely agree that there is a close conceptual connection between our work and studies of attention. We might simply invert the framing: we think that our study has much to offer the attention literature by shedding light on what may be happening “under the hood” and generally expanding the scope of the possible role of attention-like processes in the brain.

We have expanded the Discussion section to incorporate this topic more fully (see new paragraphs 7-10 in the Discussion). We now review the exciting behavioral evidence of sampling, serial processing, and periodicity of perception (per suggested references below under point “Line 432”), before turning to neural theories and experiments. Our general approach is to embrace the attention - time division multiplexing connection as mutually important concepts. We suggest the reviewer read the new manuscript section first and then return here for some additional thoughts on the connection between our current study and the attention literature.

One important factor should be noted: there are two competing frameworks for thinking about attention - biased competition vs. enhanced gain. Our analysis demonstrates fluctuating activity in relation to anchoring response distributions - the responses to each single stimulus alone - and is thus much closer conceptually to the biased competition theory. Studies that have demonstrated enhanced gain of responses to attended stimuli (or reduced gain to unattended stimuli) may be relevant but in general the analysis methods focus on comparing the response to the stimulus when it is attended vs when it is not. It is difficult to compare our results to such studies because the response to the second stimulus is not incorporated into the analysis. We also note that nearly all of the conceptually relevant work is in the visual system; single-unit studies of auditory attention are few and far between and the lack of auditory maps of space means that extrapolating from visual studies is not trivial. More on this below.

The reviewer notes the seminal work of Moran and Desimone (Science, 1985), which revealed that responses in V4 are strongly modulated by whether attention is directed toward vs. away from the stimulus in the receptive field. This work was followed up by numerous other studies that verified, extended, and refined this observation. One of the most relevant to our approach in terms of quantitative analysis is the 1999 study by Reynolds, Chelazzi and Desimone (Journal of Neuroscience), which fits both V2 and V4 responses to combined stimuli as a weighted average of the responses to the two component stimuli when presented individually. The magnitude of the weight varied depending on the attentional condition - the weighting was about equal when neither stimulus was attended and shifted to about 70:30 (V2) or 80:20 (V4) in favor of the attended stimulus in the attended condition.

To our knowledge, the partial nature of this shift has not been explained empirically, and neural fluctuations provide an obvious possible account. Put another way, this study and others like it have generally attempted to prevent attention from fluctuating as it might do naturally. That attention and underlying neural signals might actually fluctuate could account for why responses to combined stimuli tend to lie between the responses to component stimuli - attention does not bias the competition so completely as to utterly vanquish the response unattended stimulus. Thus, our study provides an important advance to the attention field by providing a possible explanation for this observation as well as an analytic method for testing it.

Extrapolating theories and experimental predictions from vision to hearing is not trivial. Since the mammalian representation of auditory space for the most part does not involve circumscribed receptive fields, it is not clear how to apply either the biased competition or attentional spotlight concepts of attention to auditory space. It is possible that the same concepts that have been developed in the visual attention field can be applied to auditory attention in the frequency domain, but work on this question is decades behind the field of vision.

With regard to the possibility raised by the reviewer that the monkeys might have been only processing one stimulus prior to the first saccade, we have evidence that speaks to this point from a followup study we are currently conducting. This study involves the superior colliculus' representation of simultaneous visual and auditory stimuli at the same location in space. This situation is akin to the proposed possibility, because only one location and saccade is involved, but because there are two stimuli that evoke different activity levels, it is also possible to evaluate the presence of fluctuating signals. Indeed, we do see evidence of fluctuating activity patterns, suggesting that SC neurons fluctuate between encoding the sound and encoding the visual stimulus. This tends to argue against a simple coding-of-one-location-at-a-time explanation (albeit in a different brain structure).

Superior Colliculus:
Combined visual-auditory stimuli at the same location in space

With regard to the predictability of the locations and whether that could indicate the monkeys had a memorized or stereotyped response elicited whenever the two frequencies were presented: one monkey has subsequently been tested in a paradigm in which both frequencies are presented from the same location in space. This monkey performed this task successfully, i.e. by making single saccades to the single location, on the first day she was exposed to the task. Below show the data; the red traces involve trials with the left hand location and the blue traces involve trials with the right hand location. For clarity of visualization, a randomly selected subset of 40 trials for each target location every other trial is shown. The data has not been filtered for “correctness” but includes any error trials that occurred. This shows that this monkey correctly understood that the point was to report the locations of the stimuli.

Monkey Y - Performance on first day with two frequencies presented from same location

Saccade to:

----- Target -12° 76% correct

----- Target 12° 89% correct

Shown are N=40 trials selected randomly from all attempted trials

We have since trained two additional monkeys on variants of this task - one on the visual-auditory version as noted above and one on the auditory-auditory version. Both monkeys learned the task quickly and with no obvious problems.

To summarize, we agree with the reviewer that the theory of time division multiplexing has a very important relationship to theories of attention, and the analyses and observations presented here constitute a conceptual advance that (we hope) will be of great interest to the attention community. We appreciate very much the suggestion to flesh out this conceptual connection and we hope we have done so satisfactorily.

2) A second, conceptual problem associated with the interpretation of the data as evidence for time division multiplexing is the switching frequency of this multiplexing. The authors explicitly relate their understanding of time division multiplexing to the common meaning in telecommunication (see lines 27 and 28). According to this context, a certain resource for data transmission (like a transmission line) or for data processing (like an analog-to-digital converter) is switched between multiple data sources with a frequency high enough to have the information of all sources available within a time span that allows a subsequent stage to assess the information of all sources as if they would be simultaneously present. Typically, the multiplexing frequency is at least 'n' times higher than the processing frequency of the stage receiving the information, with 'n' being the number of multiplexed data sources. This suggests that the multiplexed information source in the auditory pathway should alternate with a sufficiently high frequency. The authors need to explain how this expectation reconciles with a major part of the trials that show not even one switch between the two stimuli throughout the whole trial.

We believe that it is unlikely that time division multiplexing occurs in exactly the same way in biological systems as it does in technological ones. We appreciate the reviewer's point that using the analogy with telecommunications can carry with it meanings we cannot currently support. We have now added a paragraph addressing this issue directly (para 18, approximate line # 605 in the Discussion):

"Although we introduced the concept of time division multiplexing by drawing an analogy with telecommunications systems, these two models also help illustrate that multiplexing is unlikely to occur exactly the same way in biological systems as it does in technological ones. For example, the time course may be more fluid in biological systems, and different neural ensembles may operate asynchronously and/or aperiodically as opposed to operating under a regular, shared, clock cycle of some kind. The strongest shared element of the analogy with telecommunications is the transmission of more than one item or type of information via signal fluctuations at some time scale."

This is now included right after the discussion of the two models, which provide some specific ideas for how biological circuits might read out multiplexed signals. These two models both contain memory elements that serve to provide the problem of bridging between samples mentioned by the reviewer. (This solution is available in this context because the stimuli themselves are not changing rapidly, so remembering from a previous sample is good enough). Another possible solution to this is if different neural ensembles operate independently of each other; each may code one item at a time and the different items are preserved across multiple such ensembles. This is somewhat more distant from technological multiplexing, but is still analogous in that the same channel is being used to convey different information from trial to trial. It is possible these two types of neurons (within and between-trial switching) may reflect different functional populations, or different stages within the same circuit, but it is difficult to resolve this question with the current dataset.

3) The prediction of greater oscillatory activity in the LFP for trials with evidence for within-trial switching suggests that the frequencies that reflect half of the switching frequency should be enhanced. Since the acquired evidence seems to suggest not more than one or two changes per trial, it is not clear why the increase apparently covers a wide range of frequencies, at least up to 200 Hz. This (apparent) contradiction needs to be resolved.

Briefly (and see more detailed consideration under comments related to Line 288 below as well as Reviewer 2 point #5), we think the evidence for oscillatory activity at higher frequencies in the LFP indicates that there may be switching occurring at higher frequencies than is possible to ascertain with our current (spiking) data set. In particular, we draw the reviewer's attention to these two sentences at the end of section 2.4: "The LFP for "intermediate" sites showed higher energy across a range of frequencies, including frequencies well above the 20 Hz (50 ms) frequency range that we were able to evaluate at the spike-count single unit level. This suggests there may be activity fluctuations occurring at frequencies higher than we were able to test with the current analysis method."

Line 28-30

Line 44-46

For clarity it should be stated clearly which property of the LFP is predictive.

Agreed: we have modified this sentence to read: “The state of the network prior to stimulus onset (assessed by the mean voltage of the local field potentials) predicted the slower whole-trial spiking fluctuations”

Line 90, 91

This might be partially misleading. As argued above, it might be sufficient to keep a motor plan which is only indirectly “information about both sound items”.

Please see behavioral results above: when both frequencies were (unexpectedly to the monkey) presented at the same location, the monkey only made one saccade. We think this indicates that the monkey correctly understood the task as to localize whatever sounds were present.

Line 154

Aside from the number of triplets, the number of neurons from which these 363 triplets derive should be reported.

Thank you for catching this - the 363 triplets derive from 108 neurons and this is now stated.

Line 196

Aside from the number of triplets, the number of neurons from which these 72 triplets derive should be reported.

The 72 triplets derive from 46 neurons; this too has been added.

Line 243

The claim that “The DAPP model fits smoothly time-varying weights (α and $(1 - \alpha)$) ...” is difficult to follow since there is no typical example showing the time-varying weights.

We apologize but we are not sure what this comment refers to. The sentence indicated is the first sentence of the legend for Figure 3, which includes representative examples of time varying weights in panels B1, B2, and B3 (left panels, alpha vs. time for individual trials). If this is not what the reviewer was looking for, we would appreciate clarification.

Line 288-290

It is not clear why the whole “intermediate” category should be associated with stronger oscillations. Those cases which are categorized “flat” and “central” seem to change their state only from trial to trial, corresponding to very low frequencies. Since there is no description how the plots in figure 3D were generated, it is unclear how such very low frequencies would have been analyzed within trial periods that may hardly contain a full wave. Furthermore, such slow fluctuations may reflect any changes occurring in pace with the trials. Please clarify the rationale for including cases which are categorized “flat” and “central” and the methods for analyzing the frequency content.

Rationale: We agree; we consider this spectral analysis as insensitive to the frequencies that are assessed in either the whole-trial or within-trial analyses. The whole-trial fluctuations are too slow to show up at all in these plots, and even the within-trial analyses with the 50 ms binwidth will only reach 20 Hz if there is a switch at every bin; less frequent switches would involve energy at lower frequencies.

Rather, the point of this analysis is to take a peek at frequencies that are higher and which are impossible to analyze in the spiking domain.

Details: We have added a number of missing details to the figure legend and the methods section, specifically:

Figure legend: “Data are from dual-sound trials from triplets that were classified as “mixture” or “intermediate” with a winning probability of >0.95 . The left and central panels show the average of single trial spectrograms, plotted as absolute power in a logarithmic scale. There were $N=719$ “mixture” and $N=211$ “intermediate” trials. The right panel shows a thresholded ($p<0.001$) depiction of whether these differences were statistically significant according to a two-tailed t-test for each time point and frequency combination; red indicates the power for “intermediates” $>$ “mixtures” for that time frequency combination, blue indicates the converse, and green indicates the p value was >0.001 .”

Methods section “Local Field Potential” paragraph 2:

For the spectral analysis presented in Figure 3D, single trial spectrograms were computed using Short-Time Fourier transforms (spectrogram.m in MatLab Signal Processing Toolbox; MathWorks Inc.) with 256 ms segment size and 250 ms overlap for each frequency bin (0.7 - 200 Hz with 3.9 Hz bin size). Each segment was windowed with a Hamming window.

Error correction: In the process of revisiting this analysis, we discovered an error involving the squaring of the FFT amplitude values for the t-test panel; this error has been corrected. The t-test became more sensitive, such that the p -value of 0.05 was too lenient for clear visualization of the pattern. We have adjusted this cutoff to 0.001 in the revised figure.

Flat/central vs. wavy/central vs. flat/extreme:

The reviewer is absolutely right that comparison within these DAPP-defined categories would be a logical way to do the analysis. We have explored these comparisons, but we lose power because a smaller number of triplets can be included. In general, the results suggest that flat/central show more energy at higher frequencies than do either wavy/central or flat/extremes. We tentatively interpret this as suggesting that the wavy/central and flat/extremes are actually fairly similar to each other. For example, if switches tended to occur at a rate of about, say, 1 Hz, then luck of the draw might cause one triplet to exhibit a flat/extreme pattern, if the switches occurred predominantly during the intertrial interval, and another triplet to show a wavy/central pattern if the switches occurred predominantly during the sound presentation interval. This is speculative; we do not feel that we have the data or the analysis tools (yet) to resolve this question.

For the same reason as above: Wouldn't it make more sense to compare the cases which are categorized "wavy" and "central" against those cases which are categorized "flat" and "central" plus the mixed cases?

See answer above

Line 310-334

The similarity measure computed here is most likely fairly close to a kind of normalized spike rate measure yielding essentially the well-known spike rate correlations. The main difference arises if the cells have opposite preferences (in terms of spike rate) since in that case the correlation measure computed here should become positive whereas a spike rate correlation would be negative. To better understand how the result relates to the well-known spike rate correlations it would be helpful to know how many of the 91 triplets showed preference for the same stimulus for both neurons of a pair and how often it was opposite.

Since a) most IC neurons prefer contralateral locations, b) low-frequency-preferring neurons are over-represented (this is actually very typical of the IC which has an expanded low frequency representation; see Bulkin & Groh 2011 J Neurophys for details), and c) some of the pairs were recorded from the same electrode and thus sampled neurons from the same region of the IC's tonotopic representation, the vast majority of paired-recording triplets in our sample shared the same preference for "A" vs "B" sounds. We attempted to overcome this limitation by simultaneously recording from the ICs on opposite sides of the brain but gave up as the yield for this experiment was very low.

Despite the above, we do have some triplet-pairs that exhibited different preferences, and these appear to be about equally distributed on the positive and negative side of the distribution of correlation coefficients, both when computed as "Assignment scores" and when computed with spike counts. We now include panels to show this in Supplementary Figure 8 (formerly S6). Medians and means have been added to these graphs as well as Figure 4B.

Conceptually, it is easier to show population correlations if the true effect is all positive than if it is a mix of positive and negative. That is, it is possible for all neurons to fluctuate together, but it is not possible for all neurons to fluctuate exactly in opposite phase to all other neurons. If all are positive, then the overall distribution will be positive, whereas a mix may center on zero and appear as lack of correlation of any kind. So we are presently agnostic about whether the negative correlations that can be observed in some triplets in our data are real. We hope to revisit this question using U-probes that will facilitate more extensive investigation of the pattern of correlations across the neural population.

Line 336-349

Since the discriminative feature of the LFP is the average value over the 600 ms before stimulus onset as suggested also by figure 4c, this finding deserves a discussion with an attempt to explain how such an apparently simple baseline shift before stimulus onset could bias the initial selection of the stimulus. The higher signal level for the A-like dual trials is associated with a higher response amplitude. Is this expected for contralateral stimuli?

We suspect that the baseline shift reflects a difference in the relative levels of activity in the ICs contralateral and ipsilateral to the target that is saccaded to first. For example, if contra IC neurons

were slightly depolarized compared to the ipsi IC neurons, the contra target might dominate the representation and lead to that target being saccaded to first.

We have added a simplified version of this concept to the end of the paragraph in section 3.2: “What exactly that network state consists of is unknown; but it appears to derive from some form of altered balance in the levels of activity in the ICs contralateral and ipsilateral to the first-saccade target.”

This higher response amplitude could explain the higher level of the pre-stimulus signal if e.g. the data acquisition system contains digital filters which are not strictly causal, such that values at a given point in time are influenced not only by previous but also by subsequent values.

*We thank the reviewer for raising this important point. We have double checked the non-causal aspects of the data acquisition system and repeated (and replicated) the results using a baseline time window that is outside the range of potential influence of stimulus-driven responses. We also note that the *direction* of the effect is opposite - the post-stimulus response on contra-saccade-first trials is more negative voltages, whereas the pre-stimulus baseline shift is towards more positive voltages.*

Specifically, all data were recorded using the MAP system in Plexon. LFP signals came from the same electrodes used to record spiking data, but went through different filtering and amplification stages. In particular LFP signals were recorded through a Plexon HST/8o50-G20 headstage, connected to a Plexon PBX2/16FP-G50 preamplifier with a two-pole high-pass Butterworth filter (cut off at 0.7 Hz) and a four-pole low-pass Butterworth filter (cut off at 300 Hz). Such system can indeed introduce phase shifts in the LFP (REF: Nelson et al 2008, Wimmer et al 2016). At 3 Hz, there can be bleeding backwards in time up to about 16 ms and at 1 Hz that number becomes about 146 ms. Accordingly, we repeated the analysis in the window [-600, -200] ms prior to sound onset - thus in a window well separated from the sound response window. We find that the results are comparable to those obtained in the full window (-600 to 0 ms, pval = 0.0038; -600 to -200ms, pval= 0.0068).

References:

Nelson MJ, Pouget P, Nilsen EA, Patten CD, Schall JD (2008) Review of signal distortion through metal microelectrode recording circuits and filters. J Neurosci Methods 169:141–157, doi:10.1016/j.jneumeth.2007.12.010, pmid:18242715.

Klaus Wimmer, Marc Ramon, Tatiana Pasternak and Albert Compte (2016) Transitions between Multiband Oscillatory Patterns Characterize Memory-Guided Perceptual Decisions in Prefrontal Circuits, Journal of Neuroscience, 36 (2) 489-505; DOI: <https://doi.org/10.1523/JNEUROSCI.3678-15.2016>

Line 375

The term “correlations” is too unspecific. It would help the reader and prevent misunderstandings to use terms like spike count or rate correlations to inform the reader more precisely about the measure correlated.

With apologies, our line numbers and those that the reviewers had access too do not match exactly, but we think this comment refers to the legend of Figure 4. We have modified the sentence to insert “spike

count” as follows: “Pairs of neurons recorded simultaneously tended to show positive spike count correlations with each other”. If the reviewer is referring to a different location, please repeat the comment with reference to some other landmark and we’ll find it and make adjustments.

Line 432

The discussion lacks a critical evaluation of alternative explanations, in particular with respect to attention as described above. Furthermore, selective attention has been related to sampling processes that might be related to time division multiplexing as described by the authors. A few related articles that might be helpful are:

(1) Busch, Dubois, VanRullen (2009) The phase of ongoing EEG oscillations predicts visual perception. *J. Neurosci.* 29, 7869–7876.

(2) Busch, VanRullen (2010) Spontaneous EEG oscillations reveal periodic sampling of visual attention. *PNAS* 107, 16048–16053.

(3) Landau, Fries (2012) Attention Samples Stimuli Rhythmically. *Curr. Biol.* 22, 1000–1004

(4) Holcombe, Chen (2013) Splitting attention reduces temporal resolution from 7 Hz for tracking one object to <3 Hz when tracking three. *J. Vis.* 13, 12.

As described above, we have now added an extensive treatment of the relationship between attention and time division multiplexing. These suggested articles were particularly helpful and are cited and described. See new paragraphs 7-10 in the Discussion section.

Line 451-456

Since the authors claim in the subsequent discussion a close relation between firing pattern variability and time division multiplexing, they might consider the work of Mitchell, Sundberg and Reynolds (Neuron 63:879-888, 2009) who systematically tested different bin width for the computation of correlations and found that much of the variability originates from fluctuations below 5 Hz. 50 ms - bins would be well chosen to resolve such fluctuations.

Thank you for suggesting this reference, which does indeed support the choice of time scale for investigation. We have added a sentence on this at the end of the paragraph 3.1 “The 50 ms time scale is consistent with the frequency range in which spike-to-spike coherence has been observed in visual attention paradigms ³⁵”

Line 464-470

As recognized by the authors themselves, a general explanation of firing pattern variability by time division multiplexing suffers from the problem that probably the majority of corresponding studies used single stimuli and not multiple stimuli within the same receptive field. The suggested solution to assume that the missing stimuli are “stimuli not deliberately presented by the experimenter, memories of previous stimuli, or mental imagery” is not convincing. If these sources would cause responses that need to be multiplexed with the single stimulus present, these responses should show up during trials with no stimulus shown or in pre-stimulus periods. Such irregularly occurring responses would quickly indicate that the neuron is driven by unintended stimuli or processes. Since this in general is not the case it is difficult to see how time division multiplexing could be a general explanation for the variability of neuronal responses.

We acknowledge that this is a speculative point but there is some evidence for it: signals are indeed more variable in the pre-stimulus or non-stimulated periods (“Stimulus onset quenches neural

variability: a widespread cortical phenomenon”, Churchland et al., Nat. Neurosci., 13 (2010), pp. 369-378). We now mention this point explicitly: “Indeed, variability has been found to be higher when no deliberate stimulus is present at all [Churchland et al ref].”

Line 475-481

Time division multiplexing would indeed explain why a limited number of items larger than one can be simultaneously processed. However, without citing examples in which the assumption of their competition for the same set of neurons is confirmed, a typical alternative is that multiple sets of neurons implement parallel processing, especially in neural structures with well-ordered maps. The argument would be therefore more convincing if the authors could cite specific examples in which such parallel processing is excluded instead of referring to working memory or selective attention in general.

Good point; we have expanded the sentence “These limitations may stem from using the same population of neurons for each attended or remembered item, a situation that may arise whenever neurons are broadly tuned” to include the underlined phrase (Discussion paragraph # 4, approx line # 526). Some specific examples: in the IC, between 40-80% of neurons respond to 500 Hz tones; the situation only becomes worse when broadband stimuli are presented. For working memory and attention, parietal cortex may be an instructive example: the size of receptive fields is amply large enough that an experimenter could place all the stimuli to be remembered or attended within an individual neuron’s receptive field. To keep the discussion flowing along and in anticipation of possible page limits, we have not fleshed out this concept in great detail but we can certainly do more if this is a priority.

Line 486-491

These statements are all true, but it is not clear how they relate to the topic discussed. Furthermore, the relation between time division multiplexing and oscillations remains somewhat unclear. Do the authors want to suggest that in general oscillations are a consequence of time division multiplexing? How would that be reconciled with numerous studies that propose different functional roles for oscillations?

Our view is that future work should consider the possibility of a connection between oscillations and time division multiplexing. We would not want to claim that time division multiplexing accounts for all oscillations, or that oscillations are direct evidence of time division multiplexing. We have revised this paragraph to be a bit more clear about what our message is.

Line 495-500

Here, the authors seem to raise the important question which mechanism causes the multiplexing. The subsequent statement terminating this consideration that “it seems likely that the externally-multiplexed sound waves interact with neural circuit states at the time that the incoming signal arrives to govern how individual neurons respond on a moment by moment basis” is hard to understand and far from a well-defined explanation or hypothesis. The paragraph is therefore not supportive for the discussion of the results.

A problem with this paragraph was that it had multiple points that were not closely related. We have deleted the sentence referred to above as well as the lead in to it. We retained the point that sounds are mixed in the world, and blended this with the paragraph that follows concerning how the signals are demultiplexed.

Line 513-531

The description of the models is not sufficient to understand them easily. Furthermore, it is not clear whether the claim that they have developed them includes any kind of formal simulation or testing. The authors may consider whether they are required for the message they want to convey.

Thank you for catching this. We have revised the figure and added a more detailed description to the legend to Figure 6. The more complicated meter-to-map model is derived and expanded from a portion of a model that has been simulated at steady state in Groh and Sparks, 1992, Biological Cybernetics; this is now explained more clearly in the figure legend.

Line 1001

From which monkey is the data shown in that figure?

Apologies again but the line numbering in our copy is not exactly the same as the reviewer's copy. We think this comment may refer to the title of Supplementary figure 1: "Monkeys can localize two sounds". There was a typo in this title; it previously said "monkey can localize two sounds". We have corrected this. The data in this figure are from both monkeys.

Reviewer #2 (Remarks to the Author):

The manuscript shows sensory responses in macaque inferior colliculus appear to represent concurrent multiple sounds as time division multiplexing, it is temporal flipping of neuronal activity between coding of single sounds, like binocular rivalry of visual systems flipping between eyes. The analyses show it could occur in trial-wise manner between trials and moment-wise manner within each trial, while the latter conform to the time division multiplexing. The hypothesis is novel. The analyses developed to address the presence of such coding scheme look pretty and parametrically differentiating the types of multi-sound coding successfully in subsets of data. Additionally, the ms. shows similar coding of multiple s of multiple faces in inferior-temporal face patch. It is interesting to see similar results in brainstem and high cortical region of different sensory systems. I think the results appeal well, and the ms. is a good candidate for publication. One thing to note. Though I perhaps followed the model description except for the algorithm estimating ϕ and ψ , it would be better if authors give extra efforts to explain mathematical terms for neuroscientists who would be reading the paper as much as they can. I listed my concerns below.

Thank you for pointing out the difficulty with ϕ and ψ . We have added some intuitive explanation (Methods, 3rd paragraph under Dynamic Admixture Point Process Model): "Intuitively, these parameters can be thought of as related to the means and variances of the distribution of weight values α regardless of time within a trial, as well as the correlation between the weight observed at one point in time and the weight observed at another on any given trial." Please don't hesitate to point out other specific places where an intuitive explanation would be useful - this feedback is very helpful.

(1) According to the results, larger fraction of units was flat-extreme than intermediate-wavy. Correct? However, most of discussion focuses on the latter. That could be potentially misleading readers about how IC codes multiple sounds. Since most of single units are recorded in separate occasions, it may be possible that the manner animals paid attention to sounds could have shifted between recordings, and resulted in different schemes of coding. I guess authors can deny such possibilities I just describe.

However, it may be fair to say that IC neurons code multiple sounds by the flat-extreme way more often than ‘wavy’, based on the given data. Probably it will need larger database to clarify such possibilities though.

We agree with the reviewer’s concern about appearing to focus on the wavy-centrals which are comparatively rare. We have reviewed the Discussion section to ensure that both the trial & subtrial time scales are mentioned wherever one of them is. Our view is that it is premature to focus on one or the other; the greater evidence for flat-extreme/”Mixtures” may reflect the sensitivity of the statistical approach rather than reflecting the underlying features of the neural population. To better reflect this, we have added the following to Discussion paragraph 2: “Indeed, the preponderance of flat-extreme/”Mixtures” may not reflect the true state of the underlying population but rather the greater sensitivity of the analysis method for detecting slower fluctuations rather than faster ones.”.

We don’t see any particularly obvious patterns of change in the results across the duration of the experiments. Below is the data broken down into “first half” and “second half” of the included triplets. There is a higher ratio of “mixtures” to “intermediates” in the second half than the first, but both categories are well represented in both halves.

With regard to the point about attention, please see new paragraphs #7-10 in the Discussion as well as the response to reviewer 1.

(2) Many units and pairs of units are registered in the data set multiple times as different pairs of sounds are examined. One worrisome is that category of ‘wavy’ units, for example, consisted of a few units counted multiple times to create biased results to have larger fraction than ‘flat’. Are there any assurances of having no such biases?

Somehow related. Since some units are counted multiple times, do such units exhibit similar common response type categories ‘wavy’ or ‘flat extreme’ among different triplets, or do they switch response categories between triplets of sounds?

To answer the question somewhat narrowly, of 21 triplets classified as wavy, 19 were from different neurons. More broadly, we are still working on the best way to analyze whether the response patterns

are cell properties or a property of the combination of the neuron and conditions tested, so the following should be considered preliminary but we think it is satisfactory to address the reviewer's specific concern about whether a small number of neurons contributed all of the triplets within a particular category. We focus here on the whole-trial analysis because it provides a label for all of the includable-triplets and therefore provides the most power.

Below left is a color coded depiction of the classification in the whole-trial analysis of individual neurons with multiple triplets passing criteria for inclusion (responses to A-alone and B-alone differ, and both are Poisson). Triplet and Cell #s are arbitrary, and are sorted Mixture>Intermediate>Single>Outside, i.e. so the bottom left cell is the one with the most mixtures.

Manuscript dataset

From visual inspection, it is hard to tell whether there is consistency of classification of triplets within individual neurons. Indeed, re-running this analysis, but shuffling the relationship between triplets and the cells they came from, produces a graph that is only slightly different from the real data (right). This might appear to suggest that the relationship is random, i.e. that different triplets from the same neuron are not more likely to have the same classification than two triplets from different neurons. However, data collection is in progress for a more detailed examination of the effects within individual cells, and here we begin to see indications of consistency when the tested conditions are closer to each other. Below are 3 example neurons. Each is tested across a range of frequencies, paired with either a lower (420Hz) or higher (2000Hz) frequency. Only 2 locations are used (-12 or 12), and they are fixed within a session and counterbalanced across sessions. This gives us more power in the frequency domain.

The pattern that emerges is that “adjacent” conditions appear to be quite consistent, but the consistency declines with the difference in the frequency domain. In Example 1, the B=420 Hz (left panel) triplets were either uncategorized or were all categorized as Mixtures (data points labelled “M”). However, the B=2000 HZ (right panel) triplets were categorized as Singles (data points labelled “S”) or were uncategorized. In Example 2, there are Intermediates (“I”), Singles, and one Outside (“O”), but

they do not appear to be randomly organized. Instead, the Intermediates occur for a particular range of frequency values for sound A and sound B = 420Hz. In Example 3, all of the classifiable triplets were classed as "Singles".

To summarize, we can reassure the reviewer that a small number particular individual neurons did not provide all of the winning triplets in any particular classification. In our main data set, the pattern is not very different from chance, but we have reason to believe that a pattern is emerging in our more detailed investigation in the frequency domain in our followup study.

Example neuron 1

Example neuron 2

Example neuron 3

(3) I guess sound A could be presented from left or right speaker. However, IC neurons are sensitive to sound localization. Did Authors use only particular lateralized cases (e.g. sound A from only left and sound B from only right) or combine sound A from left and right? I am not sure if the ms. clarify this.

The pairs of locations we used were -24 and +6 and -6 and +24. Both pairs were used for all cells. A "triplet" involved one pair of locations; i.e. 742Hz at -24 and 500 Hz at 6 and the same frequencies at -6/24 are considered different triplets. Most neurons in the IC are quite responsive to sounds in the ipsilateral field provided the sound is in the neuron's frequency receptive field. Any triplets for which there was not a response to at least one of the two sounds was excluded by the "responses to A and B must differ" criterion.

The locations and frequencies are described in the Methods under the subheading "Data Sets and Auditory Stimuli: Locations, Frequencies, and Levels"

(4) What are the response thresholds relative to the sound levels of stimuli? Did all sampled units have monotonic rate-level (not azimuth but sound level) functions? A chief concern is that very small fraction of finding single units as the "summation-like" or the "outside" category, particularly in additive direction, may be simply due to the sound level relative to single units' threshold levels.

We did not conduct detailed rate-level sampling. In a portion of the data set ("Data Set I") we tested two slightly different levels (51 dB SPL and 55 dB SPL), so that we could compare "same-at-the-speaker" vs "same-at-the-ear" sound levels (i.e. two signals that produce 51 dB SPL sounds when presented alone produced a 55 dB SPL sound when combined). Regardless of the comparison evaluated, the majority of responses lay in the 95% confidence intervals for an "averaging" response (Supplementary Figure 3).

We do agree that the "outsides" might be really "singles" but are sensitive to the slightly louder dual sounds. Although this will be difficult to test formally, we direct the reviewer to our response to Reviewer 3 point #3. When we break down the results according to whether the neuron was responsive to both sounds in the triplet or only one, it appears that the majority of the "outsides" occurred in triplets that were responsive to both sound "A" and "B".

(5) Spectrograms of LFP. While DAPP-alpha functions go up and down slowly, why not checking whether LFP changed in the similar time scale? Thresholded test of spectrograms seem to show significant difference in that low frequency end, too. I think it is better to use wavelet transforms to reveal the frequency range similar to the dynamics of alpha in addition to high frequency gamma. Also, low frequency part may differentiate further between "intermediate flat" and "intermediate wavy," if it follows DAPP-alpha.

We agree the spectrograms are not designed to look in the frequency range of either the whole-trial or the within-trial switching. (See also response to Reviewer 1 "Line 288-290" regarding this question and note there is a correction to the stats in Fig 3D that produced a minor change in the appearance of this graph). As a ballpark estimate, one could imagine that the slower fluctuations (either whole or within-trial switching) might be evident in a roughly 5-20 Hz band. The plot below shows the energy in this band, which confirms that both the Intermediates and the Mixtures have energy in this band, but the

energy of the Intermediates is a little higher. As noted in the response to Reviewer 1, we are not quite at a point of drawing conclusions on the basis of the DAPP tags because we have fewer triplets to work with.

(6) What is the reaction time? Does it correlate with neuronal activity: e.g. more A-like responses give faster saccade to A? Is the first reaction time to dual sounds same as those to single sounds?

In general, the reaction time for dual sound trials is a little longer than for single sound trials (dual: mean 229.69 +/- 0.64 ms (SE); single mean 210.96 +/- 0.48 ms (SE)). However, the reaction times for “congruent” (A-like responses and first saccade to A or B-like response and first saccade to B) vs “incongruent” (A-like but first saccade to B or B-like but first saccade to A) did not differ from each other (congruent mean +/- SE: 254.54 +/- 2.78 ms, n=672 trials; incongruent: 261.86 +/-3.34, n=499 trials; two-tailed t-test, p=0.0891).

A bigger reaction time difference might have been found if we had used a reaction time task rather than requiring the monkeys to wait until the fixation light went off.

(7) Please make the statistics reporting consistent through the ms. Some p-val are not clear what statistics tests are used, and degree of freedom. Also, it is better to describe what statistics tests and decision criteria in Methods.

To clarify, we used two different types of statistics; we think the reviewer is referring to the instances where we used frequentist statistical tests which involve p values. We have reviewed each of these instances and added the missing information as appropriate. If the test involved a single hypothesis test, we report the actual p value and the number of degrees of freedom (e.g. Figure 4C and 4E); if the test involved multiple similar tests, we report only the threshold value that was adopted (e.g. Figure 3D, 4F).

For hypothesis testing with whole trial spike count data, we have adopted the Bayesian paradigm of statistical inference, and reported (posterior) probabilities rather than p-values. The Bayesian paradigm

require posterior probabilities of hypotheses to be calculated according to the the Bayes theorem, depending only upon prior probabilities of the same hypotheses and the likelihood score assigned to each hypothesis as given by the data model. We have provided what we hope are adequate details on these two inputs for the whole trial Poisson analysis under Data Analysis (Analyses of fluctuations in neural firing across and within-trials, and inclusion criteria)

(8) Line 91 and Suppl. Fig. 1. Are the “71% and 77%” based on a typical session or multiple sessions?

This is from all of the sessions in Data Set I; thank you for catching this omission. This detail has been added to the figure legend.

(9) Lines 114-117. According to the task training described in the Methods, animals were never encouraged to find a target in the middle. In other words, midpoint never was an option for monkeys to choose. Correct? If yes, then, the statements in the lines “... inconsistent with the behavioral results ... monkeys should respond to dual sounds as if ... midpoint,” may be overstating, and need an additional remark that animals did not have such option.

Related. Were animals presented with dual sounds both coming from midpoint, or even single sounds from midpoint? Such trials may serve as controls.

Although we did not encourage seeking a target in the middle, the pairs of targets were set up so that the “inner” target of one pair was between the two targets of the other pair. That is, monkeys had trials involving single sounds at +6 and -6. The +6 target was between the two targets used in the -6/+24 pair and the -6 target was between the -24/+6 pair. Thus, the possibility of looking at locations intermediate between the pairs of dual sounds was present in each session.

*In addition, we have indeed now tested one of the monkeys that participated in this study with two frequencies that actually were presented from the same location in space (the requested control). Under these conditions, the monkey made only one saccade – on the first day this monkey experienced it. Here are the results from a randomly selected subset of all the attempted trials involving two frequencies presented from a single location (either left, blue, or right, red).from this first day (graph repeated from Reviewer 1 point #1). We think this indicates that if this monkey *had* perceived the two sounds as coming from the midpoint, she would have made only one saccade.*

Monkey Y - Performance on first day with two frequencies presented from same location

Saccade to:

----- Target -12° 76% correct

----- Target 12° 89% correct

Shown are N=40 trials selected randomly from all attempted trials

We have also worded the statement in the manuscript more conservatively; it now reads “such apparent averaging response patterns appear inconsistent with the behavioral results.”

(10) Ordinate axes labels in Figure 2: “Probability (%)” I am not sure if this is legitimate expression, because probability is usually between 0 and 1, as described in the legend for Fig. 2F. Also, I guess, in Bayes analyses and alpha in DAPP models, probabilities are calculated to range from 0 to 1. So, the word “probability” is interchanging somehow. So, I suggest changing “12” to “0.12” for changing “Probability (%)” to “Fraction (%)” or “fraction of trials” just in case. Another suggestion could be to remove ticks for axes in Figs. 2A-D, similarly to Fig. 3A3, since those are rather schematic illustrations of distributions.

Good points - we agree and have removed the numbers on the tick marks for Fig 2A-D.

(11) Lines 209-211. When neurons were categorized as “Single,” did they actually exhibit significant responses to both sounds A and B? If not (i.e. neurons responded to only A), categorizing “single” may be due to stimuli selected for such neurons. Regardless of the answer is yes or no, this point is better clarified.

Yes, triplets categorized as “singles” often did have responses to both A and B alone. The “single” classification occurred about equally often among triplets that were responsive to only one sound vs. those responsive to both. This has been added as Supplementary Figure 5 and is referenced in the last paragraph in section 2.2 (in the vicinity of the lines the reviewer mentions here).

(12) Line 227. Left bracket at the beginning of the line is probably not needed. A period “.” is probably needed after “A and B” in the middle of the line.

Thank you! Fixed.

(13) Fig. 3B, alpha curves. Are those smoothed? I thought those should have steps every 50 ms.

The smoothness of alpha curves is a function of the model, and not an added feature for visualization. DAPP assumes the existence of smooth intensity curves, described by alpha. The effect of binning is formalized by assuming the spike count in a bin happens at the rate given by the intensity curve value at the bin mid-point, times the bin-width. Under our formalization, the data informs us only about the intensity value at the bin mid-points — the rest of the curve is interpolated according to the prior assumption of smoothness, encoded by the Gaussian process model. However, the interpolation (and resulting smoothing) of $\alpha_j(t)$ is controlled by the waviness parameter “ ℓ_j ” — which is also informed by data. We have added a sentence to the method section, “Dynamic Admixture Point Process Model”, paragraph 3, to clarify this. The new sentence is underlined:

The three parameters (ϕ , ψ , and ℓ) respectively encoded the long-term average value, the total swing magnitude and the waviness of the curve.—Intuitively, these parameters can be thought of as related to the means and variances of the distribution of weight values alpha regardless of time within a trial, as well as the correlation between the weight observed at one point in time and the weight observed at another on any given trial.

(14) Line 258. Should “distribution” be “distributions”?

Fixed, thank you!

(15) Line 263. Should “is not showed” be “is not shown”?

Fixed, thank you!

(16) Fig. 3D. average local field potentials. Are those spectrograms of averaged LFP or averages of spectrograms of LFP? Average of how many? How were spectrograms derived? FFT? Please describe what was done.

Thank you for catching these omissions! Fixed. Recapping part of a related response to Reviewer 1:

Details: *We have added a number of missing details to the figure legend and the methods section, specifically:*

Figure legend: "Data are from dual-sound trials from triplets that were classified as "mixture" or "intermediate" with a winning probability of >0.95.. The left and central panels show the average of single trial spectrograms, plotted as absolute power in a logarithmic scale. There were N=719 "mixture" and N=211 "intermediate" trials. The right panel shows a thresholded ($p < 0.001$)[JG1] [JG2] depiction of whether these differences were statistically significant according to a two-tailed t-test for each time point and frequency combination; red indicates the power for "intermediates" > "mixtures" for that time frequency combination, blue indicates the converse, and green indicates the p value was >0.001."

Methods section "Local Field Potential" paragraph 2:

For the spectral analysis presented in Figure 3D, single trial spectrograms were computed using Short-Time Fourier transforms (spectrogram.m in MatLab Signal Processing Toolbox; MathWorks Inc.) with 256 ms segment size and 250 ms overlap for each frequency bin (0.7-200 Hz with 3.9 Hz bin size). Each segment was windowed with a Hamming window.

Error correction: *In the process of revisiting this analysis, we discovered an error involving the squaring of the amplitude values for the t-test panel; this error has been corrected. The t-test became more sensitive, such that the p-value of 0.05 was too lenient for clear visualization of the pattern. We have adjusted this cutoff to 0.001 in the revised figure.*

(17) Fig. 4B. Do the distributions of xcorr coefficients differentiate after sorting pairs by paired types of categories like "wavy"- "flat-extreme", "wavy"- "wavy"? It would be interesting if positive.

This would indeed have been interesting. Unfortunately, the number of paired triplets is not sufficient for this sub-analysis. For example, there are no pairs of triplets where both were classified as "Intermediate", and only 4 where one of the two pairs was. That group of 4 does not appear to be particularly distinctive. We are working on setting up U-probe recordings that might permit us to record with a higher yield of paired units that should facilitate this analysis.

(18) Unit of LFP. Was it really milli-volt? Were signals recorded after amplification? What's "N=274" in Fig. 4C, the number of sites or triplets? Figure legend says 1902 and 1618 trials.

We thank the reviewer for catching that. The conversion between the value after the A/D converter and amplification and the original signal amplitude was not correct in figure 4C. However, since each individual LFP trace was amplified and converted in the same way, the correction amounts to a rescaling of the y-axis. Figure 4C now correctly reports the amplitude in microVolts.

N=274 is the number of triplets included in the analysis. After correcting some minor issues, the trial numbers changed slightly to 1917 A-like dual trials and 1565 B-like dual trials. These are indicated in the legend and come from the 274 triplets.

Line 355. What feature of LFP was analyzed, and corresponded to the "state of activity"?

We measure the average voltage of the local field potential without filtering it into any frequency bands. We have changed this section to read: "But the average LFP voltage also differed prior to sound onset (p -val = 0.0064), suggesting that the state of activity in the local network surrounding an individual neuron at the time of sound onset is predictive of whether the neuron "encodes" the contra-lateral or the ipsi-lateral sound on that particular trial. What exactly that network state consists of is unknown; but it appears to derive from some form of altered balance in the levels of activity in the ICs contralateral and ipsilateral to the first-saccade target."

Line 352 says "mean LFP value". Does it mean the mean magnitude of LFP during the 600 ms window? Since LFP has apparent dynamic change ranging positive and negative during that period in Fig. 4C, isn't it better to use some other parameters like peak amplitude? It would nice to see error bars or confidence intervals of LFP traces.

Yes, this analysis concerns the average voltage of the LFP during the 600 ms before vs. after sound onset. We could have certainly used peak amplitude but the thing that stands out visually is the consistent, sustained, difference in the baseline so we wanted to focus on that aspect. We have revised the figure to include error bars (mean \pm SE); see below.

See also response to Reviewer 1 under Line 336-349; that reviewer pointed out that filtering/amplification of could cause some bleeding backwards in time of the signals from after sound onset. We repeated the analysis from 600-200 ms before and after sound onset and obtained the same result.

Also, Figure 4C. In addition to units' firing during A-Like trials look like that of sound-A alone trials, do LFP of A-like trials look like LFP during sound A-alone trials?

We include below figure 4C and a similar graph showing the responses on the sound A-alone and B-alone trials. The LFP after sound onset shows generally similar results in the A-like vs actual A and B-like vs actual B - that is, the A-like and actual A involve more negative voltages and the B-like and actual B involve less negative voltages.

The astute reviewer may notice that the actual A and actual B LFPs differed very slightly from each other prior to sound onset ($p=0.0474$), but this disappeared when we included both correct and incorrect trials. This suggests that even on single-sound trials, slight difference in initial network state could contribute to performance -- similar effects have been observed by Newsome and colleagues involving motion perception.

(19) Fig. 4D. For this analysis, are there no difference when compared sound A-left and sound A-right?

With apologies, we are not sure we understand the question? Panel 4D displays the dual sound trial saccade traces for a sample triplet, with some saccades going to the lefthand target first and others to the righthand target first. All triplets have both a lefthand and a righthand target on the dual sound trials, and the designation "A" is arbitrary.

(20) What are the numbers superimposed on bar graphs in Fig. 2F and Fig. 5A? In Fig. 2F, the numbers seem the numbers of units, as they seem to correlate with the confidence level fractions within bars. But, the numbers in Fig. 5A seem cumulative of fractions.

*Thank you for catching this! This was indeed an error. We have corrected Figure 5 to match Figure 2. Both sets of numbers are now *not* cumulative, i.e. they report the number of triplets contained in each segment of the bars.*

(21) Line 694 in Sound localization task, "Figure 2A" is probably "Figure 1B."

Yes it is, thank you for catching this! Fixed.

(22) Line 841. Which section is the “Events of Task”?

We thank you for finding all of these errors - we really appreciate your attention to detail! This should have referred to “Sound localization task”. This has now been fixed.

Reviewer #3 (Remarks to the Author):

This manuscript presents evidence to suggest that neurons in the IC of the monkey multiplex information of about the locations of the sources of two, simultaneously active sounds. This is an interesting and novel idea with parallels to artificial communications systems. Supplemental data from the face area of the cortex are also reported.

The ms is well written and illustrated, and the analyses of spike trains via DAPP is rigorous. However, the acoustics are not treated rigorously, and there are a number of questions, control experiments, and comments that I would like to see addressed:

1. I would like to read a bit more about the need for multiplexing in the sound-localization task. The results show evidence of multiplexing in a condition where it is not necessary. Since the A and B stimuli differ by more than about a third of an octave, I can imagine that there will be activity in two, separate iso-frequency laminae. These separate patches of activity would differ in their firing rate assuming that source-location is rate-coded, so the two sources should be separately localizable with or without multiplexing.

Conversely, multiplexing would be needed if sounds A and B, presented from different loci, had the same amplitude spectrum (but uncorrelated fine structure). Was this condition tested?

These are excellent points. A previous version of the manuscript had included consideration of this issue and we had cut it for length. We are happy to put it back in!

Although it is commonly assumed that frequency maps in the auditory system are high enough resolution to solve this kind of problem, in fact there is reason to suspect that this may not be the case. When all units are tested at a modest sound level similar to the level we used, a large proportion of units are responsive to tones. In the frequency range that we tested (shaded in gray below), between 40-80% of IC multiunits have been shown to be responsive to tones (below is Figure 6 from Bulkin and Groh, 2011 Journal of Neurophysiology). The current study involves single units, which might reduce these numbers somewhat, but also involves bandpass stimuli (bandwidth 400 Hz), which should increase them. Indeed, in our dataset, about $\frac{2}{3}$ of the included triplets were responsive to both of the sounds that were presented (see response to point #3 below).

From Bulkin and Groh, 2011, *Journal of Neurophysiology*, Figure 6

We are presently working on a followup study to see if there is sharpening of the resolution of the frequency map (e.g. via lateral inhibition) that might partially contribute to solving the dual sound localization problem. Some examples from this study are shown above (Reviewer 2 point #2).

We have added the following paragraph to the Discussion (Paragraph #11, approximate line #556):

“The need for multiplexing extends to any domain where neural tuning is broad. The meter/firing rate code for sound location in the IC is one example. The IC’s sensitivity to sound frequency can only partially ameliorate this problem, because here too the tuning is very broad: a pure tone in the frequency range employed in this study has been shown to evoke activity in 40-80% of IC neurons⁴². Indeed, about 2/3 of the triplets in the present sample responded to both of the two sounds individually (Supplementary Figure 5). Most natural sounds are spectrally rich and will activate “hills” of neural activity with even greater overlap. That said, it is possible that the resolution of the frequency map might sharpen via lateral inhibition when more than one sound is present; such a mechanism might work in concert with time division multiplexing.”

With regard to the last point, testing of A and B sounds with the same amplitude spectrum but uncorrelated fine structure, we have only tested bandpass sounds of different center frequencies. Perceptually, when two white noise stimuli (different waveforms but both white noise), they appear to our ears to “twinkle” between the two locations, giving a rather interesting percept of a single auditory object moving or having a size spanning the two positions.

2. If the multiplexing enables the representation of the loci of the two sound sources, DAPP should show no evidence of multiplexing when sounds A and B are presented from the same location. Is this the case?

We haven’t tested this in the IC, but a follow up study explicitly tests this hypothesis in the SC with combined visual and auditory stimuli. We do see evidence for whole-trial fluctuations - a preponderance of “Mixtures” in the whole-trial analysis. We find them even when the two stimuli are at

the same location in space (shown below). It is possible a similar result would occur in the IC, which would suggest that both frequencies are retained as segregable items. In the SC, we are continuing to investigate whether there are more or more obvious fluctuations when the two stimuli are presented at different locations in space. Initial results do support the reviewers intuition of greater fluctuations with separated targets. Currently we don't have sufficient data to address this question using the DAPP but we anticipate doing so eventually.

3. What is the difference between the center frequencies of sounds A and B relative to a cell's frequency tuning curve? If the center frequency of one of the sounds is outside a cell's tuning curve, the cell should not fire to it, and the multiplexing phenomenon should not be observed. It would be helpful to see the frequency tuning curves of the neurons shown in Figures 1E and F and some description of the breadths of the frequency tuning curves in the neurons sampled and their relationship to stimulus A and B.

We have added two additional supplementary figures in response to this comment; see also discussion above under Rev 3 point #1 regarding the granularity of the IC's frequency map. Rev 2 point #2 also shows several additional frequency tuning curves.

New Supplementary Figure 4 presents the PSTHs, color coded by frequency, for each of the example cells presented in Figures 1E and 1F (same as those displayed in Figure 2E1 and 2E3 and Figure 3B).

New Supplementary Figure 5 (below) presents the whole trial analysis broken down by whether the triplet involved was responsive to both sounds individually or only to one of the two sounds. The results were similar in the two subpopulations, except that the “Outsides” tended to occur in the both-responsive subpopulation.

In short, multiplexing *does* occur even when one of the two sounds is outside the “receptive field”. This may indicate that the granularity of control of the phenomenon is not at the scale of individual neurons but at the scale of a neural ensemble.

4. I understand the rationale for integrating spike counts in 50-ms windows - the DAPP model assumes that source-locations are encoded in the number of spikes. However, perhaps a more direct test of the multiplexing hypothesis would be to examine the neuronal responses to the envelope of the frozen stimuli. Neurons in the IC lock their discharge to the envelope of the stimulus. Since the narrowband noises are frozen, their repeated presentation should reveal a PSTH that is typical of A or B. In the responses to dual-sound stimuli, the PSTH patterns should alternate between the A PSTH and the B PSTH if multiplexing occurs.

The DAPP model builds in any phaselocking in the response to the envelope of the stimulus or any other consistent temporal properties in the instantaneous firing rate across time (such as a transient followed by a sustained response). It then evaluates whether the responses on AB trials fluctuate between the A or B level that is characteristic of a particular 50 ms time bin. The examples in Figure 1E and F show this with respect to the PSTHs. The example in Figure 1F3 in particular had switching moments that had sufficiently similar timing across trials that it is possible to see the individual trials changing from “A”-like to “B”-like across time.

We have also tried 25 ms bins. Here, the tradeoff is there are more bins but each one is more variable. The results with these smaller 25 ms bins were generally consistent with the 50 ms bins. Possibly the reviewer is suggesting testing smaller bins but we think this would not be productive due to variability/likelihood of empty bins at smaller bin sizes.

5. If the spike trains of IC neurons alternately represent sources A and B, does this mean that while the neuron is firing at a rate that encodes source A's location, is information about B's location momentarily lost?

Conceivably yes, but only if the entire population fluctuates together perfectly and there is no other population that serves as a memory store. The model in Figure 6 includes a memory storage element that could "bridge" across fluctuations; it is possible that the whole-trial fluctuating neurons occur at a memory-informed stage of the representation.

Reviewers' comments:

Reviewer #1 (Remarks to the Author):

The authors provided for the majority of questions satisfactory responses and modifications in their rebuttal letter and manuscript. Unfortunately there are still open issues with respect to the three main topics of concern. Subsequently I will try to describe as clear as possible the conceptual problems which appear unresolved to me.

With respect to the first main concern the authors clearly improved the manuscript by including descriptions of relevant context from the field of selective attention. However, I have not found an argument ruling out the possibility that the experimental results, which they describe as evidence for time division multiplexing, can be explained consistently as the consequence of selective attention switching occasionally and randomly between the two stimuli since the task does not control when and to which stimulus attention is allocated. Also, the new data described in the letter do not change this situation. According to the limited information available, there appear to be two distinct stimuli (one visual and the other auditory) at the same place in space. Nevertheless, they evoke different activity levels, indicating that the recorded neurons do not only encode stimulus position. The observed evidence for fluctuations between encoding the visual and the auditory stimulus is therefore again fully compatible with the assumption that selective attention switches occasionally between the two stimuli. Thus, it is not clear why the observed switching of the neuronal state between firing rates representing stimulus A and rates representing stimulus B should be associated with the concept of a (very slow) time division multiplexing if it fits perfectly to the well-established concept of selective attention. To establish evidence for a new mechanism, it would be necessary to demonstrate (or find convincing arguments) that the experimental observations cannot be explained by selective attention.

With respect to the second main concern, the authors try to weaken the relation between their concept of time division multiplexing and what is meant by time division multiplexing in telecommunication systems by stating in the discussion that "multiplexing is unlikely to occur exactly the same way in biological systems as it does in technological ones". In addition, they mention possible differences like more fluid time courses and asynchronous operation of different neural ensembles. On the other hand, the introduction still suggests, that their concept of time division multiplexing is still determined by preserving "information about multiple simultaneous items". If the only remaining "shared element of the analogy with telecommunications is the transmission of more than one item or type of information via signal fluctuations at some time scale", the concept becomes very unspecific and difficult to grasp. On the conceptual level it therefore necessary to clarify what exactly is meant by time division multiplexing, what it is good for in terms of information processing, how it works and how it differs from selective attention or other well-known processes in which neuronal populations change the stimulus or other content which they represent. If the functional difference between the authors' concept of time division multiplexing and such known processes cannot be clarified, also the broad implications claimed in the discussion for the fluctuations cannot be maintained. Any process (like selective attention) that causes changes in the represented content explains a high variability of neural firing patterns (line 491-503), is a reason for processing limitations (line 504-513) and may explain some kind of oscillation (line 514-522).

Furthermore, the core problem to reconcile time division multiplexing for quasi concurrent processing or transmission with a major part of trials, that show not even one switch between the two stimuli throughout the whole trial remains unresolved and is apparently not touched by the authors' response. Such a lack of switching for an entire behavioral trial appears incompatible with time division multiplexing but easily fits with an explanation based on selective attention. Thus, it is not clear how the now modified concept of time division multiplexing would differ from selective attention

switching between simultaneously present stimuli. The shown experimental evidence is in fact a hallmark feature of selective attention: the same set of broadly tuned neurons can change their activity levels successively to engage in the representation of different, simultaneously present stimuli.

With respect to the third main concern and the request concerning lines 288-290 in the initial manuscript, the authors seem to agree, that there is no evidence available for power changes in the LFP that would fit to the very low switching frequencies, suggested by their results on spike rates. Their assumption that "the evidence for oscillatory activity at higher frequencies in the LFP indicates that there may be switching occurring at higher frequencies than is possible to ascertain with our current (spiking) data set" might be correct, but there is apparently no experimental evidence available which links these higher frequency oscillations to a switching of rates as described by their results for very slow switching rates. If this is true, the suggested interpretation would be rather speculative and likely it would strengthen the manuscript to skip it.

With respect to flat/central vs. wavy/central vs. flat/extreme, the authors state in their letter: "In general, the results suggest that flat/central show more energy at higher frequencies than do either wavy/central or flat/extremes. We tentatively interpret this as suggesting that the wavy/central and flat/extremes are actually fairly similar to each other." This tentative interpretation and the observed power suggest a clear difference between flat/central and wavy/central trials. Nevertheless, they are still taken together into one group for the analysis described in the manuscript. Given the clear differences, this can hardly be justified by the loss of statistical due to the smaller number of triplets available for the separate groups. The headline 2.4 ("Local field potential shows greater oscillatory energy associated with "intermediate classification") appears therefore also misleading since the claimed effect is apparently only carried by the flat/central trials. In summary, I do not see how the analysis of the LFP power provides convincing support to the concept of time division multiplexing.

Reviewer #2 (Remarks to the Author):

I think explanations are fair enough, and the revised manuscript looks good.

Reviewer #3 (Remarks to the Author):

I have re-reviewed the ms, and have found that the authors have adequately addressed my concerns. Overall, my comments had to do with the acoustics of the two stimuli and the conditions under which multiplexing may or may not be necessary. The questions posed in the initial review came from a theoretical perspective, but the answers cited empirical data (e.g., "2/3 of the included triplets were responsive to both of the sounds that were presented."). It is difficult to argue against empirical observations. In a similar vein, the authors have now included the frequency tuning curves of some cells (Supplemental Figures) that address the same point.

The authors also corrected my misunderstanding of the 50 ms integration time-window. I was initially under impression that amplitude modulations will be lost, but the authors clarified that envelope phase locking or onset bursts were taken into account.

Finally, the authors considered my point that certain portions of the two signals can be lost as cells respond to one signal and not the other. Their model incorporates a memory storage mechanism that could retain the information regarding the portion of the signal to which cells were not responding at a given time.

Overall, I am satisfied with the authors' answers and appreciate the detail with which they addressed my questions.

Overview:

Reviewers 2 and 3 found the manuscript satisfactorily revised, whereas reviewer 1 had one remaining conceptual concern and two suggestions for improvement. The conceptual issue concerns the relationship between our study and previous work on selective attention. We cover this issue below in detail, including both consideration of attention studies in other brain areas and new analyses of the monkeys' behavior in our study. We are confident that our findings are distinct from those in the attention literature and we hope the reviewer will agree. We also adopted the reviewer's suggestions for improvement: we removed the spectral LFP analysis and improved the foreshadowing of the complexities of the time division multiplexing hypothesis in the introduction.

Reviewers' comments:

Reviewer #1 (Remarks to the Author):

The authors provided for the majority of questions satisfactory responses and modifications in their rebuttal letter and manuscript. Unfortunately there are still open issues with respect to the three main topics of concern. Subsequently I will try to describe as clear as possible the conceptual problems which appear unresolved to me.

With respect to the first main concern the authors clearly improved the manuscript by including descriptions of relevant context from the field of selective attention. However, I have not found an argument ruling out the possibility that the experimental results, which they describe as evidence for time division multiplexing, can be explained consistently as the consequence of selective attention switching occasionally and randomly between the two stimuli since the task does not control when and to which stimulus attention is allocated. Also, the new data described in the letter do not change this situation. According to the limited information available, there appear to be two distinct stimuli (one visual and the other auditory) at the same place in space. Nevertheless, they evoke different activity levels, indicating that the recorded neurons do not only encode stimulus position. The observed evidence for fluctuations between encoding the visual and the auditory stimulus is therefore again fully compatible with the assumption that selective attention switches occasionally between the two stimuli. Thus, it is not clear why the observed switching of the neuronal state between firing rates representing stimulus A and rates representing stimulus B should be associated with the concept of a (very slow) time division multiplexing if it fits perfectly to the well-established concept of selective attention. To establish evidence for a new mechanism, it would be necessary to demonstrate (or find convincing arguments) that the experimental observations cannot be explained by selective attention.

General response:

There are two parts to the reviewer's concern: a) does the animal detect only one stimulus during the analysis period (prior to commencing the saccade sequence); and b) if so, does that render the observed neural fluctuations redundant with previously demonstrated phenomena from the attention literature?

We believe the answer is "no" on both counts.

1. Does the animal detect only one stimulus during the analysis period?

No: the reaction times and saccade trajectories strongly indicate that the saccades are generally programmed as a planned sequence, because the interval between the first and second saccades is substantially shorter than is typically observed in reaction time paradigms with only a single saccade target. Specifically, the intersaccade interval was a median of 112 ms in a sample of 5 sessions. This compares to a mean reaction time to initiate the first saccade of about 210 ms on single sound trials. In short, the second saccade occurs a good 100 ms faster than it should if the animal was waiting to get to the first target, then sensing the second sound and planning the second saccade.

In addition, we frequently observed curved saccades on dual sound trials. Curved saccades are nearly non-existent when saccades are directed to a single target, but have been observed occasionally when saccades to two locations are programmed together (e.g. Groh and Sparks, 1996, *J. Neurophys*, "Saccades to somatosensory targets. I. Behavioral characteristics"). The figure below includes a sampling of a few saccade traces, with eye position plotted every 2 ms. Most of the illustrated saccades go first towards the target on the right and stop (the dot spacing gets very narrow), and then go toward the target on the left and stop. On some trials (blue dots), the eyes don't stop at the first target but curve leftward and stop at the second target.

Figure. Sample eye trajectories on a selected set of dual sound trials from one session. Eye position is plotted every 2 ms. The curved saccades (blue) were unrewarded because they did not pause in the reinforcement window for the target on the right.

Together, we think these observations strongly suggest the monkey is processing both stimuli during the analysis interval, resulting in a short latency between saccades.

Furthermore, we believe our task has advantages over classical selective attention paradigms. Sequentially looking to each stimulus deploys a naturally occurring behavioral pattern to demonstrate that both stimuli are detected and localized. We are not aware of comparable studies in the single unit attention literature in which animals were tasked with responding to more than one simultaneously presented target.

2. If the animal were processing only one stimulus during the analysis period, would that make the observed neural fluctuations redundant with previously demonstrated phenomena from the attention literature?

No. We believe the reviewer to be mistaken about what has previously been demonstrated in the literature at the single unit level with regard to attention. In particular, we can restate our finding as saying that there are periods of time in some neurons in which the response to two sounds is indistinguishable from the response to one or the other alone, and that which of the two sounds the response patterns match varies across time. We do not think the attention literature has previously demonstrated cases in which the response pattern to attended stimuli matches the response to that stimulus when presented alone.

To be specific, the commonly performed visual attention experiment that is relevant here involves presenting two stimuli (at different locations, say) and comparing how the neurons respond when stimulus A is attended vs. when stimulus B is attended. A difference in response in the attended vs. unattended condition is considered to be evidence of selective attention acting on the neural representation. However, the studies we reviewed did not demonstrate that when A was attended, the response was akin to when A was the only stimulus presented, or vice versa for B.

To put some made up but plausible numbers on this, suppose A alone evokes 20 spikes and B alone evokes 40 spikes. When both A and B are presented and the animal attends A, perhaps 28 spikes are evoked – on the “A” side of the mean (30 spikes), but not the same as the number of spikes observed when A was actually the only stimulus. A finding of 28 spikes would be interpreted as suggesting a biasing in a competition of the representations of A and B, but it is quite different from observing a response of 20 spikes in the A-attended condition and 40 spikes in the B-attended condition.

A second issue is that our study concerned evaluating distributions of responses, and not just means. The single unit attention literature largely concerns means, pooled across time and trials. It is easy to calculate means from distributions, but it is not possible to go the other way. Consider:

I. Our experiment:

II. Typical single unit attention experiment:

The top panel is our Figure 2A, and illustrates the way in which we considered the distribution of spike counts across trials in the whole-trial analysis. The most common outcome on dual-sound AB trials was an overdistributed, bimodal-like distribution of responses (black line) that could be modeled as deriving from a mixture of two component distributions based on the single sound trials A and B.

The bottom panel depicts the values that are usually reported for individual units or populations of them in single-unit attention studies: a comparison of the mean responses to the attend-A vs. attend-B conditions (two purple arrows). Many studies do not even measure the responses to A or B alone (blue and red arrows), although we were able to find one study that did (Reynolds et al. 1999).

The top panel can be used to predict the values in the bottom panel, but the converse is not the case.¹

¹ The degeneracy of the relationship between distributions and summary values is visualized with an amusing gif here: <https://www.autodeskresearch.com/publications/samestats>.

In fact, one of the important contributions of our paper is that the new statistical analyses we present can be deployed on data collected in attention experiments to facilitate evaluation of which stimuli are encoded and when.

More broadly, we again point out that we have had to go quite far afield from our own work to seek out even conceptually relevant literature: all of the selective attention work described above has been conducted in visual brain regions (e.g. V4), not in auditory ones. It can never be assumed that what holds true in one sensory modality (or brain area) will hold true in another.

The reviewer sums up: “To establish evidence for a new mechanism, it would be necessary to demonstrate (or find convincing arguments) that the experimental observations cannot be explained by selective attention.”

To sum up our response:, we studied new brain areas using a new task that requires preservation of multiple items, and evaluated the distribution rather than time-and-trial-pooled mean activity values. The selective attention literature has not previously demonstrated the phenomena we explore here.

Superior Colliculus data:

With regard to the SC data we included in our response, the reviewer states “they evoke different activity levels, indicating that the recorded neurons do not only encode stimulus position. The observed evidence for fluctuations between encoding the visual and the auditory stimulus is therefore again fully compatible with the assumption that selective attention switches occasionally between the two stimuli.”

The point of the SC data included in the previous response to reviewers was to show that fluctuations occur even when spatial attention is directed to one location. This analysis was made possible by the fact that visual and auditory stimuli do not evoke the same levels of activity in individual SC neurons, even when presented at the same location in space, as noted by the reviewer. (Barry Stein’s group has demonstrated this extensively. Our lab has shown that in general the sensory responses to auditory stimuli are weaker than visual responses although the discrepancy is reduced for more eccentric stimuli; Lee and Groh, PLOS One, 2014). Furthermore, the monkey’s behavior is clearly consistent with attention to a single location as only one saccade is made. Thus, this data provides an example of fluctuations occurring despite spatial attention being directed to a single focus point.

With respect to the second main concern, the authors try to weaken the relation between their concept of time division multiplexing and what is meant by time division multiplexing in telecommunication systems by stating in the discussion that “multiplexing is unlikely to occur exactly the same way in biological systems as it does in technological ones”. In addition, they mention possible differences like more fluid time courses and asynchronous operation of different neural ensembles. On the other hand, the introduction still suggests, that their concept of time division multiplexing is still determined by preserving “information about multiple simultaneous items”. If the only remaining “shared element of the analogy with telecommunications is the transmission of more than one item or type of information via signal fluctuations at some time scale”, the concept becomes very unspecific and difficult to grasp.

We think the discussion and introduction are consistent with each other but we understand the reviewer's point that foreshadowing the caveats presented in the discussion at an earlier point would be valuable. Accordingly, we have added a new sentence mentioning details such as time scale to the end of the introduction, after the reader has learned about sufficient details of our findings to have an inkling of why we are saying these things.

To recap, where we first define time division multiplexing, we say:

"We investigated whether the brain solves this problem at the neuronal level using time division multiplexing, i.e. via signals fluctuating between the activity patterns evoked by each stimulus alone (Figure 1A). Such time division multiplexing is commonly used in telecommunications to permit encoding of multiple items in a single channel across time. Similar strategies have been postulated to occur in some form in the brain ^{1, 2, 3, 4, 5, 6, 7, 8, 9, 10, 11}, but empirical evidence and statistical methods of assessment are presently lacking."

We think this paragraph does about as good a job as possible for its position at the beginning of the introduction. It briefly introduces the term, making reference to its engineering history, while quickly re-defining it for a neuroscience audience ("via signals fluctuating...") and cites the relevant neuroscience literature. For readers with an engineering background, the term should evoke the general concept while alerting them that the instantiation may be different ("similar strategies... postulated...in some form in the brain"). For neuroscience readers who are unfamiliar with the engineering concept, they are not likely to be misled by a meaning they are unaware of.

This paragraph is followed by a brief summary of our experiment and findings. Such details set the stage for a new sentence in the last paragraph of the introduction that foreshadows the issues that receive full treatment in the Discussion: "We note that the brain's version(s) of time division multiplexing may differ from the telecommunications analog, especially with regard to such factors as time scale and coordination across ensembles of neurons."

On the conceptual level it therefore necessary to clarify what exactly is meant by time division multiplexing, what it is good for in terms of information processing, how it works and how it differs from selective attention or other well-known processes in which neuronal populations change the stimulus or other content which they represent. If the functional difference between the authors' concept of time division multiplexing and such known processes cannot be clarified, also the broad implications claimed in the discussion for the fluctuations cannot be maintained. Any process (like selective attention) that causes changes in the represented content explains a high variability of neural firing patterns (line 491-503), is a reason for processing limitations (line 504-513) and may explain some kind of oscillation (line 514-522).

Time division multiplexing presents a theory of how multiple items can be preserved in neural representations, whereas selective attention focuses on how the rich sensory scene is winnowed down to ~1 item to be processed. As noted above, the attention literature has not described fluctuations at the single unit level; thus the "known processes" mentioned above have not actually been demonstrated.

We think our presentation of the theory of time division multiplexing, what purpose it serves, and its relationship to selective attention is as clear as we can presently make it. We realize the paper may

engender debate in the field and look forward to writing subsequent papers on the topic to further refine the ideas in keeping with any emerging empirical findings.

Furthermore, the core problem to reconcile time division multiplexing for quasi concurrent processing or transmission with a major part of trials, that show not even one switch between the two stimuli throughout the whole trial remains unresolved and is apparently not touched by the authors' response. Such a lack of switching for an entire behavioral trial appears incompatible with time division multiplexing but easily fits with an explanation based on selective attention. Thus, it is not clear how the now modified concept of time division multiplexing would differ from selective attention switching between simultaneously present stimuli. The shown experimental evidence is in fact a hallmark feature of selective attention: the same set of broadly tuned neurons can change their activity levels successively to engage in the representation of different, simultaneously present stimuli.

A key feature of attention is that it is thought to work similarly in the same broad set of neurons – i.e. all the neurons should show the same type of pattern, consistent with all the neurons representing the same stimulus at once. Here, we find considerable heterogeneity. Pairs of neurons recorded at the same time are in some cases classified differently (i.e. one is a “mixture” while the other is an “intermediate” on the same set of trials). The same neurons are also sometimes classified differently for different sets of stimulus conditions (see detailed analysis in previous response to reviewers). This heterogeneity of response patterns is not predicted by theories of selective attention.

That said, the whole-trial fluctuations were a surprise to us too. They are however a major aspect of the observed results. We consider it possible that they represent either an output stage of time division multiplexing, and this is stated in the manuscript in connection with the models presented in Figure 6 (Discussion, last paragraph on page 24, “It is possible that within-trial fluctuating units lie at the input stage of such a circuit, and that between-trial fluctuating units actually lie at the output stage.”. (We originally included this comment in an earlier draft of our response to reviewers but took it out in an attempt to shorten what ended up as 30+ pages of single-spaced response).

Regardless of what one thinks about time division multiplexing as a theory, the observation that there are mixed distributions of responses when combined stimuli are presented stands on its own. As noted previously, we agree that in general our findings have implications for theories of selective attention, and that our methods can be deployed in that domain, but disagree strongly that this phenomenon has been reported in the attention literature before.

With respect to the third main concern and the request concerning lines 288-290 in the initial manuscript, the authors seem to agree, that there is no evidence available for power changes in the LFP that would fit to the very low switching frequencies, suggested by their results on spike rates. Their assumption that “the evidence for oscillatory activity at higher frequencies in the LFP indicates that there may be switching occurring at higher frequencies than is possible to ascertain with our current (spiking) data set” might be correct, but there is apparently no experimental evidence available which links these higher frequency oscillations to a switching of rates as described by their results for very slow switching rates. If this is true, the suggested interpretation would be rather speculative and likely it would strengthen the manuscript to skip it.

With respect to flat/central vs. wavy/central vs. flat/extreme, the authors state in their letter: “In general, the results suggest that flat/central show more energy at higher frequencies than do either wavy/central or flat/extremes. We tentatively interpret this as suggesting that the wavy/central and

flat/extremes are actually fairly similar to each other.” This tentative interpretation and the observed power suggest a clear difference between flat/central and wavy/central trials. Nevertheless, they are still taken together into one group for the analysis described in the manuscript. Given the clear differences, this can hardly be justified by the loss of statistical due to the smaller number of triplets available for the separate groups. The headline 2.4 (“Local field potential shows greater oscillatory energy associated with “intermediate classification”) appears therefore also misleading since the claimed effect is apparently only carried by the flat/central trials. In summary, I do not see how the analysis of the LFP power provides convincing support to the concept of time division multiplexing.

We have removed the LFP power analysis and the associated text.

Reviewer #2 (Remarks to the Author):

I think explanations are fair enough, and the revised manuscript looks good.

Thank you! We thank you again for your comments.

Reviewer #3 (Remarks to the Author):

I have re-reviewed the ms, and have found that the authors have adequately addressed my concerns. Overall, my comments had to do with the acoustics of the two stimuli and the conditions under which multiplexing may or may not be necessary. The questions posed in the initial review came from a theoretical perspective, but the answers cited empirical data (e.g., “2/3 of the included triplets were responsive to both of the sounds that were presented.”). It is difficult to argue against empirical observations. In a similar vein, the authors have now included the frequency tuning curves of some cells (Supplemental Figures) that address the same point.

The authors also corrected my misunderstanding of the 50 ms integration time-window. I was initially under impression that amplitude modulations will be lost, but the authors clarified that envelope phase locking or onset bursts were taken into account.

Finally, the authors considered my point that certain portions of the two signals can be lost as cells respond to one signal and not the other. Their model incorporates a memory storage mechanism that could retain the information regarding the portion of the signal to which cells were not responding at a given time.

Overall, I am satisfied with the authors’ answers and appreciate the detail with which they addressed my questions.

Thank you again for your thoughtful evaluation and critique.

** See Nature Research’s author and referees’ website at www.nature.com/authors for information about policies, services and author benefits

REVIEWERS' COMMENTS:

Reviewer #1 (Remarks to the Author):

Revision 2

The authors provided for one of the three major issues a good solution. Two others are still open. Within the manuscript they removed the text related to the resolved issue. Apart from that there is not much more than an added sentence in the introduction. Therefore I will describe the remaining problems, mainly along the rebuttal letter. In the end I sum up my view on the current state of the manuscript.

Rebuttal letter, page 1-3, First main concern

General Response, 1. and Lines 94-95 and 119-120 of manuscript

The authors' response appears to be related to one of several critical statements of the previous (first) review which stated: "(3) The task does not strictly require keeping representations of both stimuli active. If the animal recognizes that more than one stimulus is present, it would be sufficient to attend one of the two stimuli to identify its location. This location predicts the second location (a stimulus at the position close to the midline is always associated with a second stimulus at the position more distant from the midline at the other side of the midline and vice versa). Generating a motor plan for the two saccades therefore does not need to localize the position of both stimuli. Such an interpretation would be in line with the large part of trials in which no within-trial switching is observed but a constant rate fitting the expectation for one of the stimuli throughout the trial."

I agree with the authors that a very short intersaccade interval and curved saccades "indicate that the saccades are generally programmed as a planned sequence" but for the reasons given previously and above these data do not "suggest the monkey is processing both stimuli during the analysis interval". Processing one of them is entirely sufficient since the second position is predictable from the first perceived. This is in line with the large part of trials in which no within-trial switching is observed.

Rebuttal letter, page 3-5, General Response, 2.

The second question, whether the possibility that "the animal detect only one stimulus during the analysis period" renders the observed neural fluctuations redundant with previously demonstrated phenomena from the attention literature is difficult to relate to my requests. The possibility that the animals do not process both stimuli because the task does not require this is one of several problems detailed in the first review. Neither the first nor the second review stated, that solely this individual issue "renders the observed neural fluctuations redundant with previously demonstrated phenomena from the attention literature". The real point is that the entire set of observation appears entirely compatible with an explanation based on selective attention fluctuating sometimes slower (or not at all within the analysis period) and sometimes faster between the two stimuli.

An entirely different issue is the authors' claim, that their data selected during the presentation of both stimuli matches better with the corresponding single stimulus data. This may have several reasons, like the difference between IC and visual cortical areas or the fact that the authors used a highly advanced data analysis procedure which strongly selects the data for such a matching whereas in the classical "biased competition" paradigm all correctly performed trials in which the animal was instructed to attend a stimulus are simply averaged, certainly containing many episodes in which attention fluctuated temporarily away from the instructed target. However, it does not rule out, that the observed fluctuations between stimulus-A and stimulus-B-like responses reflect fluctuations of attention. The correspondence between the neuronal response patterns and the first saccade target is good evidence for this assumption (see first review). Similarly, other methodological differences and interesting advantages of the method developed to analyze the data, the methods potential usefulness in attention experiments or general differences between auditory and visual brain regions do not resolve this interpretational problem.

The authors finally state "To sum up our response:, we studied new brain areas using a new task that requires preservation of multiple items, and evaluated the distribution rather than time-and-trial-pooled mean activity values. The selective attention literature has not previously demonstrated the phenomena we explore here." While the first three claims (new brain areas, new task, and evaluation of the distribution) hold true, the claim that the attention literature would not have previously demonstrated the phenomena which the authors explore misses again the point which has been detailed several times.

(While in the attention literature usually depends on a task that directs attention to one of several stimuli, the so called divided attention tasks are closer to the authors' task since they require the subject to attend to more than one stimulus. See the very recent work by Ricardo Kienitz et al (bioRxiv 252130; doi: <https://doi.org/10.1101/252130>) which provides an example (and cites corresponding literature) of a divided attention task in which spontaneous activity fluctuations in the theta-range go along with attention switching between the stimuli.)

Rebuttal letter, page 5, Superior Colliculus data

I agree with the authors that the direction of spatial attention may not have changed under the conditions in this experiment. Furthermore the authors confirm my understanding of the results that the response strength for visual and auditory stimulus differ and that despite of lacking motivation to change the direction of spatial attention fluctuations between the prototypical response levels of the two different stimuli are observed. I further agree that spatial selective attention is therefore an unlikely source for the fluctuations. However, there is a considerable literature which has demonstrated object based and feature based attention including corresponding changes in activity. The results are therefore still compatible with the likely assumption that attention has occasionally switched between the two different objects.

In summary: The central issue remains that the observed fluctuations are fully compatible with the fluctuations of attention expected for a task which does not control which stimulus is attended at which period of time. None of the arguments provided the required evidence that would exclude this straight forward explanation.

Rebuttal letter, page 6, Second main concern

With respect to this issue the manuscript further emphasizes the possibility of differences between the technical meaning of time division multiplexing and their usage of the term. Since no precise deviations from the common understanding of the term are described, the concept does not become better defined. It rather loses more of the straight forward meaning which it has in its common usage.

Thus, the problem indicated in the first review remains: Time division multiplexing suggests that a certain resource for data transmission (typically a transmission line) is switched between multiple data sources with a frequency high enough to have the information of all sources available within a time span that allows a subsequent stage to assess the information of all sources as if they would be simultaneously conveyed by multiple, dedicated lines. At least the aspect of simultaneity is also pointed out in the rebuttal letter (Time division multiplexing presents a theory of how multiple items can be preserved in neural representations, ...).

This concept contrasts with the authors' experimental observation which shows for many trials that there is only one of the stimuli processed. The switching between processing the two different stimuli is therefore extremely slow (contrary to the suggestion in line 313 of the manuscript) and does not support a quasi-simultaneous presence of information at subsequent stages. The experimental evidence supports in contrast selective processing which occasionally switches between stimuli (or other content) and which is typically a consequence of switching attention. If one would extend the usual meaning of the term time division multiplexing as suggested by the authors, almost all

observations of selective or flexible processing (as in the attention literature, but also in the motor system) would be examples of time division multiplexing. At the same time the term would become almost meaningless and would lose its current meaning. Since there is no lack of established terminology for the experimental observations, such redefinitions and degradations of established terminology should be avoided.

Rebuttal letter, bottom of page 6:

The problem is not whether the attention-literature demonstrated fluctuations on the single unit level for at least two reasons: 1. There is strong evidence that such fluctuations exist (See literature cited in Kienitz et al. and the literature on theta frequency switching between stimuli in divided attention tasks). 2. The task does not control the direction of attention to the two stimuli and therefore fluctuations of attention are very likely and – more important – can not be ruled out. Single unit spontaneous fluctuations have been shown by Logothetis and Coworkers also for another “well-known process” which is binocular rivalry. The problem is that the experimental results appear very well compatible with fluctuations caused by attention while features like a reasonable switching frequency as expected for time division multiplexing serving preservation of representations of multiple items are not observed.

Rebuttal letter, middle of page 7:

It is not true that attention “is thought to work similarly in the same broad set of neurons – i.e. all the neurons should show the same type of pattern, consistent with all the neurons representing the same stimulus at once.” No matter whether spatial attention, feature based or object based attention is considered. There will be almost always cells which are enhanced, reduced or unaffected in their activity level or synchronization patterns, depending on the relation between their RF-properties, stimulus constellation and attention. See for example the literature on normalization models for attention.

Where in the manuscript is the evidence for simultaneously recorded neurons of different types? I did not find a description detailing how many cells of which type have been recorded simultaneously. Part of the intermediate cases (those of the “flat central” type) are conceptually reconcilable “flat extreme” type of the “mixture” cases.

A similar kind of speculative explanation of the different types of cells could be suggested for an interpretation in terms of fluctuating attention: Whole trial fluctuations would characterize an output stage that would try to keep the information generated sequentially by the set of cells characterized by within-trial fluctuations.

In summary:

- 1) The authors provide convincing evidence that part of the IC neurons switch between two states of activity related to the activity evoked if the stimuli were shown individually.
- 2) These switches are not frequent: For most of the triplets (and neurons) they did not occur during the investigated and behaviorally relevant trial period. In some of the other cases switching during the relevant trial period is observed, but apparently it is typically one such switch.
- 3) These observations are made for a task which does not strictly require processing (localization) of both stimuli, since one of them predicts reliably where the other one is positioned. The short intersaccade time supports the expectation that the monkeys have learned these relations and respond to each of the different sound stimuli with a specific motor program containing the two saccades.
- 4) This experimental observations in the IC are a new and interesting finding.
- 5) The attempt to interpret this observation as evidence for time division multiplexing is not convincing. With respect to the classical meaning a much higher switching frequency would be expected. Having in the majority of cases not a single switch rules out that these neurons contribute

to the quasi-simultaneous representation of two different stimuli. Also the small number of cases with perhaps something on the order of one switch within 800 or 1000 ms is not convincing in terms of the common understanding of time division multiplexing.

6) The attempt to redefine the term "time division multiplexing" fails since it is predominantly characterized by stating that it differs in some general categories (like fluidity, asynchrony of different ensembles, or aperiodicity (common also in technical systems)) from the common meaning without defining clearly the properties and functions of the new concept and how it differs precisely from the common one. Therefore it remains currently open whether such a redefinition of the term would be helpful or whether the (unknown) deviating concept would be better served by another term that avoids confusion of concepts.

7) Since the task does not direct attention at specific time periods to specific stimuli but allows for random allocation of attention with respect to the two stimuli, various types of selective attention may select either stimulus A or stimulus B and may change this at any time. Occasionally changing the attended stimulus would explain many features of the experimental observations. In particular the association between the observed response pattern and the stimulus addressed by the first saccade would be strong evidence for selective processing due to selective attention.

8) Even if one would find an interpretation in terms of selective attention not convincing or could even exclude it, this would not improve the case for time division multiplexing because it does not remove 5) and 6).

9) Consequently, at that stage and with the information available it is difficult to see how the presentation of the undoubtedly interesting data can be under the headline of time division multiplexing. Similarly, it is not entirely straight forward to do it under the headline of selective attention, since the task does not obviously fit with tasks typically chosen if attention would have been the context of the working hypothesis. However, results, even if they do not fit the expectations at the beginning of a study should be reported. This is particularly true if new and unusual patterns of data are observed that appear in itself interesting because they do not easily fit into the common view, but may open new ways to new views and ideas. Therefore the authors may consider the possibility to present their observations not as a test (and confirmation) for (whatever kind of) time division multiplexing but simply describe the interesting observation that neurons in the IC and the context of the present task switch between activity suggesting their involvement in processing of one stimulus and their involvement in processing of another stimulus for certain time periods, etc., etc. If there are sufficient simultaneously recorded neurons a more detailed description which kind of response patterns and switching behavior go along with each other or whether switches observed within trials occur simultaneously might support such an presentation of the phenomenon itself. Based on a description the authors may than carefully discuss the pros and cons of the relation to different possible functions. There is no need to come to a final conclusion for one of the possibilities. At that stage of our knowledge a comprehensive description of the phenomena and a critical discussion of their relation to possible functions would be particularly worthwhile. Of course any other way to solve the interpretational problem would be equally welcome.

Overview of response to reviewer:

We found in reading Reviewer 1's comments that while we might disagree with aspects of the reviewer's reasoning or have different takes on what the literature on attention has shown, we are not as far apart in interpreting the results of our study as it might seem. This became most clear to us when reading Reviewer 1's summary evaluation. Accordingly, we start with the summary, and will keep our responses brief.

Reviewer 1 Summary (Part II of the review)

In summary:

1) The authors provide convincing evidence that part of the IC neurons switch between two states of activity related to the activity evoked if the stimuli were shown individually.

Agreed.

2) These switches are not frequent: For most of the triplets (and neurons) they did not occur during the investigated and behaviorally relevant trial period. In some of the other cases switching during the relevant trial period is observed, but apparently it is typically one such switch.

Agreed, although we note that our analysis method likely would not be able to detect faster switching. (This was the chief argument in favor of at least looking at the spectral LFP data, since removed).

3) These observations are made for a task which does not strictly require processing (localization) of both stimuli, since one of them predicts reliably where the other one is positioned. The short intersaccade time supports the expectation that the monkeys have learned these relations and respond to each of the different sound stimuli with a specific motor program containing the two saccades.

To perform the task correctly, the monkey must at least detect whether there are 1 or 2 sounds, and, if 2 sounds, which pair of locations is involved. In the first response to reviewers (last graph under Round 1 Response to Reviewer 1, point #1), we included behavioral data from one of the two monkeys that indicated that when she was tested with sounds of two frequencies at the same location, she made only one saccade. She did this in the first session she was exposed to this stimulus configuration, with no training necessary. We thought this issue was already resolved in the first round and thus did not repeat this argument in the second round.

We also note that as yet we don't have any reason to think the task particularly matters. The visual face patch data involves fixation only and yet we see the same results; similarly the SC data that we included in the response to reviewers involves a somewhat different version of the task but produces similar results.

Additional details are provided on this point below in the response to part 1 of the review.

4) This experimental observations in the IC are a new and interesting finding.

Agreed!

5) The attempt to interpret this observation as evidence for time division multiplexing is not convincing. With respect to the classical meaning a much higher switching frequency would be expected. Having in the majority of cases not a single switch rules out that these neurons contribute to the quasi-simultaneous representation of two

different stimuli. Also the small number of cases with perhaps something on the order of one switch within 800 or 1000 ms is not convincing in terms of the common understanding of time division multiplexing.

6) The attempt to redefine the term “time division multiplexing” fails since it is predominantly characterized by stating that it differs in some general categories (like fluidity, asynchrony of different ensembles, or aperiodicity (common also in technical systems)) from the common meaning without defining clearly the properties and functions of the new concept and how it differs precisely from the common one. Therefore it remains currently open whether such a redefinition of the term would be helpful or whether the (unknown) deviating concept would be better served by another term that avoids confusion of concepts.

7) Since the task does not direct attention at specific time periods to specific stimuli but allows for random allocation of attention with respect to the two stimuli, various types of selective attention may select either stimulus A or stimulus B and may change this at any time. Occasionally changing the attended stimulus would explain many features of the experimental observations. In particular the association between the observed response pattern and the stimulus addressed by the first saccade would be strong evidence for selective processing due to selective attention.

8) Even if one would find an interpretation in terms of selective attention not convincing or could even exclude it, this would not improve the case for time division multiplexing because it does not remove 5) and 6).

9) Consequently, at that stage and with the information available it is difficult to see how the presentation of the undoubtedly interesting data can be under the headline of time division multiplexing. Similarly, it is not entirely straight forward to do it under the headline of selective attention, since the task does not obviously fit with tasks typically chosen if attention would have been the context of the working hypothesis. However, results, even if they do not fit the expectations at the beginning of a study should be reported. This is particularly true if new and unusual patterns of data are observed that appear in itself interesting because they do not easily fit into the common view, but may open new ways to new views and ideas. Therefore the authors may consider the possibility to present their observations not as a test (and confirmation) for (whatever kind of) time division multiplexing but simply describe the interesting observation that neurons in the IC and the context of the present task switch between activity suggesting their involvement in processing of one stimulus and their involvement in processing of another stimulus for certain time periods, etc., etc. If there are sufficient simultaneously recorded neurons a more detailed description which kind of response patterns and switching behavior go along with each other or whether switches observed within trials occur simultaneously might support such an presentation of the phenomenon itself. Based on a description the authors may then carefully discuss the pros and cons of the relation to different possible functions. There is no need to come to a final conclusion for one of the possibilities. At that stage of our knowledge a comprehensive description of the phenomena and a critical discussion of their relation to possible functions would be particularly worthwhile. Of course any other way to solve the interpretational problem would be equally welcome.

While we don't agree with every aspect of points 5-8, we do agree with the conclusion reached in point 9, i.e. that both the attention and multiplexing conceptual frameworks have something to offer (as well as limitations) and both should be given consideration, without reaching a strong conclusion. We had already (with enthusiasm) incorporated a detailed section on attention, and it seems that what remains is to downgrade the emphasis on the specific form of multiplexing that is encompassed by the term “time division multiplexing”. Accordingly, we have stripped this term from manuscript nearly entirely; it is now only mentioned once in the Discussion, and parenthetically at that (see lines 605-608). Instead, we chiefly refer to fluctuating activity patterns in the Abstract, Introduction, and Results. The nonspecific term “multiplexing” now appears for the first time in Results section 4 where we begin consideration of the general principles. We use either that term or a new variant, “stimulus multiplexing across time”, in the Discussion. We think this captures the essential elements of what we have been able to show without dragging along quite so much baggage.

Reviewer 1 Review (Part I of the Review):

Revision 2

The authors provided for one of the three major issues a good solution. Two others are still open. Within the manuscript they removed the text related to the resolved issue. Apart from that there is not much more than an added sentence in the introduction. Therefore I will describe the remaining problems, mainly along the rebuttal letter. In the end I sum up my view on the current state of the manuscript.

Rebuttal letter, page 1-3, First main concern

General Response, 1. and Lines 94-95 and 119-120 of manuscript

The authors' response appears to be related to one of several critical statements of the previous (first) review which stated: "(3) The task does not strictly require keeping representations of both stimuli active. If the animal recognizes that more than one stimulus is present, it would be sufficient to attend one of the two stimuli to identify its location. This location predicts the second location (a stimulus at the position close to the midline is always associated with a second stimulus at the position more distant from the midline at the other side of the midline and vice versa). Generating a motor plan for the two saccades therefore does not need to localize the position of both stimuli. Such an interpretation would be in line with the large part of trials in which no within-trial switching is observed but a constant rate fitting the expectation for one of the stimuli throughout the trial."

I agree with the authors that a very short intersaccade interval and curved saccades "indicate that the saccades are generally programmed as a planned sequence" but for the reasons given previously and above these data do not "suggest the monkey is processing both stimuli during the analysis interval". Processing one of them is entirely sufficient since the second position is predictable from the first perceived. This is in line with the large part of trials in which no within-trial switching is observed.

See response to Summary point 3. To recap: at a minimum, the task requires detecting whether there are 1 or 2 sounds; if 1, localize it; if 2, localize whether it is the pair at -24/6 or the pair at -6/24 degrees and make appropriate saccades. We provided evidence in the first round of reviews that one of the two monkeys made only one saccade in the first session in which she was tested with 2 frequencies sounds presented from a single speaker. These findings, together with the fast intersaccade intervals in the regular dual trials, support that the monkeys were performing dual localization. But even if they were only performing dual detection and some form of memorized sequence, this would not meaningfully undermine our results, particularly since the visual face data results were similar despite the monkeys not performing a stimulus response of any kind (they performed a fixation task only).

Rebuttal letter, page 3-5, General Response, 2.

The second question, whether the possibility that "the animal detect only one stimulus during the analysis period" renders the observed neural fluctuations redundant with previously demonstrated phenomena from the attention literature is difficult to relate to my requests. The possibility that the animals do not process both stimuli because the task does not require this is one of several problems detailed in the first review. Neither the first nor the second review stated, that solely this individual issue "renders the observed neural fluctuations redundant with previously demonstrated phenomena from the attention literature". The real point is that the entire set of observation appears entirely compatible with an explanation based on selective attention fluctuating sometimes slower (or not at all within the analysis period) and sometimes faster between the two stimuli.

An entirely different issue is the authors' claim, that their data selected during the presentation of both stimuli matches better with the corresponding single stimulus data. This may have several reasons, like the difference between IC and visual cortical areas or the fact that the authors used a highly advanced data analysis procedure which strongly selects the data for such a matching whereas in the classical "biased competition" paradigm all correctly performed trials in which the animal was instructed to attend a stimulus are simply averaged, certainly containing many episodes in which attention fluctuated temporarily away from the instructed target. However, it does not rule out, that the observed fluctuations between stimulus-A and stimulus-B-like responses reflect fluctuations of attention. The correspondence between the neuronal response patterns and the first saccade target is good evidence for this assumption (see first review). Similarly, other methodological

differences and interesting advantages of the method developed to analyze the data, the methods potential usefulness in attention experiments or general differences between auditory and visual brain regions do not resolve this interpretational problem.

The authors finally state “To sum up our response:, we studied new brain areas using a new task that requires preservation of multiple items, and evaluated the distribution rather than time-and-trial-pooled mean activity values. The selective attention literature has not previously demonstrated the phenomena we explore here.” While the first three claims (new brain areas, new task, and evaluation of the distribution) hold true, the claim that the attention literature would not have previously demonstrated the phenomena which the authors explore misses again the point which has been detailed several times.

(While in the attention literature usually depends on a task that directs attention to one of several stimuli, the so called divided attention tasks are closer to the authors’ task since they require the subject to attend to more than one stimulus. See the very recent work by Ricardo Kienitz et al (bioRxiv 252130; doi:

<https://doi.org/10.1101/252130>) which provides an example (and cites corresponding literature) of a divided attention task in which spontaneous activity fluctuations in the theta-range go along with attention switching between the stimuli.)

*Here we direct the reader back to the summary, and in particular point #9, our response, and the changes we have made to the manuscript. In our previous rebuttals, we focused on what has been observed in the attention literature because we thought the reviewer was arguing that our results were *confounded* by attention or otherwise not novel. We are perfectly comfortable acknowledging the conceptual connection to attention and acknowledging that we did not control attention separately from the behavioral responses in the IC task. We maintain that the specific observations we present here have not previously been reported, although we are very happy to know about the Kienitz paper as well as the Rollenhagen and Olson (2005) study cited in it, both of which show oscillations in visual unit activity in connection with multiple stimuli. Clearly, these studies are conceptually related and should be cited.*

Rebuttal letter, page 5, Superior Colliculus data

I agree with the authors that the direction of spatial attention may not have changed under the conditions in this experiment. Furthermore the authors confirm my understanding of the results that the response strength for visual and auditory stimulus differ and that despite of lacking motivation to change the direction of spatial attention fluctuations between the prototypical response levels of the two different stimuli are observed. I further agree that spatial selective attention is therefore an unlikely source for the fluctuations. However, there is a considerable literature which has demonstrated object based and feature based attention including corresponding changes in activity. The results are therefore still compatible with the likely assumption that attention has occasionally switched between the two different objects.

In summary: The central issue remains that the observed fluctuations are fully compatible with the fluctuations of attention expected for a task which does not control which stimulus is attended at which period of time. None of the arguments provided the required evidence that would exclude this straight forward explanation.

In general, we note that the phrasing “straight forward explanation” suggests that the reviewer is referencing well established findings in the literature, rather than open possibilities that do not intrinsically have a stronger footing than the possibility we are advancing. In the case of the SC data (not included in the manuscript), we are not aware of behavioral results that suggest any attentional switching between the visual and auditory components of a single object (although it is plausible that this may happen in certain circumstances). In fact, as far as we know, our findings are inconsistent with the previous SC literature on combined modality stimuli, which has shown that the responses to combined modality stimuli typically reflect a rough sum of the unimodal responses.

Rebuttal letter, page 6, Second main concern

With respect to this issue the manuscript further emphasizes the possibility of differences between the technical meaning of time division multiplexing and their usage of the term. Since no precise deviations from the common

understanding of the term are described, the concept does not become better defined. It rather loses more of the straight forward meaning which it has in its common usage.

Thus, the problem indicated in the first review remains: Time division multiplexing suggests that a certain resource for data transmission (typically a transmission line) is switched between multiple data sources with a frequency high enough to have the information of all sources available within a time span that allows a subsequent stage to assess the information of all sources as if they would be simultaneously conveyed by multiple, dedicated lines. At least the aspect of simultaneity is also pointed out in the rebuttal letter (Time division multiplexing presents a theory of how multiple items can be preserved in neural representations, ...).

This concept contrasts with the authors' experimental observation which shows for many trials that there is only one of the stimuli processed. The switching between processing the two different stimuli is therefore extremely slow (contrary to the suggestion in line 313 of the manuscript) and does not support a quasi-simultaneous presence of information at subsequent stages. The experimental evidence supports in contrast selective processing which occasionally switches between stimuli (or other content) and which is typically a consequence of switching attention. If one would extend the usual meaning of the term time division multiplexing as suggested by the authors, almost all observations of selective or flexible processing (as in the attention literature, but also in the motor system) would be examples of time division multiplexing. At the same time the term would become almost meaningless and would lose its current meaning. Since there is no lack of established terminology for the experimental observations, such redefinitions and degradations of established terminology should be avoided.

See above; term "time division multiplexing" has been nearly completely removed.

Rebuttal letter, bottom of page 6:

The problem is not whether the attention-literature demonstrated fluctuations on the single unit level for at least two reasons: 1. There is strong evidence that such fluctuations exist (See literature cited in Kienitz et al. and the literature on theta frequency switching between stimuli in divided attention tasks). 2. The task does not control the direction of attention to the two stimuli and therefore fluctuations of attention are very likely and – more important – can not be ruled out. Single unit spontaneous fluctuations have been shown by Logothetis and Coworkers also for another "well-known process" which is binocular rivalry. The problem is that the experimental results appear very well compatible with fluctuations caused by attention while features like a reasonable switching frequency as expected for time division multiplexing serving preservation of representations of multiple items are not observed.

Rebuttal letter, middle of page 7:

It is not true that attention "is thought to work similarly in the same broad set of neurons – i.e. all the neurons should show the same type of pattern, consistent with all the neurons representing the same stimulus at once." No matter whether spatial attention, feature based or object based attention is considered. There will be almost always cells which are enhanced, reduced or unaffected in their activity level or synchronization patterns, depending on the relation between their RF-properties, stimulus constellation and attention. See for example the literature on normalization models for attention.

Where in the manuscript is the evidence for simultaneously recorded neurons of different types? I did not find a description detailing how many cells of which type have been recorded simultaneously. Part of the intermediate cases (those of the "flat central" type) are conceptually reconcilable "flat extreme" type of the "mixture" cases. A similar kind of speculative explanation of the different types of cells could be suggested for an interpretation in terms of fluctuating attention: Whole trial fluctuations would characterize an output stage that would try to keep the information generated sequentially by the set of cells characterized by within-trial fluctuations.

Our general point was that "attention" refers to the cognitive state of the animal. If individual neurons do different things from each other at the same moments in time, then those differences would not fall under the heading of attention as the term is commonly used.

Regarding the last point, this is already in the manuscript: "It is possible that within-trial fluctuating units lie at the input stage of such a circuit, and that between-trial fluctuating units actually lie at the output stage." (Discussion, lines 600-602).